# Cretaceous dinosaur bone contains recent organic material and provides an environment conducive to microbial communities

Evan T Saitta[1]*, Renxing Liang[2], Maggie CY Lau[2,3†], Caleb M Brown[4], Nicholas R Longrich[5,6], Thomas G Kaye[7], Ben J Novak[8], Steven L Salzberg[9,10,11], Mark A Norell[12], Geoffrey D Abbott[13], Marc R Dickinson[14], Jakob Vinther[15,16], Ian D Bull[17], Richard A Brooker[15], Peter Martin[18], Paul Donohoe[13], Timothy DJ Knowles[17,19], Kirsty EH Penkman[14], Tullis Onstott[2]

[1]Integrative Research Center, Section of Earth Sciences, Field Museum of Natural History, Chicago, United States; [2]Department of Geosciences, Princeton University, Princeton, United States; [3]Institute of Deep-Sea Science and Engineering, Chinese Academy of Sciences, Sanya, China; [4]Royal Tyrrell Museum of Palaeontology, Drumheller, Canada; [5]Department of Biology and Biochemistry, University of Bath, Bath, United Kingdom; [6]Milner Centre for Evolution, University of Bath, Bath, United Kingdom; [7]Foundation for Scientific Advancement, Sierra Vista, United States; [8]Revive and Restore, San Francisco, United States; [9]Department of Biomedical Engineering, Center for Computational Biology, McKusick-Nathans Institute of Genetic Medicine, Johns Hopkins University, Baltimore, United States; [10]Department of Computer Science, Center for Computational Biology, McKusick-Nathans Institute of Genetic Medicine, Johns Hopkins University, Baltimore, United States; [11]Department of Biostatistics, Center for Computational Biology, McKusick-Nathans Institute of Genetic Medicine, Johns Hopkins University, Baltimore, United States; [12]Division of Paleontology, American Museum of Natural History, New York, United States; [13]School of Natural and Environmental Sciences, Newcastle University, Newcastle upon Tyne, United Kingdom; [14]Department of Chemistry, University of York, York, United Kingdom; [15]School of Earth Sciences, University of Bristol, Bristol, United Kingdom; [16]School of Biological Sciences, University of Bristol, Bristol, United Kingdom; [17]School of Chemistry, University of Bristol, Bristol, United Kingdom; [18]School of Physics, University of Bristol, Bristol, United Kingdom; [19]School of Arts, University of Bristol, Bristol, United Kingdom

*For correspondence:
evansaitta@gmail.com

Present address: †Institute of Deep-Sea Science and Engineering, Chinese Academy of Sciences, Sanya, China

Competing interests: The authors declare that no competing interests exist.

**Abstract** Fossils were thought to lack original organic molecules, but chemical analyses show that some can survive. Dinosaur bone has been proposed to preserve collagen, osteocytes, and blood vessels. However, proteins and labile lipids are diagenetically unstable, and bone is a porous open system, allowing microbial/molecular flux. These 'soft tissues' have been reinterpreted as biofilms. Organic preservation versus contamination of dinosaur bone was examined by freshly excavating, with aseptic protocols, fossils and sedimentary matrix, and chemically/biologically analyzing them. Fossil 'soft tissues' differed from collagen chemically and structurally; while degradation would be expected, the patterns observed did not support this. 16S rRNA amplicon sequencing revealed that dinosaur bone hosted an abundant microbial community different from lesser abundant communities of surrounding sediment. Subsurface dinosaur bone is a relatively

fertile habitat, attracting microbes that likely utilize inorganic nutrients and complicate identification of original organic material. There exists potential post-burial taphonomic roles for subsurface microorganisms.

## Introduction

Fossils have traditionally been thought to retain little original organic material after undergoing decay and diagenesis. However, recent discoveries of relatively intact macromolecular organic material in fossils and sub-fossils challenge this view. These include ancient DNA (*Meyer et al., 2012*; *Orlando et al., 2013*) and peptide (*Buckley, 2015*; *Demarchi et al., 2016*; *Cappellini et al., 2018*) sequences in sub-fossils, as well as ancient biomolecules such as sterols (*Melendez et al., 2013*), melanin (*Vinther et al., 2008*), amino acids (*Curry et al., 1991*), and porphyrins (*Wiemann et al., 2018a*). These findings show that organic remains can potentially persist for thousands to millions of years, depending on the biomolecules and environmental conditions. Such remains have already provided important insights into evolution, including the origins of our species (*Krause et al., 2010*) and the affinities of extinct Pleistocene megafauna (*Welker et al., 2015*). In theory, millions to tens of millions of years old organic remains could offer palaeontologists new insights and a unique window into the biology of organisms distantly related to any living species. Such organic molecular fossils could potentially shed light on the biology and evolution of extinct organisms, including their coloration, structure, behavior, and phylogeny, providing unique insights into past life, and the origins of present life.

However, it remains unclear how long different types of organic molecules and organic structures can survive and under which conditions. DNA, which is relatively unstable, is thought to persist no longer than a million years under optimal conditions (*Orlando et al., 2013*). In comparison, structural proteins such as collagen are more stable, however, and are predicted to persist for longer (*Nielsen-Marsh, 2002*), although how much longer is unclear. Pigments such as melanin and porphyrins are highly stable and can persist for hundreds of millions of years (*Gallegos and Sundararaman, 1985*; *Vinther, 2015*).

Dinosaur bone has previously been reported to contain endogenous organic remains such as DNA, collagen, osteocytes, erythrocytes, and blood vessels (*Pawlicki et al., 1966*; *Pawlicki and Nowogrodzka-Zagórska, 1998*; *Schweitzer et al., 2005a*; *Schweitzer et al., 2005b*; *Schweitzer et al., 2007a*; *Schweitzer et al., 2007b*; *Schweitzer et al., 2008*; *Schweitzer et al., 2009*; *Schweitzer et al., 2013*; *Schweitzer et al., 2014*; *Schweitzer et al., 2016*; *Asara et al., 2007*; *Organ et al., 2008*; *Schweitzer, 2011*; *Bertazzo et al., 2015*; *Cleland et al., 2015*; *Schroeter et al., 2017*). These reports, if verified, could change the study of macroevolution and the physiology of extinct organisms, particularly considering the potential for protein sequence data to shed light on the biology and systematics of extinct organisms. Most of these reports rely on structural observations, mass spectrometry, and immunohistochemistry.

Sub-fossil and fossil vertebrate remains are primarily composed of bone, dentine, and/or enamel. These represent calcified tissues with both a mineral component, primarily calcium phosphate, and a protein component that is dominated by collagen. As such collagen is a common target in the analysis of ancient organic remains. Collagen is also non-labile relative to many other vertebrate proteins because of its decay resistance, partly due to its triple helical quaternary structure and high concentration of thermally stable amino acids (*Engel and Bächinger, 2005*; *Persikov et al., 2005*; *Sansom et al., 2010*; *Wang et al., 2012*), and it is therefore reasonable to predict that collagen would be more resistant to microbial decay and diagenesis than many other proteins.

Others have criticized reports of ancient collagen based on mass spectrometric results, suggesting that they represent laboratory or environmental contamination (*Buckley et al., 2008*; *Buckley et al., 2017*; *Bern et al., 2009*) or statistical artefacts (*Pevzner et al., 2008*). The use of antibodies to detect ancient collagen is also problematic since they are known to cause occasional false positives (*True, 2008*) and have been suggested to do so in fossil samples (*Saitta et al., 2018*). Furthermore, various organic and inorganic demineralization products of fossil bone that morphologically resemble blood vessels, osteocytes, and erythrocytes have alternatively been identified as biofilm or a network of microbiological materials (*Kaye et al., 2008*), degraded and distorted

**eLife digest** The chances of establishing a real-world Jurassic Park are slim. During the fossilization process, biological tissues degrade over millions of years, with some types of molecules breaking down faster than others. However, traces of biological material have been found inside some fossils. While some researchers believe these could be the remains of ancient proteins, blood vessels, and cells, traditionally thought to be among the least stable components of bone, others think that they have more recent sources. One hypothesis is that they are in fact biofilms formed by bacteria.

To investigate the source of the biological material in fossil bone, Saitta et al. performed a range of analyses on the fossilized bones of a horned dinosaur called *Centrosaurus*. The bones were carefully excavated in a manner to reduce contamination, and the sediment the bones had been embedded in was also tested for comparison. Saitta et al. found no evidence of ancient dinosaur proteins. However, the fossils contained more organic carbon, DNA, and certain amino acids than the sediment surrounding them. Most of these appeared to have a very recent source.

Sequencing the genetic material revealed that the fossil had become a habitat for an unusual community of microbes that is not found in the surrounding sediment or above ground. These buried microbes may have evolved unique ways to thrive inside fossils. Future work could investigate how these unusual organisms live and whether the communities vary in different parts of the world.

organic contamination (*Saitta et al., 2017a*), or minerals such as pyrite/iron oxide framboids (*Martill and Unwin, 1997*; *Kaye et al., 2008*).

Reports of dinosaur protein and complex organic structure preservation are problematic for several reasons. Firstly, it remains unclear how such organics would be preserved for tens of millions of years. If endogenous, putative dinosaur soft tissues should contain diagenetically unstable proteins and phospholipids (*Bada, 1998*; *Briggs and Summons, 2014*), vulnerable to hydrolysis (*Eglinton and Logan, 1991*; *Zuidam and Crommelin, 1995*), although the released fatty acid moieties from phospholipids could be stabilized through in situ polymerization into kerogen-like aliphatic structures (*Stankiewicz et al., 2000*; *Gupta et al., 2006a*; *Gupta et al., 2006b*; *Gupta et al., 2007a*; *Gupta et al., 2007b*; *Gupta et al., 2008*; *Gupta et al., 2009*). At 25°C and neutral pH, peptide bond half-lives from uncatalyzed hydrolysis are too short to allow for Mesozoic peptide preservation, although hydrolysis rates can be decreased through terminal modifications and steric effects on internal bonds (*Kahne and Still, 1988*; *Radzicka and Wolfenden, 1996*; *Testa and Mayer, 2003*). Estimates based on experimental gelatinization suggest that, even when frozen (0°C), relatively intact collagen has an upper age limit of only 2,700,000 years (*Nielsen-Marsh, 2002*). Secondly, the instances of dinosaur peptide preservation reported are older than the oldest uncontested protein preservation reported by at least an order of magnitude. The oldest non-controversial peptides include partially intact peptides from 3.4 Ma in exceptionally cold environments (*Rybczynski et al., 2013*), as well as short peptides bound to eggshell calcite crystals from 3.8 Ma stabilized via unique molecular preservation mechanisms (*Demarchi et al., 2016*). The youngest non-avian dinosaur bones are 66 million years old; on both theoretical and empirical grounds, it seems exceptional that original proteins could persist for so long.

Furthermore, a long-term trend of protein loss and increasing contamination in ancient organismal remains, such as bone, has been shown (*Armstrong et al., 1983*; *Dobberstein et al., 2009*; *High et al., 2015*; *High et al., 2016*). Fossil bones are open systems capable of organic and microbial flux (*Bada et al., 1999*). Such a system might lead not only to the loss of endogenous organics, but also to the influx of subsurface microorganisms that could complicate the detection of any surviving organics, as well as potentially metabolizing them. The possibility of a microbiome inhabiting fossil bone is very high, especially given that decades of research have revealed the existence of a substantial 'deep biosphere' of living microorganisms actively degrading everything organic from shallow soil organic matter to deeply buried petroleum (*Onstott, 2016*; *Magnabosco et al., 2018*), even in million year old permafrost (*Amato et al., 2010*).

Since there are theoretical and empirical reasons to believe that dinosaur organics are unlikely to persist for tens of millions of years, and given the potential for contamination, we argue that the null

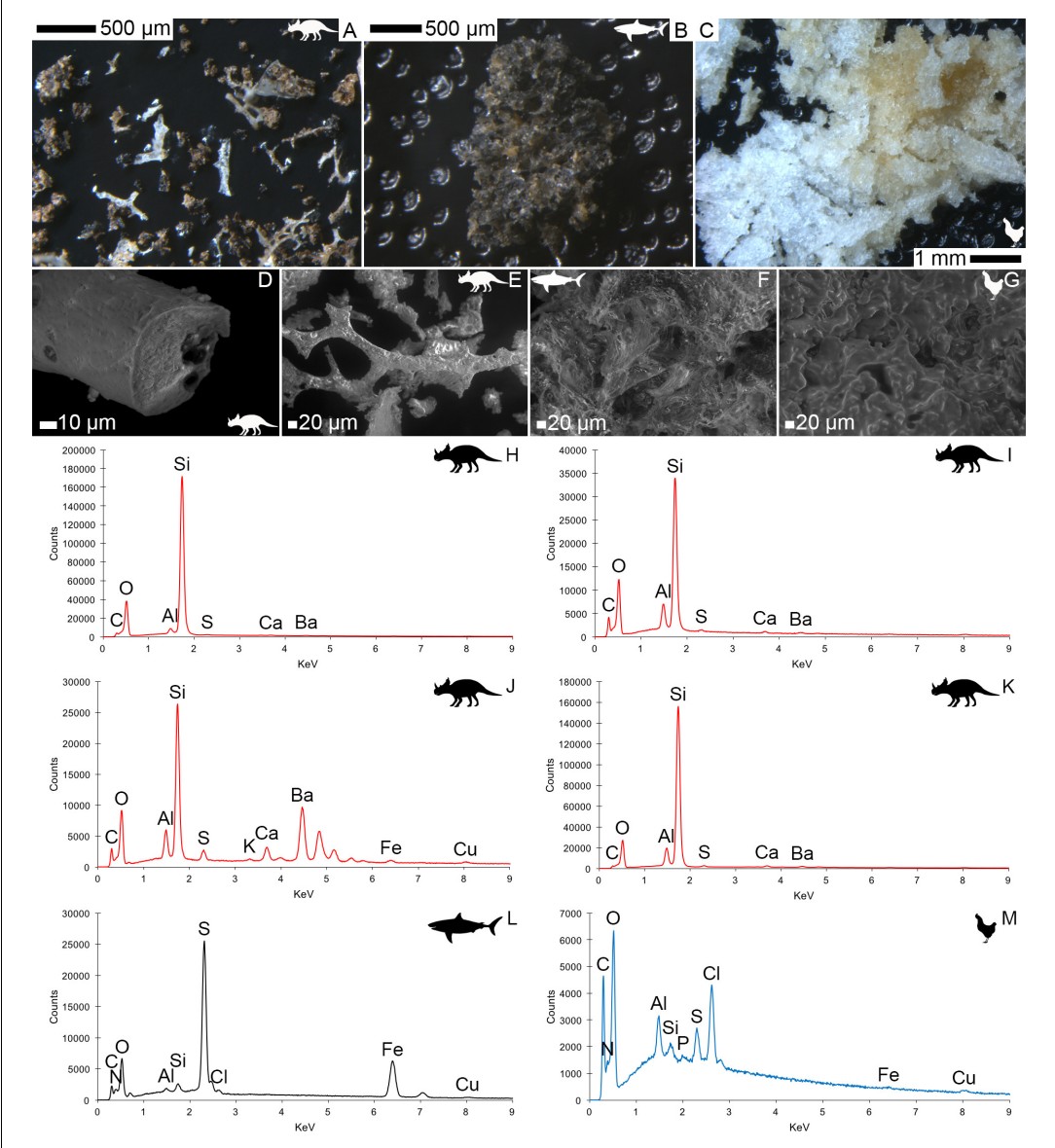

**Figure 1.** Light microscopy (**A–C**) and VPSEM (**D–G**) images and EDS spectra (**H–M**) of HCl demineralized, freeze-dried samples. (**A–C**) Samples rested on carbon tape upon SEM stubs and the pitting was a result of prior VPSEM and EDS analysis. (**A**) *Centrosaurus* vessels and associated minerals. (**B, F, L**) *Carcharias* tooth. (**C, G, M**) *Gallus*. (**D**) Infilled *Centrosaurus* vessel. (**E**) *Centrosaurus* vessel, fibrous material along the center of the vessel, and associated reddish minerals around the vessel. (**H**) *Centrosaurus* vessel exterior from D. (**I**) *Centrosaurus* vessel infilling from D. (**J**) Associated reddish mineral in *Centrosaurus*. (**K**) *Centrosaurus* fibrous material from E. *Centrosaurus* samples are matrix-surrounded subterranean bone.

hypothesis is that complex biomolecules (e.g. nucleic acids or proteins) recovered from dinosaur bones are not original material, more likely representing recent contamination. This hypothesis makes a series of testable predictions: (1) organic material recovered from fossil dinosaur bone will differ in composition (both chemistry and structure) from modern vertebrate proteins and tissues, beyond differences expected through normal diagenesis; (2) fossils will show evidence for microbial presence (e.g., through nucleic acids or protein); (3) fossil bone organic material will show signatures of recent biological activity (e.g. L-amino acid dominance or $^{14}C$ abundance, which would suggest that the fossils are not isolated from surface processes).

Here, chemical and molecular analyses of freshly collected, aseptically acquired, Late Cretaceous surface-eroded and excavated subterranean dinosaur bones, when compared to associated

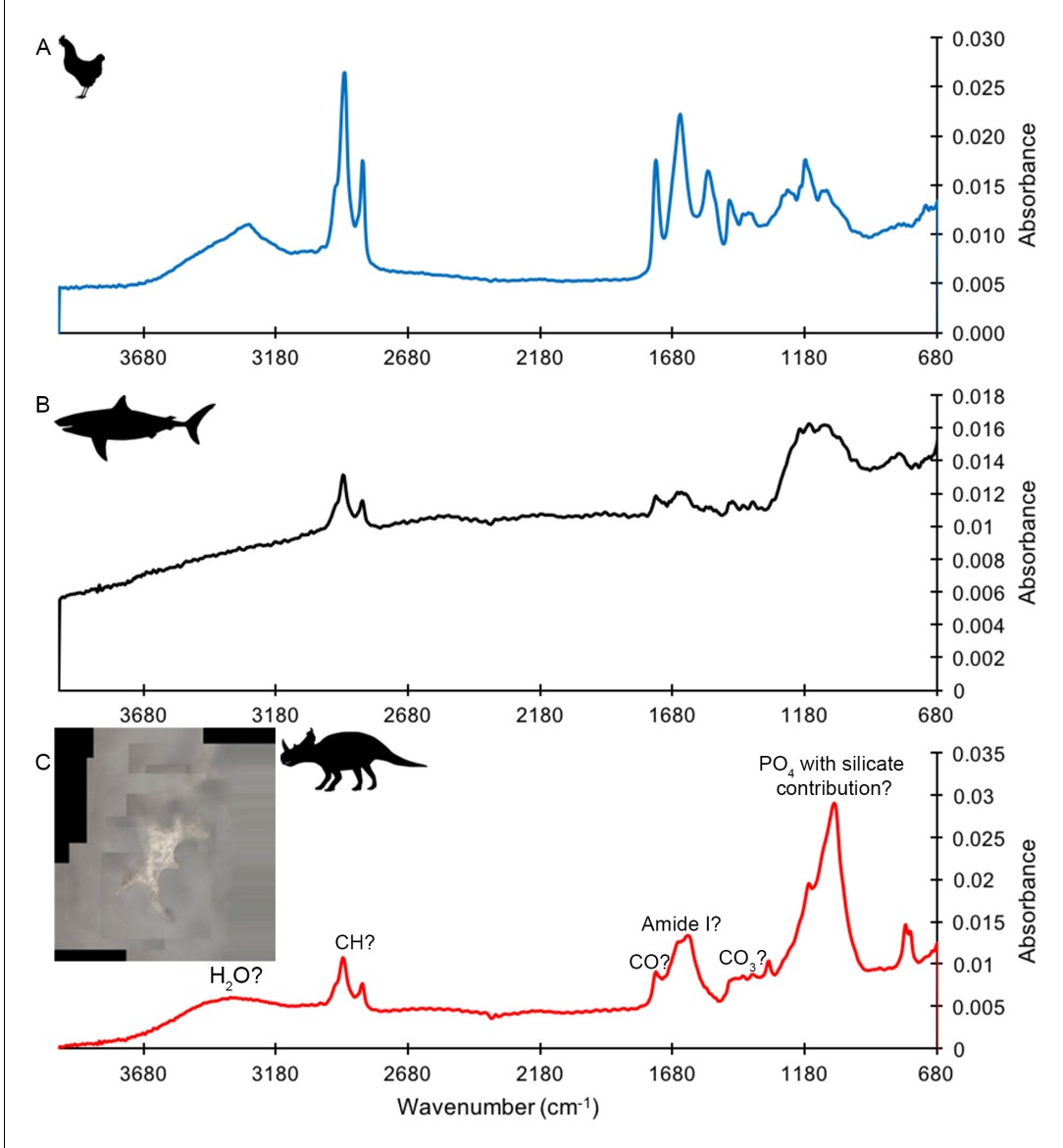

**Figure 2.** ATR FTIR spectra of HCl demineralized, freeze-dried samples. (**A**) Gallus. (**B**) Carcharias tooth. (**C**) Matrix-surrounded subterranean *Centrosaurus* bone vessel with inset showing a composite image of the vessel that was analyzed.

sediment and soil, younger fossil, and modern bone controls, show evidence for a contemporary microbiome. Analyses were conducted using variable pressure scanning electron microscopy (VPSEM), energy dispersive X-ray spectroscopy (EDS), light microscopy, attenuated total reflectance Fourier-transform infrared spectroscopy (ATR FTIR), pyrolysis-gas chromatography-mass spectrometry (Py-GC-MS), high-performance liquid chromatography (HPLC), radiocarbon accelerator mass spectrometry (AMS), Qubit fluorometer, epifluorescence microscopy (propidium iodide (PI) and SYTO 9 staining), and 16S rRNA gene amplicon sequencing.

In addition to finding little evidence for the preservation of original proteinaceous compounds, our findings suggest that bones not only act as open systems just after death and exhumation, but also act as favorable habitats as fossils in the subsurface. Microbial communities appear to be localized inside the dinosaur bones collected here.

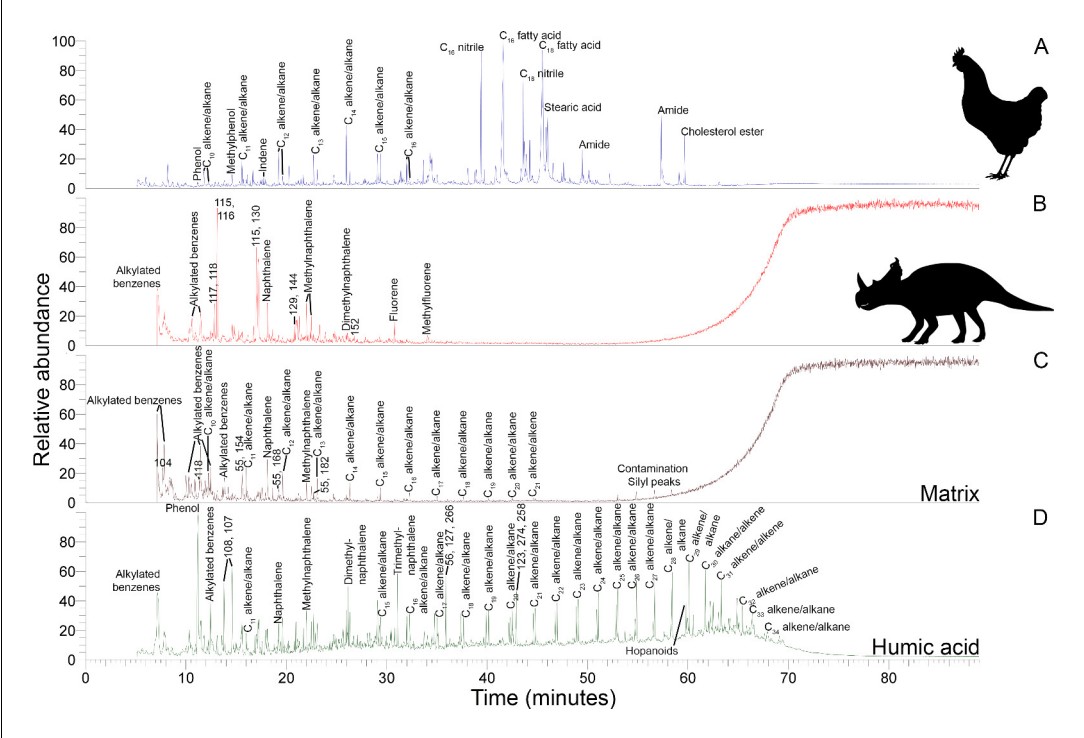

**Figure 3.** Py-GC-MS total ion chromatograms of samples ethanol rinsed before powdering. Some of the major pyrolysis products are labeled with the compound name or prominent m/z peaks. (**A**) Gallus bone. (**B**) Matrix-surrounded subterranean *Centrosaurus* bone. (**C**) Adjacent mudstone matrix of subterranean *Centrosaurus* bone in B. (**D**) Humic acid (technical grade) powder with a series of branched and cyclic alkanes, several aromatic ions, and several hopanoid (m/z = 191, 189, 367) and steroid (m/z = 217, 129, 257) ions.

## Materials and methods

For details on the analytical methods of ATR FTIR, light microscopy, VPSEM, EDS, Py-GC-MS, HPLC amino acid analysis, radiocarbon AMS, DNA extraction, 16S rRNA amplicon sequencing, and epi-fluoresence microscopy see Appendix 1.

### Fossil acquisition

Samples of Late Cretaceous fossil dinosaur bone, along with associated sediment and soil controls were obtained from the Dinosaur Park Formation (Late Campanian) in Dinosaur Provincial Park, Alberta, Canada (*Appendix 1—figures 8–20*, *Appendix 1—table 2*). The Dinosaur Park Formation is a well-sampled, alluvial-paralic unit deposited during a transgressive phase of the Western Interior Seaway. A diverse vertebrate fauna has been documented from the formation by more than a century of collection (*Currie and Koppelhus, 2005*). The bone samples were collected from a monodominant bonebed (BB180) of the centrosaurine *Centrosaurus apertus* (Ornithischia; Ceratopsidae), located 3 m above the contact with the underlying Oldman Formation (precise location data available at the Royal Tyrrell Museum of Palaeontology). The mudstone-hosted bone-bearing horizon is an aggregation of disarticulated but densely packed bones, with a vertical relief of 15–20 cm. Similar to other ceratopsid bonebeds from the same stratigraphic interval (*Ryan et al., 2001*; *Eberth and Getty, 2005*), the recovered skeletal remains are nearly exclusively from Ceratopsidae, and with all diagnostic ceratopsid material assignable to *Centrosaurus apertus*, with the site interpreted as a mass-death assemblage. Fossil material was collected under a Park Research and Collection Permit (No. 16–101) from Alberta Tourism, Parks and Recreation, as well as a Permit to Excavate Palaeontological Resources (No. 16–026) from Alberta Culture and Tourism and the Royal Tyrrell Museum of Palaeontology, both issued to CM Brown.

Sandstone and mudstone overburden was removed with pick axe and shovel (~1 m into the hill and ~1 m deep) to expose a previously unexcavated region of the bonebed, stopping within ~10 cm of the known bone-bearing horizon. A few hours after commencement of overburden removal,

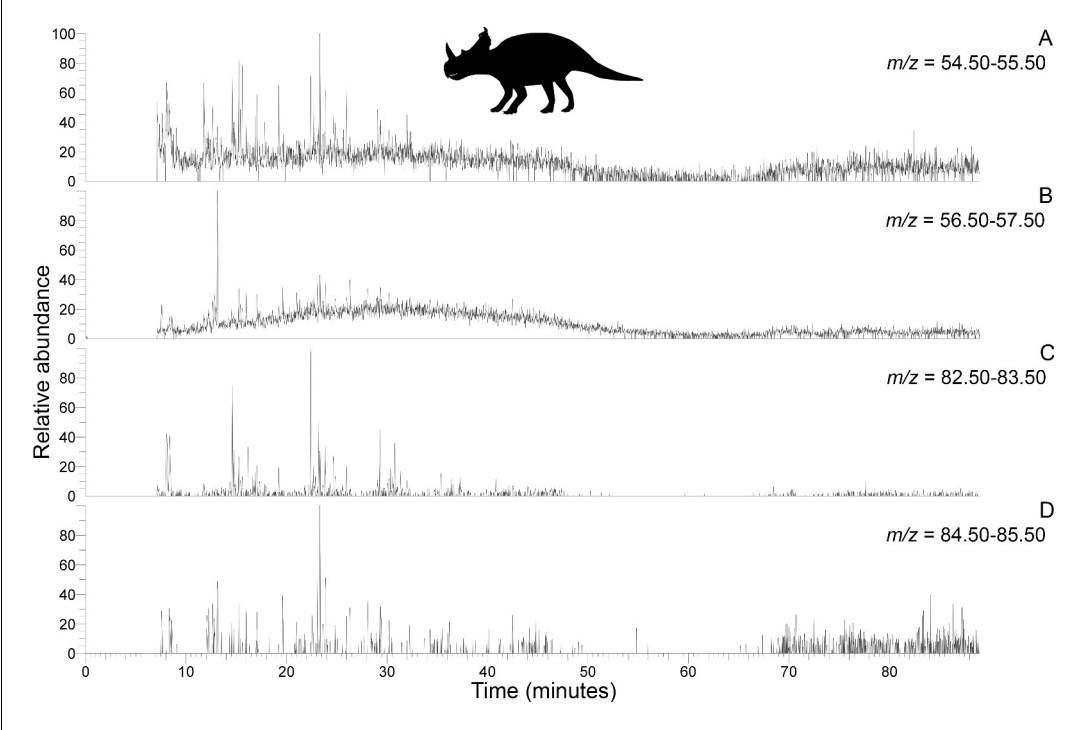

**Figure 4.** Py-GC-MS chromatograms searching for ion m/z ranges typical of n-alkanes and n-alkenes from kerogen in the matrix-surrounded subterranean *Centrosaurus* bone ethanol rinsed before powdering. Potential doublets indicative of n-alkanes/n-alkenes are weakly apparent at best. A, m/z = 55. B, m/z = 57. C, m/z = 83. D, m/z = 85.

excavation of the mudstone to the bone-bearing horizon was conducted using awl and scalpel. Subterranean *Centrosaurus* bones (identified as a small rib and a tibia) were first discovered and exposed to the air under typical paleontological excavation techniques to allow for rapid detection of bones.

At this point, aseptic techniques were then implemented to expose more of the bone in order to determine its size and orientation. It is necessary to qualify the usage of the term 'aseptic' in this study. Paleontological field techniques have changed little over the last century, and it is practically impossible to excavate fossils in a truly sterile manner (e.g. the process of matrix removal induces exposure, the wind can carry environmental contaminants onto exposed fossils, etc.). Considering this, the term 'aseptic' is used here to acknowledge the inability to provide completely sterile sampling conditions, while still indicating that efforts were taken to minimize contamination of the samples. Our success at reasonably reducing contamination is supported by the fact that our samples yielded consistent and interpretable results.

During aseptic excavation and sampling, nitrile gloves washed in 70% ethanol and a facemask were worn. All tools (i.e. awl, scalpel, Dremel saw) were sterilized with 10% bleach, followed by 70% ethanol, and then heat-treated with a propane blowtorch at the site. Bone samples several centimeters long were obtained using a diamond-coated Dremel saw or utilizing natural fractures in the bone. Certain segments of the bones, designated herein as matrix-surrounded subterranean *Centrosaurus* bone, were sampled without first removing the surrounding matrix, although fractures in the mudstone did appear during sampling so that the samples cannot be said to have been unexposed to the air, especially prevalent in the small rib sample sent to Princeton University for analysis. Also sampled were the aseptically excavated but completely exposed portions of the subterranean bone immediately next to the matrix-surrounded region, designated herein as uncovered subterranean *Centrosaurus* bone. In other words, these were the regions of the bone fully exposed using aseptic techniques after initial discovery of the bone in order to determine size and orientation. All samples were collected in autoclaved foil without applying consolidants, placed in an ice cooler kept in the shade, and brought back to the field camp freezer that evening.

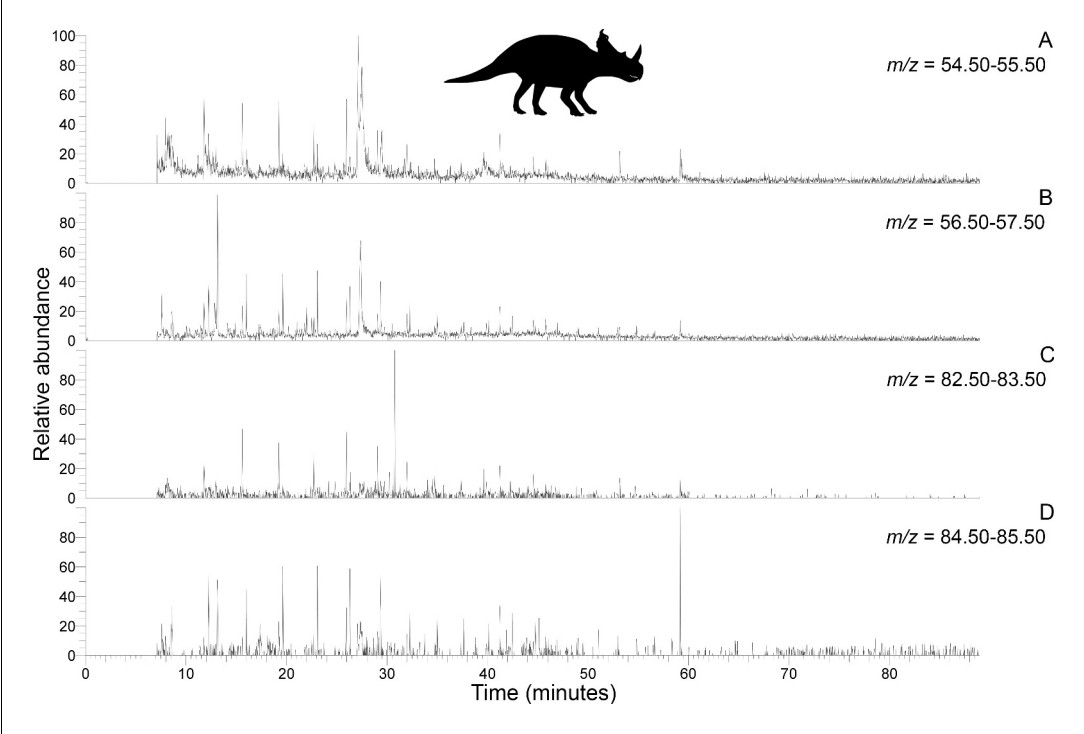

**Figure 5.** Py-GC-MS chromatograms searching for ion m/z ranges typical of n-alkanes and n-alkenes from kerogen in the uncovered subterranean *Centrosaurus* bone ethanol rinsed before powdering. Doublets indicative of n-alkanes/n-alkenes are relatively more abundant than in *Figure 4*. A, m/z = 55. B, m/z = 57. C, m/z = 83. D, m/z = 85..

Additionally, surface-eroded bone from BB180 and on the same ridge above BB180, mudstone excavated from the overburden-removed area of BB180 and several cm below the weathered surface of the same ridge above BB180, and topsoil on the same ridge above BB180 were similarly aseptically acquired and stored (i.e. sterile tools, foil, and personal wear; kept cool). In total, eight bone samples, eight sediment samples, and two soil samples were collected.

Samples were transported to the Royal Tyrrell Museum of Palaeontology in a cooler. Following accession at the museum, similar sets of samples were either mailed to Princeton on ice or transported via plane to Bristol without refrigeration with a maximum time unrefrigerated under 24 hr (i.e. both Princeton and Bristol received a sample of matrix-surrounded bone, BB180 mudstone, topsoil, etc.). Note that warming after cold storage could lead to condensation, altering the behavior of any potential microbiome. Upon arrival, samples were stored at 4°C in Bristol or −80°C in Princeton as required for analysis.

The aseptically collected Dinosaur Provincial Park fossil bone, mudstone, and soil samples were compared to younger fossils and modern bone. Chicken (*Gallus gallus domesticus*) bone was obtained frozen from a Sainsbury's grocery store in Bristol, UK and was kept refrigerated (4°C). Other controls included amino acid composition data from a reference bone (fresh, modern sheep long bone) and radiocarbon data from an 82–71 ka radiocarbon-dead bovine right femur used as a standard from the literature (*Cook et al., 2012*). Black, fossil sand tiger shark teeth (*Carcharias taurus*) eroded from Pleistocene-Holocene sediments were non-aseptically collected from the surface of the sand on a beach in Ponte Vedra Beach, Florida, USA without applied consolidants and stored at room temperature. It should be noted that Florida experiences a high-temperature climate relative to many samples typically studied for palaeoproteomics. Teeth samples represent a mix of dentine and enamel as opposed to normal bone tissue, with relative concentrations depending on how easily the different tissues fragmented during powdering. The decision to include subfossil shark teeth was made based on their ready availability (i.e. they are incredibly common fossils and are easy to collect from the surface of the sand), the minimal loss to science when destructively analyzed due to their

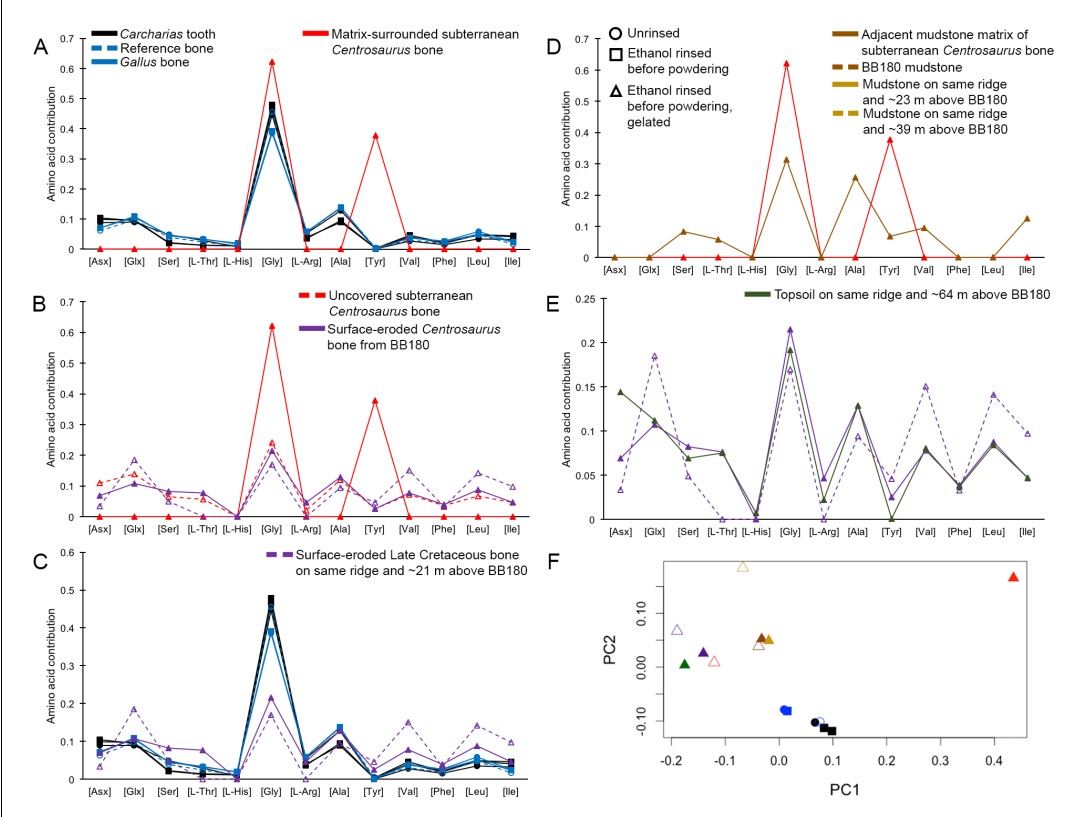

**Figure 6.** THAA compositional profiles of the KOH-treated samples based on amino acid percentages. Lines connecting points are added to aid visualization. (A) Late Cretaceous subterranean bone compared to non-aseptically collected Pleistocene-Holocene teeth (with a repeated measurement for the ethanol rinsed sample) and modern bone. (B) Late Cretaceous subterranean bone compared to surface-eroded Late Cretaceous bone from the same outcrop. (C) Surface-eroded Late Cretaceous bone compared to Pleistocene-Holocene teeth and modern bone. (D) Late Cretaceous subterranean bone aseptically collected compared to the adjacent mudstone matrix. (E) Surface-eroded Late Cretaceous bone compared to topsoil at higher elevation (i.e., prairie level) on the same ridge. (F) PCA on non-normalized amino acid percentages (i.e. percentages that do not require further normalization) (see A–E legends). PC1 and PC2 describe 55.04% and 22.66% of the data variation, respectively. See Appendix 1 (*Appendix 1—figure 21*; *Appendix 1—table 9*) for PCA summary. Color and symbol coding is constant throughout.

ubiquity, and that the protein composition of the tooth dentine would be dominated by collagen, as in bone. Technical grade humic acid was also purchased from Sigma Aldrich as an additional control.

## Results

### Light microscopy, VPSEM, and EDS of HCl demineralized bone

VPSEM and EDS of HCl demineralized, freeze-dried dinosaur bones revealed that vessels (and rare fibrous fragments) (*Figure 1A,D–E,H–J*) were white, Si-dominated with O present, contained holes, and were sometimes infilled with a slightly more prominent C peak internally. Vessels occurred alongside white quartz crystals, which had strong Si peaks and overall were elementally similar to the vessels, and smaller reddish minerals, originally presumed to be iron oxide or pyrite, but which had high-Si content with Ba also present.

Demineralization products differed from those of chicken bone (*Figure 1C,G,M*) and Pleistocene-Holocene shark tooth (*Figure 1B,F,L*), which were much more homogenous and consisted of large fibrous masses. These more recent samples were enriched in C, O, N, and S, but the shark tooth also had a strong Fe signature and a relatively more prominent S peak than the chicken bone. The chicken demineralisation product was white, while that of the shark tooth was black.

These results show that the dinosaur bone yielded different structures when the bone apatite was removed compared to the more recent bone (i.e. primarily vessels as opposed to large fibrous

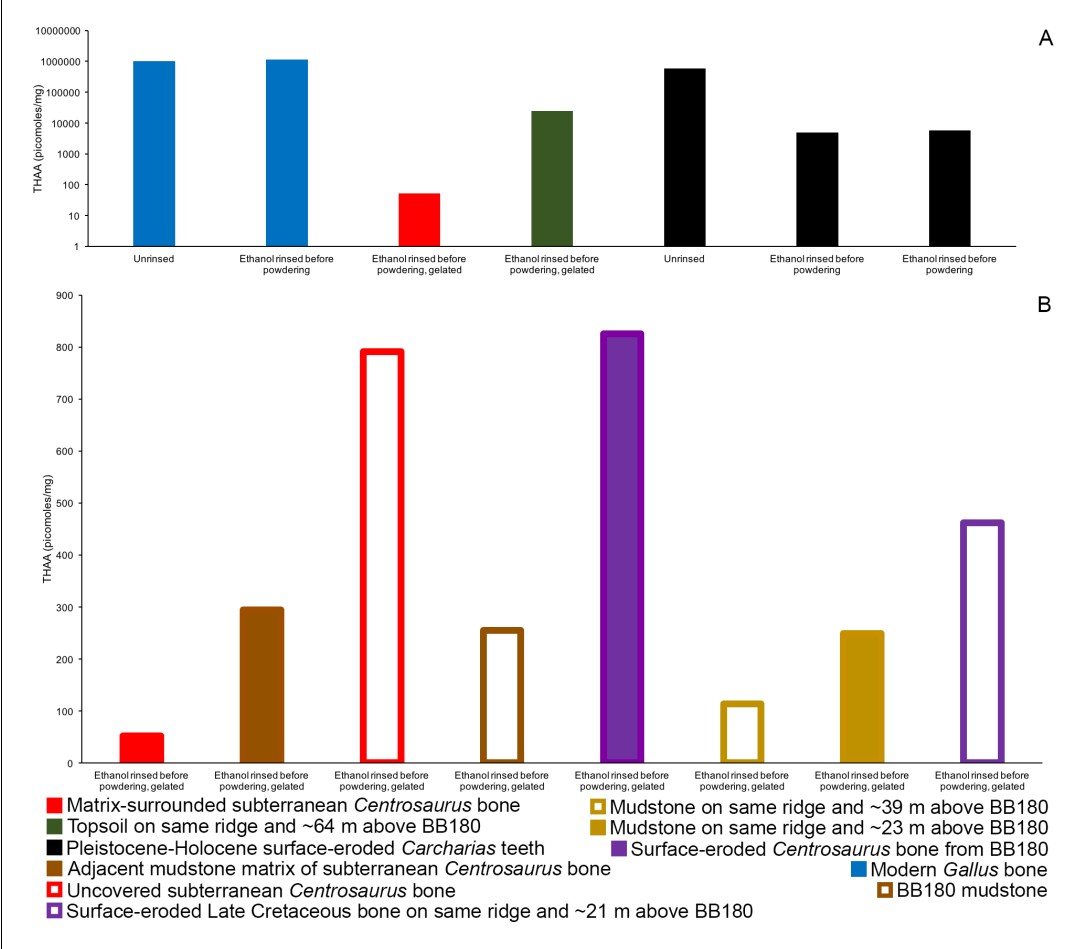

**Figure 7.** THAA concentrations (summed total of all amino acids measured) of the KOH-treated samples. (**A**) Logarithmic scale comparison of modern bone, matrix-surrounded subterranean Late Cretaceous bone, Pleistocene-Holocene surface-eroded teeth (with a repeated measurement for the ethanol rinsed sample), and topsoil on same ridge and ~64 m above BB180. (**B**) Comparison between fossil Late Cretaceous bone and mudstone.

masses). Furthermore, the dinosaur vessels are relatively inorganic in composition compared to the more recent bone, consistent with a mineralized biofilm (*Schultze-Lam et al., 1996*; *Decho, 2010*).

## ATR FTIR of HCl demineralized bone

ATR FTIR of a HCl demineralized, freeze-dried vessel from subterranean *Centrosaurus* bone revealed somewhat poorly resolved, broad organic peaks (*Figure 2C*) that were close in position to peaks that might be expected from various CH, CO, and amide bonds, as well as water, phosphate, and potentially carbonate and silicate bonds (*Lindgren et al., 2011*; *Surmik et al., 2016*; *Lee et al., 2017*); also see publicly available NIST libraries). Pleistocene-Holocene shark tooth (*Figure 2B*) and modern chicken bone (*Figure 2A*) demineralization products similarly revealed peaks consistent with organic and phosphatic peaks, and the chicken bone had particularly strong organic peaks relative to phosphate. Maintaining close contact of the sample to the Ge crystal was difficult, resulting in the poorly resolved peaks, especially in the shark tooth sample.

These results show how, although potentially poorly resolved, the ATR FTIR peaks in the dinosaur bone demineralization products could be consistent with various organic bonds present in more recent bone demineralization products. However, note that these bonds are relatively simple and could therefore be present in various organic molecules. Furthermore, they are not necessarily ancient, endogenous, or protein-derived.

**Table 1.** Comparison of Late Cretaceous, Pleistocene-Holocene, and modern amino acid racemization values of the KOH-treated samples.

NA indicates that amino acid concentration was below detection limit.

| Sample treatment | Asx D/L | Glx D/L | Ser D/L | Ala D/L | Val D/L |
|---|---|---|---|---|---|
| Matrix-surrounded subterranean *Centrosaurus* bone | | | | | |
| Ethanol rinsed before powdering, gelated | NA | NA | NA | NA | NA |
| Subterranean *Centrosaurus* bone uncovered from matrix before collection | | | | | |
| Ethanol rinsed before powdering, gelated | 0.21 | 0.55 | 0 | 0.21 | 0 |
| Adjacent mudstone matrix of subterranean *Centrosaurus* bone | | | | | |
| Ethanol rinsed before powdering, gelated | NA | NA | 0 | 0.30 | 0 |
| Surface-eroded *Centrosaurus* bone from BB180 | | | | | |
| Ethanol rinsed before powdering, gelated | 0 | 0 | 0 | 0 | 0 |
| Surface-eroded Late Cretaceous bone on same ridge and ~ 21 m above BB180 | | | | | |
| Ethanol rinsed before powdering, gelated | 0 | 0.95 | 0 | 0.32 | 0.90 |
| Topsoil on same ridge and ~ 64 m above BB180 | | | | | |
| Ethanol rinsed before powdering, gelated | 0.14 | 0.14 | 0.05 | 0.09 | 0.04 |
| Pleistocene-Holocene surface-eroded *Carcharias* teeth | | | | | |
| Unrinsed | 0.21 | 0.04 | 0.09 | 0.03 | 0.01 |
| Ethanol rinsed before powdering | 0.51 | 0.15 | 0.30 | 0.16 | 0.11 |
| Ethanol rinsed before powdering | 0.53 | 0.15 | 0.30 | 0.17 | 0.11 |
| Modern *Gallus* bone | | | | | |
| Unrinsed | 0.05 | 0.03 | 0 | 0.02 | 0 |
| Ethanol rinsed before powdering | 0.06 | 0.03 | 0 | 0.02 | 0 |

## Py-GC-MS

Data-rich Py-GC/MS results are primarily used here as a fingerprinting method via total ion chromatograms in order to complement the other analyses of this study. *Centrosaurus* bone had a low pyrolysate yield (*Figure 3B*) as evidenced by the significant column bleed at the end of the run and contained mostly early eluting compounds. Similarly, humic acid also contained many early eluting pyrolysis products (*Figure 3D*). The pyrogram for *Centrosaurus* bone does not match that of modern collagen-containing bone (*Figure 3A*), which contained many clear protein pyrolysis products such as nitriles and amides, and was most similar to mudstone matrix (*Figure 3C*).

Subterranean *Centrosaurus* bone pyrolysates included alkylated benzenes and some polycyclic aromatic hydrocarbons (e.g. naphthalenes and fluorenes), and these can also be detected in the surrounding mudstone matrix (*Figure 3C*) and humic acid standard (*Figure 3D*). Weak *n*-alkane/*n*-alkene doublets were possibly detected in the Late Cretaceous bones (*Figure 4A–D*), and such doublets were also observed in the surrounding mudstone matrix (*Figure 3C*) and humic acid standard (*Figure 3D*). Variation in the conspicuousness of these doublets between the matrix-surrounded and uncovered subterranean *Centrosaurus* bone samples was apparent (*Figure 5A–D*).

These results show how the dinosaur bone lacked any clear pyrolysis products indicative of high levels of protein preservation and instead had a chemical composition that more closely resembles potential environmental sources (i.e. mudstone matrix or humic acids) than bone proteins. Homologous series of n-alkane/n-alkene doublets may signify the presence of a kerogen-like substance which could potentially be an ancient lipid-derived geopolymer in the dinosaur bone.

**Table 2.** Carbon data from Late Cretaceous fossil bone, mudstone, topsoil, and younger bone.

| Sample | % mass after HCl demineralization | C % (organic fraction) | F$^{14}$C (organic fraction) |
|---|---|---|---|
| Matrix-surrounded subterranean *Centrosaurus* bone core (surface scraped prior to powdering) | 53.98 | 2.777 | 0.0149 |
| Adjacent mudstone matrix of subterranean *Centrosaurus* bone | 82.27 | 1.32 | 0.0573 |
| Topsoil on same ridge and ~ 64 m above BB180 | 91.63 | 2 | 0.766 |
| Mudstone on same ridge and ~ 23 m above BB180 | 90.38 | 0.89 | 0.0628 |
| Surface-eroded Late Cretaceous bone core on same ridge and ~ 21 m above BB180 (surface scraped prior to powdering) | 43.4 | 1.63 | 0.0422 |
| Yarnton bovine right femur (82–71 ka, *Cook et al., 2012*) | 16.73 | 44.9 | 0.0056* |

*This sample was used for blank correction in the AMS analyses, therefore this value is not blank-subtracted.

## HPLC amino acid analysis

Interpretation of amino acid data is restricted here to only those samples that were prepared to counter peak suppression (KOH-treated; *Dickinson et al., 2019*), although examination of the conventionally prepared (*High et al., 2016*) samples results in similar patterns, albeit with more noise (*Appendix 1—figures 21–28*, *Appendix 1—tables 3–13*). Matrix-surrounded subterranean *Centrosaurus* bone had a total hydrolysable amino acid (THAA) compositional profile that did not match collagen (*Figure 6A,F*). The matrix-surrounded subterranean *Centrosaurus* bone appeared to be dominated by Gly, with Tyr also prominent, while being highly depleted in all the other amino acids. Surface-eroded Late Cretaceous bone from the same outcrop showed a different THAA composition to the matrix-surrounded subterranean *Centrosaurus* bone, even when examining bone eroded out of the BB180 quarry itself (*Figure 6B,F*). Furthermore, the uncovered subterranean *Centrosaurus* bone did not match the matrix-surrounded subterranean bone and was similar to the surface-eroded Late Cretaceous bone in THAA composition. Relative Gly concentration in surface-eroded Late Cretaceous bone was not as high as in the matrix-surrounded subterranean *Centrosaurus* bone, where Gly dominated the compositional profile. The surface-eroded Late Cretaceous bone showed somewhat more similarity to collagen in THAA compositional profile than did the matrix-surrounded subterranean *Centrosaurus* bone, but ultimately did not align (*Figure 6C,F*). These results suggest that not only did the subterranean dinosaur bone not have an amino acid composition similar to collagen (i.e. *Gallus* and reference bone), but also that exposure to the surface changes the amino acid profile within these Cretaceous fossils.

Subterranean *Centrosaurus* bone had a far lower THAA concentration (summed total of all amino acids measured) than did the modern chicken bone (*Figure 7A*), as would be expected, and the uncovered subterranean *Centrosaurus* bone showed a higher THAA concentration than the matrix-surrounded subterranean *Centrosaurus* bone (*Figure 7B*). Surface-eroded Late Cretaceous bone showed relatively high variability in THAA concentrations, with some higher THAA concentrations than subterranean *Centrosaurus* bone. These results are consistent with the expectation that any potential proteins present in the subterranean dinosaur bone would be reduced in concentration compared to bone in vivo, an expectation that might hold regardless of whether proteins are endogenous or exogenous.

Late Cretaceous bone tended to be L-amino acid dominated when amino acids were above detection limit (*Table 1*). Surface-eroded Late Cretaceous fossil bone seemed to show more variability in D/L values than the subterranean bone samples. Similar to the samples described here, other non-aseptically collected, room-temperature-stored Jurassic and Cretaceous surface-eroded bones have low amino acid concentrations and lack significant concentrations of D-amino acids (*Appendix 1—tables 3–4*). These low levels of racemization suggest that the amino acids in the dinosaur bone are not particularly ancient.

The adjacent mudstone matrix did not match the subterranean *Centrosaurus* bone in THAA compositional profile (*Figure 6D,F*). Surface-eroded Late Cretaceous bone showed some degree of similarity to topsoil in THAA composition (*Figure 6E,F*), as did the various mudstone samples. Matrix-surrounded subterranean *Centrosaurus* bone showed the most different THAA compositional profile

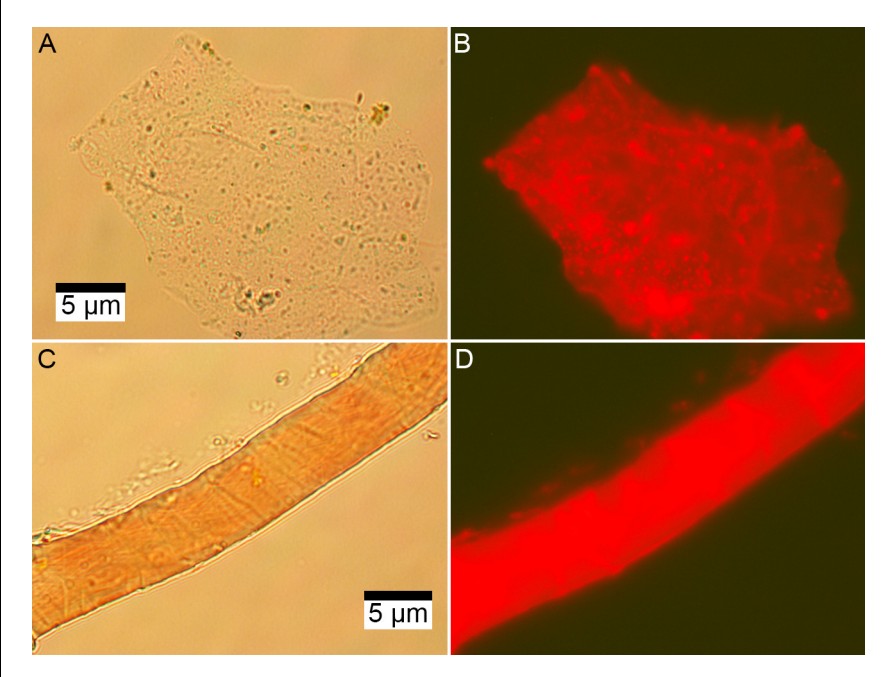

**Figure 8.** Microscopic images of EDTA demineralized, PI stained matrix-surrounded subterranean *Centrosaurus* bone. (**A–B**) Fibrous material. (**C–D**) Vessel. (**A, C**) Transmission light. (**B, D**) Fluorescence.

within the study (i.e. greatest separation from other data points in PC space). All these groups plotted separately in the PCA from modern collagen (*Figure 6F*). The greatest variation between the samples of this study was in relative Gly and Tyr concentrations, with the matrix-surrounded subterranean *Centrosaurus* bone tending to have notably higher Gly and Tyr than collagen. These results suggest that subterranean dinosaur bone had a different amino acid composition than the surrounding mudstone and that the amino acid composition changes upon surface exposure, approaching that of topsoil.

Topsoil showed a greater THAA concentration than subterranean and surface-eroded *Centrosaurus* bones, but not as high as modern chicken bones (*Figure 7A*). Mudstone tended to have a very low THAA concentration, even compared to some of the Late Cretaceous bone samples (*Figure 7B*). The highest THAA concentration in mudstone appeared to be observed in the mudstone matrix adjacent to the subterranean *Centrosaurus* bone. When amino acids were above detection limit, mudstone was L-amino acid dominated, similar to the Late Cretaceous bone (*Table 1*). Topsoil, on the other hand, showed consistently moderate levels of racemization. These results show that topsoil contained a high amino acid concentration with relatively high rates of protein degradation, indicative of active biological accumulation and recycling, while mudstone contained low concentrations with very recent amino acids, indicative of low residence times of proteins within the mudstone. The fossil bones appeared to show instances of relatively greater accumulation of amino acids than the mudstone but with very recent amino acids, indicative of preferential localization of biologically active amino acids to the bone compared to the mudstone, but with less amino acid content than topsoil.

Pleistocene-Holocene surface-eroded shark teeth had THAA compositional profiles that closely matched collagen (*Figure 6A,C,F*) and fairly high amino acid concentration with THAA concentrations between those of subterranean *Centrosaurus* bone and modern chicken bone (*Figure 7A*). Pleistocene-Holocene surface-eroded shark teeth, unlike the Late Cretaceous bone and mudstone, had consistently high racemization (*Table 1*), even more so than the topsoil sample. Ethanol rinsing appeared to lower amino acid concentration in the shark teeth but did not strongly affect the THAA compositional profile (*Figures 6A,C,F* and *7A*). These results suggest that the Pleistocene-Holocene teeth contained detectable, ancient amino acids consistent with endogenous collagen.

**Table 3.** DNA concentrations in mudstone matrix and bone quantified with Qubit fluorometry.

| Sample | Average DNA concentration (ng/µL) | Total DNA (ng) | DNA per 1 g of bone or mudstone (ng/g) |
|---|---|---|---|
| Matrix-surrounded subterranean *Centrosaurus* bone core (surface scraped prior to powdering) | 0.79 | 3965 | 793 |
| Adjacent mudstone matrix of subterranean *Centrosaurus* bone | 0.03 | 164 | 16.4 |
| Laboratory blank | Below detection (<0.01[*]) | | |

[*]Note: the detection limit corresponds to the actual concentration of DNA in the assay tube (0.0005 ng/µL) after 200 times dilution of the original sample according to the manufacturer's protocol.

## Radiocarbon AMS

Total organic carbon (TOC) content was higher in the subterranean and surface-eroded *Centrosaurus* bone than the matrix, even the directly adjacent matrix, and was comparable to that found in the topsoil (*Table 2*). However, the organic carbon content in the *Centrosaurus* bones was significantly lower than the 82–71 ka Yarnton bovine bone sample known to contain well-preserved (radiocarbon-dead) collagen (*Cook et al., 2012*). TOC in the *Centrosaurus* bone was not found to be radiocarbon dead, but did exhibit lower $F^{14}C$ values than both the mudstone and especially the topsoil. Assuming all endogenous bone C is radiocarbon 'dead', based on these $F^{14}C$ values, a simple two-end-member mixing model would suggest that ~26% of the C in subterranean *Centrosaurus* bone originates in the adjacent mudstone matrix (for formula, see Appendix 1 under the section entitled Carbon analysis).

The fossil dinosaur bone therefore yielded a TOC content similar to relatively rich environmental carbon sources, such as topsoil, but not as high as more recent bone proteins. Although, some of the C in the fossil dinosaur bone is potentially ancient, there is still a sizable contribution of recent C from the immediate environment, consistent with the presence of a microbiome.

## Fluorescence microscopy, DNA extraction, and 16S rRNA gene amplicon sequencing

DNA concentration was about 50 times higher in subterranean *Centrosaurus* bone than in adjacent mudstone matrix (*Table 3*; *Appendix 1—table 25*). PI staining for DNA on EDTA demineralized *Centrosaurus* bone revealed multi-cell aggregates forming organic vessel and fibrous conglomerate structures that fluoresce red (*Figure 8A–D*). The DNA concentration in the bone indicates a cell concentration of ~$4\times10^8$ cells/g (calculation of cell abundance from DNA based on that of *Magnabosco et al., 2018*; also see *Appendix 1—table 26*). This is fairly similar to the observed THAA concentration indicating ~$3\times10^8$ cells/g (calculation of cell abundance from total amino acids based on that of *Onstott et al., 2014* and *Lomstein et al., 2012*), consistent with the idea that the amino acids within the bone are likely to be largely cellular (i.e. lipid-bound within living organisms) due to the discrepancy between DNA and amino acid stability over time. The DNA concentration in the adjacent mudstone matrix indicate a cell concentration of ~$5\times10^6$ cells/g, but the observed THAA concentration is consistent with a cell concentration of ~$2\times10^9$ cells/g. The greater amino acid abundance is a common feature of marine sediment and likely represents the amino acids of a microbial necromass (e.g. *Braun et al., 2017*). The adjacent mudstone matrix contains amino acids that seem to largely represent dead prokaryote remains, unlike the amino acids in the dinosaur bone that seem to largely represent a more recent, likely living community in comparison (i.e. the adjacent mudstone matrix has a greater amino acid concentration relative to the DNA concentration than does the dinosaur bone). These results suggest that the subterranean dinosaur bone was enriched in cell-bound DNA relative to the mudstone matrix. Furthermore, EDTA-extracted structures appeared to contain DNA from cells that aggregate within these structures, consistent with a modern biofilm; the DNA itself had possibly been exposed due to the EDTA treatment.

The 16S rRNA gene amplicon sequencing revealed the predominance of Actinobacteria and Proteobacteria in subterranean *Centrosaurus* bone. Sequences affiliated with classes Nitriliruptoria and Deltaproteobacteria were more abundant relative to adjacent mudstone or even the surface scrapings from the bone itself (*Figure 9*). The majority of the sequences within Deltaproteobacteria were

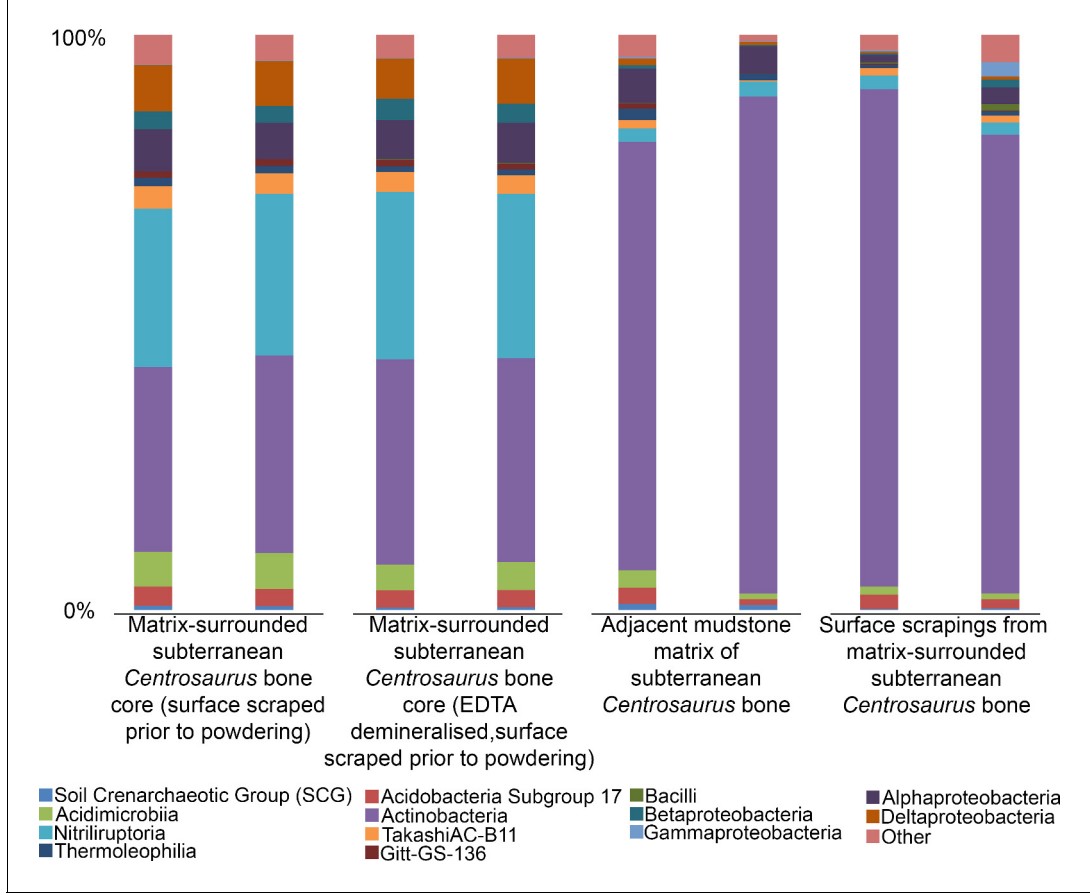

**Figure 9.** Comparison of microbial community (class level) from matrix-surrounded subterranean *Centrosaurus* bone and adjacent mudstone matrix. There are two replicates per sample. Classes with <1% representation in all replicates and samples are combined into an 'other' category.

identified as belonging to the family Desulfurellaceae, which contains some sulfur-respiring species. However, the short reads prevented species level identification. In *Centrosaurus* bone, about 30% of sequences were phylogenetically close to the genus *Euzebya*, a deeply branching, aerobic, marine Actinobacterium (*Appendix 1—figure 58*). Furthermore, PCA of the species-level percentage data from these eight samples shows that the differences between the interior bone cores and the mudstone or bone surface scrapings is greater than the difference between the mudstone and bone surface scrapings (*Appendix 1—figure 60*). Likewise, one-way permutational multivariate analysis of variance (PERMANOVA) performed in PAST3 software of species-level sequence percentages of the two replicates of each of the four sample categories in *Figure 9* yielded significant differences (Euclidean similarity index; 9999 permutations; *F* = 53.16; p-value=0.0084), with greater similarity between the mudstone and bone surface scrapings than between either of these and the interior bone core samples (*Appendix 1—table 27*). These results suggest that the subterranean dinosaur bone contained a different microbial community than the surrounding mudstone matrix with some species potentially impacting fossil bone taphonomy and chemical composition. Our initial sequence data, furthermore, suggests that some of these microbes might represent rare, poorly understood taxa.

## Discussion

### Evidence for recent, exogenous organic material in dinosaur bones

Structure and elemental composition

The occasional infilling observed in the HCl demineralized dinosaur bone vessels with greater C concentration in the interior compared to the exterior of the vessel is consistent with a growing biofilm, given the assumption that a biofilm would grow inside the porous spaces of the bone while preserved vessels might be expected to be hollow.

The Si dominance of the HCl demineralization products from the dinosaur bone likely suggest that they are at least partly silicified. HCl demineralization (especially the relatively intensive demineralization used on the samples that underwent microscopy, EDS, and ATR FTIR) may favor mineralized biofilm retrieval (assuming that low pH might degrade organically preserved biofilms), explaining why all the observed demineralization products of the dinosaur bone have high-Si content under EDS (especially in comparison to the presumably largely organic vessels and fibrous masses revealed using EDTA demineralization). If this is the case then it would indicate that any original organics are significantly more susceptible to acid (i.e. of different composition) than the organic masses in the identically treated younger bone samples, which survive well. It seems likely that mineral infilling in the *Centrosaurus* bone is largely of silicates, which have partly replaced the originally organic vessel-like structures, as well as, potentially, some barite or gypsum with minimal amounts of iron oxide or pyrite. Some of these inorganic compounds might contribute to the color of the fossils. The fibrous material from HCl demineralization of the dinosaur bone may be silicified biofilm with a collagenous texture imprinted from the surrounding apatite matrix or may simply be a small misinterpreted quartz crystal. A silicified biofilm might be a result of unique environmental conditions (either early or late in the taphonomic history) or microbial communities that these fossils experienced. Therefore, examining fossils from different localities, climates, lithologies, and taphonomic histories is vital to understanding variation in how biofilms in fossil bone might be mineralized.

Pyrolysis products

Humic acids are common in soils and contain low molecular weight, aromatic components (*Hatcher et al., 1981*; *Sutton and Sposito, 2005*), and these were also detected in the humic acid standard, meaning that the early eluting, more volatile (i.e. lower boiling point), total ion chromatogram peaks from Py-GC/MS (e.g. the detected aromatic hydrocarbons) of the *Centrosaurus* bone may come from sources other than proteins.

Py-GC/MS evidence of kerogen in the form of a homologous series of *n*-alkane/*n*-alkene doublets appears to have been detected in the *Centrosaurus* bone, but the doublets were very weak. Variation in the visibility of the doublets between the matrix-surrounded and uncovered subterranean *Centrosaurus* bone samples is likely representative of intra-bone variation in kerogen content rather than contamination since a strong kerogen signature is not likely to result from exposure to air or the sterilized excavating equipment. Future analyses should examine these samples by mass spectrometry under selected ion monitoring (SIM) scanning mode with comparison to authentic standards of *n*-alkane/*n*-alkenes or modify extraction methods prior to analysis in order to more clearly observe these potential doublets. Kerogen forming from in situ polymerization of endogenous labile lipids such as cell membranes might not be expected to preserve the tubular or hollow shape of 'soft tissues' such as vessels or cells in bone with high fidelity since initial hydrolytic cleavage from a hydrophilic group will eliminate the amphiphatic nature of these molecules and make them incapable of retaining their bilayer configuration in aqueous solution (*Rand and Parsegian, 1989*). Thus, cell membranes might lose their structure, and presumably, the tubular or hollow structure of vessels or cells might be influenced as well. The possibility that the resulting kerogen could contribute to a non-tubular, low-resolution organic mold of such 'soft tissues' formed in the cavities of the bone's inorganic matrix should be considered in cases in which bone demineralization products are not mineralized. However, EDS revealed prevalent mineralization of the structures studied here, a common observation for such 'soft tissue' remains (*Schweitzer et al., 2014*). This mineralization is more consistent with a biofilm origin (*Schultze-Lam et al., 1996*; *Decho, 2010*) rather than a kerogen origin. Furthermore, kerogen-like aliphatic pyrolysis products have previously been detected using Py-GC-MS from the humic fraction of soil as well as the humic acid standard, potentially derived from stable

plant biopolymers from the cuticle (*Saiz-Jimenez and De Leeuw, 1987*), so kerogen-like material in the fossil bone could be derived, at least partly, from soil contaminants rather than being derived from endogenous lipids.

## Amino acids

The dominance of L-amino acids in the dinosaur fossils suggests significant leaching and degradation of endogenous amino acids, as well as relatively recent amino acid input. There appears to be a trend toward greater concentration of amino acids in the dinosaur bone compared to the mudstone, suggesting that the fossil bone might be preferentially concentrated in exogenous amino acids. Furthermore, Cretaceous bone samples, as in the uncovered subterranean *Centrosaurus* bone and the surface-eroded Late Cretaceous bone on the same ridge and ~21 m above BB180, show evidence of bacterial contamination. Their greater Glx D/L values than Asx D/L values are the reverse of that expected due to chemical racemization (*Smith and Evans, 1980*; *Crisp et al., 2013*), but in association with racemized Ala in these samples, they support the presence of peptidoglycans from bacterial cell walls, which contain D-amino acids, particularly D-Glu and D-Ala, in vivo (*Höltje, 1998*; *Lam et al., 2009*). This is consistent with the observed dominance of gram-positive Actinobacteria in the Cretaceous bone microbiome, since 26–75 wt% of the total cell dry weight of gram-positive bacteria comprises cell wall polymers and 7–56 wt% of the cellular amino acids comprises peptides from the cell wall peptidoglycan and teichoic acid and from S-layer glycoproteins (*Onstott et al., 2014*).

The differences in THAA compositions between surface-eroded and matrix-surrounded subterranean Late Cretaceous bone might suggest that subterranean bone provides a different microenvironment than surface-eroded bone, perhaps largely driven by differences in oxygen availability, and thereby containing a different microbial community. This is further evidenced by the close match between topsoil and surface-eroded Late Cretaceous bone THAA composition, while the matrix-surrounded subterranean bone plots more closely to, although still very distinct from, mudstone than to topsoil (*Figure 6*). This might suggest that surface-eroded bone supports a microbial community more similar to other surface communities, such as topsoil, while the subterranean bone contains a more different community.

Variability in THAA concentration in the surface-eroded Late Cretaceous bones is not surprising given that one of the surface-eroded fragments appeared to have relatively higher mineral infiltration (evidenced by greater difficulty in powdering the sample), suggesting the potential for different microenvironments inside the bone and different carrying capacities for a microbial community. One surface-eroded sample came from an active paleontological quarry (BB180) and was likely exposed to higher levels of contamination as a result. The high THAA concentrations observed in one of the surface-eroded Late Cretaceous bones compared to the subterranean bone further suggests that bone can be colonized by exogenous microbes, since surface exposure would be expected to result in adverse conditions for any surviving endogenous proteins. However, such comparisons should be done cautiously given the small sample size of this study.

The most surprising result might be that the uncovered subterranean *Centrosaurus* bone sample had a THAA composition more closely matching surface-eroded bone than the matrix-surrounded subterranean bone, and also had an elevated THAA concentration compared to the matrix-surrounded subterranean bone, suggesting that even relatively brief aerial exposure might lead to rapid contamination of the subterranean microbial community by surface microbes. The high THAA concentration in the adjacent mudstone matrix of the subterranean bone compared to the other mudstone samples may indicate that bone provides a nutrient source that encourages microbial proliferation.

## Radiocarbon

As the C in the dinosaur bone is not radiocarbon dead, this suggests an influx of more modern C (i.e. not radiocarbon dead) into the fossil. However, lower $F^{14}C$ in the dinosaur bone compared to the mudstone or topsoil suggests some biologically inaccessible, old, and possibly endogenous C within the fossils. One possibility for this pattern is kerogen derived from in situ polymerization of endogenous dinosaur labile lipids, although this type of aliphatic geopolymer has only been weakly detected in the *Centrosaurus* bones through Py-GC-MS (potentially due to methodology rather than low concentration), and it should be kept in mind that the surrounding mudstone matrix yields a

series of $n$-alkanes/$n$-alkenes after pyrolysis. Exogenous C could also become metabolically inaccessible in bone through biofilm mineralization, as suggested by the EDS data, allowing for $^{14}$C depletion. Additionally, biofilm formation and proliferation in bones could trap mobile organic C from sediment and groundwater at a rate faster than C exits the bone when not colonized by a biofilm. This would allow for a lower $F^{14}$C steady state to be reached during the time it takes C outflux to increase in order to match C influx, assuming a simple 1-box model. Perhaps a combination of these three mechanisms influences $F^{14}$C.

### Nucleic acids

Analyses of nucleic acids reveal a diverse, unusual microbial community within the dinosaur bone, even when compared to the immediate mudstone matrix or the exterior surface of the bone, as evidenced by a strong enrichment in DNA and differing community composition in the bone relative to the surrounding matrix. The microbial community from the EDTA demineralized bone was similar to that of the non-demineralized bone, important since EDTA can be used as the demineralizing agent to study the 'soft tissues' of fossil bone (*Cleland et al., 2012*). Thus, bone samples treated with common methods of demineralization in other taphonomic studies (e.g. antibody-based studies) are also amenable for nucleic acid analyses that can be used to help test the endogeneity of organics (i.e. whether there are microbes present that could possibly explain the presence of specific organics).

PI staining of soft tissues is very likely due to cell rupture from exposure to the high concentrations of EDTA used during demineralization (i.e. non-intact cells). The dominance of the aerobic *Euzebya* is consistent with the shallow depth of burial, although the presence of the Deltaproteobacteria lineages may indicate that the microenvironment inside the fossil bone creates anaerobic niches to support anaerobic metabolism. Further work is required to understand the relationship of the observed mineral phases and the microbiome. The fact that Actinobacteria were the most common microbes in the dinosaur bone based on 16S rRNA amplicon sequencing is reminiscent of the results from a 38 ka Neanderthal bone, where the majority of detected DNA sequences derived from non-ancient Actinobacteria (*Zaremba-Niedźwiedzka and Andersson, 2013*). The high cell concentrations of ~5×10$^8$ cells/g in the subterranean *Centrosaurus* bone and the consistency in the DNA and amino-acid-based estimates indicates a microbial community that is more substantial than that of the adjacent mudstone.

## Lack of evidence for ancient, endogenous proteins in dinosaur bones

### Structure and elemental composition

HCl demineralization products of dinosaur fossil bone differ structurally and elementally from the Pleistocene-Holocene and modern samples when examined using light microscopy and VPSEM. Low-pressure conditions of VPSEM and EDS, as well as charging during these analyses, may have affected subsequent light microscopy observation, but this is mitigated by the fact that light microscopy was done under a comparative framework between the samples.

The Pleistocene-Holocene shark tooth and modern chicken bone demineralize to reveal large organic masses (i.e. rich in C and O) consistent with collagen protein as evidenced by discernable N and S peaks, unlike the much older dinosaur bone demineralization products. The relatively more pronounced S peak in the shark tooth as compared to the chicken bone might indicate sulfurization of the collagen protein or some other taphonomic incorporation of inorganic S from the environment into the tooth, the latter being consistent with pyrite. After all, the teeth are the only fossils in this study to derive from a marine depositional environment, so the potential for pyrite formation under euxinic conditions, for example, would not be surprising. The high Fe content in the shark tooth suggests some taphonomic mineral accumulation (e.g. iron oxide or pyrite) and may explain some of the dark discoloration in the teeth, potentially alongside a browning effect caused by the taphonomic formation of melanoidin-like N-heterocyclic polymers known as advanced glycoxidation/lipoxidation end products. Raman spectroscopy has not only been used to suggest that these N-heterocyclic polymers are present in ancient teeth, bone, and eggshell, but also that they lead to brown staining (*Wiemann et al., 2018b*). However, it should be kept in mind that, given the open system behavior of bone, detected polymers could derive from exogenous sources of polypeptides and lipids/polysaccharides (e.g. either ancient or more recent infiltrating microbes), and the presence of any amide bands in Raman spectra is likely insufficient evidence for endogenous

oligopeptide preservation (see a similar discussion of amide bands in FTIR below) especially in association with polymers that form as a result of protein degradation (*Singh et al., 2001*; *Vistoli et al., 2013*).

## IR active bonds

Similar, albeit higher resolution, FTIR peaks to those detected here are used as evidence for purported dinosaur collagen (*Lindgren et al., 2011*; *Surmik et al., 2016*; *Lee et al., 2017*), but, it should be noted that such results are not conclusive of collagen. Detection of peaks such as those associated with amide bonds may not necessarily indicate intact proteins/peptides, since amide bonds are not specific to peptides and can be found in protein degradation products such as diketopiperazines (*Chiavari and Galletti, 1992*; *Martins and Carvalho, 2007*; *Saitta et al., 2017b*). CH and CO bonds are even more widely distributed, found in a variety of organic molecules. Some researchers have indeed attempted to observe how ATR FTIR spectra of bone collagen is modified when carbonaceous contamination (e.g. applied organics like consolidants, humic acids, or soil carbonate) is present (*D'Elia et al., 2007*), but it can be tempting for taphonomists to observe organic peaks in such IR spectra and attribute them to endogenous protein (*Lindgren et al., 2011*; *Surmik et al., 2016*; *Lee et al., 2017*). Even if such bonds are from proteins, without deconvolution of peaks to produce fingerprints of protein secondary structure (*Byler and Susi, 1986*), one cannot say from the presence of such organic bonds alone that the protein is collagen, let alone endogenous or ancient. Such deconvolution could be performed on the data collected here in the future.

Despite the strong HCl demineralization treatment, it appears that some phosphate remained in the samples. It has been shown experimentally and theoretically that variation in the phosphate bands derived from ATR FTIR of bone can be affected by bone collagen content, with low-frequency symmetry of the phosphate peaks more apparent in bone containing lower amounts of collagen (*Aufort et al., 2018*). The observation of sharper, more symmetric phosphate peaks in the *Centrosaurus* bone compared to the younger bone might suggest lower relative collagen content. However, it should be noted that the described pattern in phosphate peak alteration was observed using a diamond ATR, and this method can result in differences in spectra from those made using Ge ATR, as was done here, due to different refractive indices of the crystals (*Aufort et al., 2016*), so such a comparison may be inappropriate. Additionally, it would be advisable to obtain ATR FTIR data from non-demineralized samples before trying to interpret the results here, since it is unclear how HCl demineralization might affect this correlation between phosphate peak symmetry and collagen content. Regardless, it might be worth discussing symmetry in the phosphate peaks on any future papers that attempt to use ATR FTIR data as evidence for purported Mesozoic collagen. Future work on the specimens analyzed here should also attempt ATR FTIR mapping on polished sections to examine how peaks are spatially distributed, perhaps in combination with time-of-flight secondary ion mass spectrometry (TOF-SIMS).

## Pyrolysis products

The dinosaur fossil bones show greater chemical resemblance in their total ion chromatograms to mudstone than to fresh, modern bone and appear somewhat low in organics relative to fresh, modern bone. Although compounds such as benzenes are protein pyrolysis products, the detected prominent pyrolysis products in the *Centrosaurus* bone are relatively simple and are not as indicative of high proteinaceous content as would be amides (as in the chicken bone studied here), succinimides, or piperazines (*Saitta et al., 2017b*), or even less protein-specific pyrolysis products such as the prominent nitriles detected in the chicken bone sample. Regardless, the presence of protein-related pyrolysis products does not indicate that these proteins are necessarily ancient, endogenous, or collagenous.

## Amino acids

Amino acids in the dinosaur bone are dominated by proteins other than collagen and appear to be relatively recent. Low amino acid concentrations, low D-amino acid concentrations, and THAA compositional profiles that do not match collagen, despite high Gly content, suggest that the majority of the endogenous collagen protein has been lost from the dinosaur fossils. Changes in the THAA compositional profile as a result of taphonomic alteration and preferential loss of less stable amino acids

would be expected in samples of this age, with any remaining endogenous protein likely to have low levels of sequence and higher order structural preservation, with a consequent impact on the preservation of epitopes for antibodies.

In contrast to the Late Cretaceous bone, the much younger shark teeth from the Pleistocene-Holocene have relatively high amino acid concentrations whose THAA compositional profiles are consistent with a dominance of collagen. Since ethanol rinsing did not change the THAA compositional profile of shark teeth, this suggests that the majority of the organics are deriving from insoluble collagen with fairly well preserved higher order structure, rather than highly fragmented peptides with greater mobility. This observation is consistent with the results from light microscopy and VPSEM, which revealed fibrous masses. The shark teeth also have relatively high racemization, a testament to the antiquity of the amino acids as would be expected from endogenous proteins.

## Conclusions

Previous studies have often reported purported endogenous 'soft tissues' within fossil dinosaur bone (*Pawlicki et al., 1966*; *Schweitzer et al., 2005a*; *Schweitzer et al., 2005b*; *Schweitzer et al., 2007a*; *Schweitzer et al., 2007b*; *Schweitzer et al., 2008*; *Schweitzer et al., 2009*; *Schweitzer et al., 2013*; *Schweitzer et al., 2014*; *Schweitzer et al., 2016*; *Asara et al., 2007*; *Organ et al., 2008*; *Schweitzer, 2011*; *Bertazzo et al., 2015*; *Cleland et al., 2015*; *Schroeter et al., 2017*). However, these studies often do not fully address fossil bones being open systems that are biologically active. This can be seen in field observations, in Dinosaur Provincial Park and elsewhere, where fossil bone is frequently colonized by lichen on the surface or overgrown and penetrated by plant roots in the subsurface. This forces researchers to consider that subsurface biota (e.g. plant roots, fungi, animals, protists, and bacteria) could contaminate bone. Given that fungi can produce collagen (*Celerin et al., 1996*), the need to rule out exogenous sources of organics in fossil bone is made all the greater. Even deeply buried bone has the potential to be biologically active, given the high concentration of microorganisms in continental subsurface sedimentary rock (*Magnabosco et al., 2018*). The analyses presented here are consistent with the idea that far from being biologically 'dead', fossil bone supports a diverse, active, and specialized microbial community. Given this, it is necessary to rule out the hypothesis of subsurface contamination before concluding that fossils preserve geochemically unstable endogenous organics, like proteins.

We detected no evidence of endogenous proteins in the bone studied here and were therefore unable to replicate claims of protein survival from deep time, such as the Mesozoic (*Pawlicki et al., 1966*; *Schweitzer et al., 2005a*; *Schweitzer et al., 2007a*; *Schweitzer et al., 2007b*; *Schweitzer et al., 2008*; *Schweitzer et al., 2009*; *Schweitzer et al., 2013*; *Schweitzer et al., 2014*; *Schweitzer et al., 2016*; *Asara et al., 2007*; *Organ et al., 2008*; *Schweitzer, 2011*; *Bertazzo et al., 2015*; *Cleland et al., 2015*; *Schroeter et al., 2017*). In contrast, recent Pleistocene-Holocene material often shows clear, and multiple lines of, evidence for endogenous, ancient collagen. These may be found even when the fossil (dentine/enamel in this case) is stained black, indicating taphonomic alteration, and the sample is found exhumed in a warm climate and not treated with aseptic techniques. Detection of specific organic signatures in fossils (e.g. amide bands in FTIR or Raman spectroscopy) requires corroborating evidence before claims of ancient proteins can be substantiated. In addition to reliable markers of general protein presence (e.g. amide, succinimide, or piperazine pyrolysis products), evidence is required to identify the type of protein (i.e. amino acid composition or sequence) as well demonstrate its endogenous origin (e.g. localization) and age (i.e. degree of degradation as revealed by amino acid racemization, post-translational modifications such as deamidation, or peptide length/degree of hydrolysis). Degradation of collagen polypeptides follows a pattern of gradual hydrolysis of amino acids at the terminal ends followed by catastrophic degradation and rapid hydrolysis due to rupture of the triple helix quaternary structure, making the resulting gelatinous fragments susceptible to rapid leaching from the bone or microbial degradation (*Collins et al., 1995*; *Collins et al., 2009*; *Dobberstein et al., 2009*). It might therefore be suspected that if ancient collagen does indeed persist in a fossil bone, then such preservation would often provide clear, strong structural and chemical signatures like that in the Pleistocene-Holocene shark teeth. Recently it has been suggested that techniques that do not provide information on the precise sequence or post-translational modification of peptides, such as Py-GC-MS or HPLC amino acid analysis, are outdated for palaeoproteomic studies (*Cleland and Schroeter, 2018*). This might be the case when samples are very young and from cold environments, in which case, more precise

mass spectrometric analyses such as liquid chromatography-tandem mass spectrometry might be employed early on in the course of research with elevated confidence that ancient proteins are capable of persisting in the sample. However, our results here suggest that techniques like Py-GC-MS or HPLC that give more general information on protein presence versus absence or general amino acid composition should be considered frontline approaches when dealing with samples of significant age and/or thermal maturity (e.g. *Demarchi et al., 2016*; *Hendy et al., 2018*; *Cappellini et al., 2018*). Treating Mesozoic bone that has experienced diagenesis, low latitudes, and permineralisation identically to more recent, less altered bone is ill-advised, and any work on such samples should employ these fundamental methods before attempting to sequence peptides that might not be present, ancient, or endogenous.

Fossil bone has fairly high concentrations of recent organics (e.g. L-amino acids, DNA, and non-radiocarbon dead organic C), even when buried and often in comparison to the immediate environment. Fossil bone likely provides an ideal, nutrient-rich (e.g. phosphate, iron) open system microbial habitat inside vascular canals capable of moisture retention. The absence of evidence for endogenous proteins and the presence of a diverse, microbial community urge caution regarding claims of dinosaur bone 'soft tissues'. Microbes can colonize bones while buried, likely traveling via groundwater. Therefore, it is unsurprising that the prevalence of these 'soft tissues' is not correlated with overburden depth above the fossil or cortical versus cancellous bone tissue (*Ullmann et al., 2019*). Rather, minimum distance from the surface is probably of importance and microbes likely readily colonize a variety of bone tissue types since both presumably behave as open systems. Our results support the hypothesis that at least some 'soft tissue' structures derived from demineralised fossil bones represent biofilms. We suggest that unless in an inaccessible form (e.g. kerogen, depending on microbial metabolic ability) or matrix (e.g. well-cemented concretion), endogenous dinosaur organics that survive prior taphonomic processes (e.g. diagenesis) may be subject to subsequent microbial metabolic recycling.

The study of fossil organics must consider potential microbial presence throughout a specimen's taphonomic history, from early to late. Microbial communities interact with fossils immediately following death and after burial, but prior to diagenesis. Microbes are known to utilize bone and tooth proteins (*Child et al., 1993*) and fossil evidence of early fungal colonization has even been detected (*Owocki et al., 2016*). More recent microbial colonization of fossil bone will occur as it nears the surface during uplift and erosion in the late stages of the taphonomic process. Furthermore, given that microbes can inhabit the crust kilometres below the surface (*Magnabosco et al., 2018*), it might be predicted that bone remains a biologically active habitat even when buried hundreds of meters deep for millions of years. The extensive potential for microbial contamination and metabolic consumption makes verifying claims of Mesozoic bone protein extremely challenging.

## Acknowledgements

Part of this work was financially supported by the Scott Vertebrate Paleontology Fund from the Department of Geosciences, Princeton University. Many thanks to Wei Wang and Jessica Wiggins (Princeton University, Genomics Core Facility) for 16S rRNA gene amplicon sequencing, Paul Monaghan (University of Bristol) for assistance in preparing samples for radiocarbon AMS, Sheila Taylor (University of York) for assistance in preparing the samples for HPLC, Kirsty High (University of York) for provision of the comparator sheep bone sample, Kentaro Chiba (University of Toronto) for assistance in the quarry of BB180 and for taking photographs, the Royal Tyrrell Museum of Palaeontology for accessioning the fossils collected in Dinosaur Provincial Park and assisting in the paperwork allowing for export and study, and Adam Maloof (Princeton University), Jasmina Wiemann (Yale University), and Michael Buckley (University of Manchester) for helpful discussion. The Natural Environment Research Council, UK, provided partial funding of the mass spectrometry facilities at Bristol (Contract No. R8/H10/63). Amino acid analyses were undertaken thanks to support to KP from the Leverhulme Trust (PLP-2012–116). *Gallus gallus domesticus* (Public Domain Dedication 1.0, https://creativecommons.org/publicdomain/zero/1.0/legalcode), Odontapsidae (Dmitry Bogdanov, vectorised by T Michael Keesey, Creative Commons Attribution 3.0 Unported, https://creativecommons.org/licenses/by/3.0/legalcode, CC BY 3.0), and *Centrosaurus apertus* (credit: Andrew A Farke, Creative Commons Attribution 3.0 Unported, https://creativecommons.org/licenses/by/3.0/legalcode, CC BY 3.0, modified in *Figure 5*) silhouettes were obtained from phylopic.org.

## Additional information

### Funding

| Funder | Grant reference number | Author |
| --- | --- | --- |
| Princeton University | Scott Vertebrate Paleontology Fund | Renxing Liang<br>Maggie CY Lau<br>Tullis Onstott |
| Leverhulme Trust | PLP-2012-116 | Marc R Dickinson<br>Kirsty E H Penkman |

The funders had no role in study design, data collection and interpretation, or the decision to submit the work for publication.

### Author contributions

Evan T Saitta, Conceptualization, Formal analysis, Investigation, Methodology, Writing—original draft, Writing—review and editing; Renxing Liang, Maggie CY Lau, Marc R Dickinson, Conceptualization, Formal analysis, Investigation, Methodology, Writing—review and editing; Caleb M Brown, Conceptualization, Resources, Data curation, Investigation, Methodology, Writing—review and editing; Nicholas R Longrich, Thomas G Kaye, Ben J Novak, Steven L Salzberg, Conceptualization, Writing—review and editing; Mark A Norell, Conceptualization, Resources, Data curation; Geoffrey D Abbott, Conceptualization, Resources, Software, Formal analysis, Investigation, Methodology, Writing—review and editing; Jakob Vinther, Conceptualization, Supervision, Investigation, Methodology, Project administration, Writing—review and editing; Ian D Bull, Resources, Formal analysis, Investigation, Methodology, Writing—review and editing; Richard A Brooker, Peter Martin, Timothy DJ Knowles, Kirsty EH Penkman, Tullis Onstott, Conceptualization, Resources, Formal analysis, Investigation, Methodology, Writing—review and editing; Paul Donohoe, Resources, Software, Formal analysis, Investigation, Methodology, Writing—review and editing

### Author ORCIDs

Evan T Saitta [iD] https://orcid.org/0000-0002-9306-9060
Maggie CY Lau [iD] https://orcid.org/0000-0003-2812-9749
Caleb M Brown [iD] https://orcid.org/0000-0001-6463-8677
Steven L Salzberg [iD] https://orcid.org/0000-0002-8859-7432
Jakob Vinther [iD] http://orcid.org/0000-0002-3584-9616
Richard A Brooker [iD] http://orcid.org/0000-0003-4931-9912
Kirsty EH Penkman [iD] https://orcid.org/0000-0002-6226-9799

### Decision letter and Author response

Decision letter https://doi.org/10.7554/eLife.46205.sa1
Author response https://doi.org/10.7554/eLife.46205.sa2

## Additional files

### Supplementary files

- Source data 1. Raw data for ATR FTIR, EDS, Py-GC-MS, and 16S rRNA amplicon sequencing.
- Transparent reporting form

### Data availability

All data generated or analysed during this study are included in the manuscript and supporting files. Sequencing data has been placed in the NCBI Sequence Read Archive under the accession number SRR7947417.Raw data are available through the Field Museum's collections database: multimedia record containing raw files (GUID: 60c79cec-4da1-4bea-8535-a332c70ae4c9, URI: https://mm.field-museum.org/60c79cec-4da1-4bea-8535-a332c70ae4c9), event record with surrounding information

about the project (GUID: 34e15532-2c46-47cf-aac0-5d29cc5a2c22, URI: https://pj.fieldmuseum.org/event/34e15532-2c46-47cf-aac0-5d29cc5a2c22).

The following datasets were generated:

| Author(s) | Year | Dataset title | Dataset URL | Database and Identifier |
|---|---|---|---|---|
| Saitta ET, Liang R, Lau MCY, Brown CM, Longrich NR, Kaye TG, Novak BJ, Salzberg SL, Norell MA, Abbott GD, Dickinson MR, Vinther J, Bull ID, Brooker RA, Martin P, Donohoe P, Knowles TDJ, Penkman KEH, Onstott T | 2019 | Cretaceous dinosaur bone contains recent organic material and provides an environment conducive to microbial communities | http://www.ncbi.nlm.nih.gov/sra?term=SRR7947417 | NCBI Sequence Read Archive, SRR7947417 |
| Saitta ET | 2019 | Dataset for Taphonomic research on organic material in Cretaceous dinosaur bones | https://mm.fieldmuseum.org/60c79cec-4da1-4bea-8535-a332c70ae4c9 | Field Museum collections, 60c79cec-4da1-4bea-8535-a332c70ae4c9 |

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

## Appendix 1

### Further introduction

Dinosaur bone organic structures, if indeed Mesozoic 'soft tissue', would be expected to consist primarily of extracellular structural proteins and phospholipids of cell membranes, which are unstable through diagenesis and deep time (**Bada, 1998**; **Briggs and Summons, 2014**). While protein sequences are lost through hydrolysis of peptide bonds, phospholipids hydrolyse at their ester bonds, freeing fatty acids from glycerol-phosphate polar heads (**Eglinton and Logan, 1991**; **Zuidam and Crommelin, 1995**). The resulting free fatty acids, however, can polymerise in situ to form kerogen-like aliphatic hydrocarbons which are stable through diagenesis (**Stankiewicz et al., 2000**; **Gupta et al., 2006a**; **Gupta et al., 2006b**; **Gupta et al., 2007a**; **Gupta et al., 2007b**; **Gupta et al., 2008**; **Gupta et al., 2009**). The abiotic conversion of the hydrocarbon tails of phospholipids into kerogen is far more likely than preserving a protein sequence of amino acids. For example, one survey of amino acids in fossil bone (**Armstrong et al., 1983**) yielded data in which fossils older than the Upper Pleistocene no longer had amino acid compositions similar to fresh bone proteins. Additionally, amino acid concentration decreased from recent bone to Mesozoic bone, while racemisation increased sharply from recent bone to Upper Pleistocene bone, but then gradually decreased from that point through to Mesozoic bone, suggesting protein loss and contamination (see below for a reanalysis of this data from **Armstrong et al., 1983**). This pattern of protein loss and contamination is supported by more recent work (**Dobberstein et al., 2009**; **High et al., 2015**; **High et al., 2016**). However, partially intact Pliocene peptides about 3.4 Ma have been verified from exceptionally cold environments (**Rybczynski et al., 2013**) and under what, for now at least, seems like unique molecular preservational mechanisms in the calcite crystals of eggshell from 3.8 Ma (**Demarchi et al., 2016**). Both of these examples, however, are far younger than Mesozoic fossils.

Even simple estimations do not predict protein survival in the deep geologic record. Assuming fairly average human body composition (**Janaway et al., 2009**) and mass, it only takes ~5% of the water already present in the body to hydrolyse all of the peptide bonds in the proteome (see 'Hydrolysis estimates' section below). This calculation assumes a closed system with no additional water, and it seems unlikely that any fossil matrix would be anhydrous throughout its entire taphonomic history. For example, it requires a significant amount of diagenetic alteration to fossilise resin into desiccated copal and amber (**Langenheim, 1969**; **Langenheim, 1990**; **Lambert and Frye, 1982**; **Mills et al., 1984**; **Pike, 1993**; **Child et al., 1993**; **Ragazzi et al., 2003**; **Villanueva-García, 2005**). Calculating exponential decay curves assuming first order kinetics paints an even more pessimistic picture. Modifications at terminal regions and internal peptide bonds shielded by steric effects can result in longer observed half lives of peptide bonds under hydrolysis (**Kahne and Still, 1988**; **Radzicka and Wolfenden, 1996**; **Testa and Mayer, 2003**). At 25°C and neutral pH, peptide bond half lives as a result of uncatalysed hydrolysis for the relatively stable acetylglycylglycine (C-terminal), acetylglycylglycine *N*-methylamide (internal), and the dipeptide glycylglycine are 500, 600, and 350 years, respectively (**Radzicka and Wolfenden, 1996**). Even assuming a very conservative half life of 600 years for all peptide bonds in the average human body at arguably unextreme conditions (25°C and neutral pH), no bonds would remain after ~51,487 years, keeping in mind that hydrolysis rates depend on the surrounding peptide/protein environment such that observed peptides can greatly exceed this estimate (also see 'Hydrolysis estimates' section below):

$$b = h/(2^\wedge(y/t_{1/2}))$$

where $b$ = number of peptide bonds = 1,

$y$ = years,

$$h = \text{h} = \text{estimated number of peptide bonds in average human body} = 6.78844 \times 10^{25},$$
$$\text{and } t_{1/2} = \text{half life in years under uncatalysed hydrolysis} = 600.$$

The half life in this case is 3 orders of magnitude too short in order to get peptide bonds surviving into the Mesozoic (~66 Ma); a half life of ~769,130 years would be required. This does not even take into consideration environmental or diagenetic increases in temperature or pH fluctuations, nor does it take into account scavenging or microbial/autolytic decay. Of course, these values are based on some very extreme assumptions and should not be taken as precise estimates, but rather, as framing the enormity of the challenge for Mesozoic protein survival. Empirically derived estimates for collagen and osteocalcin upper age limits based on experimentally observed gelatinisation and Gla-rich mid-region epitope loss, respectively, can give widely different estimates at 20°C: 15,000 years for collagen and 580,000 years for osteocalcin. The estimates vary according to temperature. For example, at 0°C, the upper age limit for collagen and osteocalcin are estimated at 2,700,000 and 110,000,000 years, respectively (*Nielsen-Marsh, 2002*). Even frozen collagen by these estimates fails to survive long enough for the possibility of survival in Mesozoic specimens, and no Mesozoic fossils have been preserved frozen since they predate the appearance of the current polar ice caps.

The kinetics of thermal instability under non-enzymatic reactions are just one hurdle that such 'soft tissues' would have to clear. Bone is also an open system (*Bada et al., 1999*), allowing for organic and microbial influx. Invasion of microbes into the bone could lead to the enzymatic degradation of endogenous organics (in addition to any autolytic degradation from endogenous enzymes) and mobile breakdown products of organics can be lost from the bone into the surroundings.

## Detailed Methods

### ATR FTIR

ATR FTIR was carried out at the University of Bristol in order to detect any bonds present in the samples that might derive from proteins. Samples were powdered in a sterile mortar and pestle (70% ethanol rinsed) and then demineralised in 10 mL of 0.5 M hydrochloric acid (HCl) for 5 days, with the acid replaced three times during that period by spinning in a centrifuge and pipetting off the old acid and replacing with fresh acid. After demineralisation and pipetting out the last acid volume, the samples were rinsed with 10 mL of milli-Q water and spun in a centrifuge three times, replacing the water each time. After pipetting out the last water volume, samples were freeze dried overnight.

The demineralization products were analyzed using a Nicolet iN10 MX FTIR spectrometer with a KBr beamsplitter and MCT/A detector. Thin flakes of sample were placed on a transparent KBr 'zero background' plate and specific areas of interest identified in transmitted light. A microATR attachment was then inserted in the beampath and a background spectra collected before the Ge tip (repeatedly cleaned with ethanol) was forced into the sample. 128 scans were then collected over a wavelength range from 675 to 4000 cm$^{-1}$ at 8 cm$^{-1}$ resolution and converted to an absorbance spectrum. The aperture windows were set to 50 μm giving an effective collection area of about 17 μm at the sample.

### Light microscopy, VPSEM, and EDS

The same demineralised samples that underwent ATR FTIR were subsequently analysed by VPSEM and EDS performed at the University of Bristol in order to characterize the ultrastructural texture and elemental makeup of any 'soft tissue' structures resulting from demineralization. Specimens were mounted onto carbon tape on standard SEM pin-stubs and were not electrically coated. A Zeiss SIGMA-HD VPSEM instrument was used in this work, with the instrument's chamber filled with a recirculated nitrogen supply to negate against the electrical surface charge accumulation on the sample. Typical vacuum levels during analysis varied between 0.1 and 0.25 mbar. Control of the SEM, with a specified spatial resolution of

1.2 nm under such low-vacuum conditions, was performed using the microscope's standard SmartSEM user interface. For standard sample imaging, a beam current of 1.7 nA (30 μm aperture), 15 kV accelerating voltage, and a 10 mm working height (horizontal sample; no tilt) were used. During the accompanying EDS compositional analysis of regions of interest within the sample, both the beam current and accelerating voltage were increased to 2.9 nA and 20 kV, respectively, with the sample position in the instrument remaining unchanged. An EDAX Ltd. (Amatek) Octane Plus Si-drift detector was used for the EDS analysis, with control performed through the accompanying TEAM analytical software. Collection periods of 100 s were used, with the electrically (Peltier) cooled detector operating with a dead-time of 20% to permit for individual peak discrimination from the 30,000–40,000 counts per second incident onto the device. Elemental quantification of the spectra obtained was performed using the eZAF deconvolution and peak-fitting algorithm based upon the ratios of the differing K, L, and M X-ray emissions.

After VPSEM and EDS analysis, the same samples were imaged using light microscopy utilising a Leica DFC425 C digital camera under magnification from a Leica M205 C stereomicroscope in order to characterize the microscopic structure of any 'soft tissue' structures resulting from demineralization.

## Py-GC-MS

Py-GC-MS was conducted at the School of Chemistry, University of Bristol in order to produce a chemical fingerprint of the samples for comparative purposes as well as to search for potential protein-related pyrolysis products and alkane/alkene signatures of kerogen. Sample fragments were rinsed in 70% ethanol prior to powdering with a sterile mortar and pestle (70% ethanol rinsed) to remove exterior contamination. A quartz tube was loaded with ~1 mg of the sample powder and capped with glass wool. A pyrolysis unit (Chemical Data Systems (CDS) 5200 series pyroprobe) was coupled to a gas chromatograph (GC; Agilent 6890A; Varian CPSil-5CB fused column: 50 m length, 0.32 mm inner diameter, 0.45 μm film thickness, 100% dimethylpolysiloxane stationary phase) and a double focussing dual sector (reverse Niers Johnson geometry) mass spectrometer (ThermoElectron MAT95, ThermoElectron, Bremen; electron ionisation mode, 310°C GC interface, 200°C source temperature) with a 2 mL min$^{-1}$ helium carrier gas. Samples were pyrolysed in quartz tubes (20 s, 610°C), transferred to the GC (310°C pyrolysis transfer line), and injected into the GC (310°C injector port temperature was maintained, 10:1 split ratio). The oven was programmed to heat from 50°C (held for 4 min) to 300°C (held for 15 min) by 4 °C min$^{-1}$. A $m/z$ range of 50–650 was scanned (one scan per second). There was a 7 min delay whereby the filament was switched off for protection against any pressure increases at the start of the run. MAT95InstCtrl (v1.3.2) was used to collect data. QualBrowser (v1.3, ThermoFinnigan, Bremen) was used to view data. Compounds were identified with the aid of the National Institute of Standards and Technology (NIST) database.

## HPLC amino acid analysis

Reversed-phase high performance liquid chromatography (RP-HPLC) for analysing amino acid composition and racemization was done at the University of York on samples originally sent to the University of Bristol. Samples from Dinosaur Provincial Park were transported to York on ice. Replicate sample fragments were either ethanol (70%) rinsed prior to powdering or powdered without a rinse with a sterile mortar and pestle (also 70% ethanol rinsed). Following the methods of *High et al. (2016)*, several mg of powder were accurately weighed into sterile 2 mL glass vials (Wheaton). Then, 7 M HCl (Aristar, analytical grade) was added, and the vials were flushed with $N_2$. Hydrolysis (18 hr, 110°C) was performed, and samples were rehydrated with a solution containing HCl (0.01 mM) and L-*homo*-arginine (LhArg) internal standard. Chiral amino acid pairs were analysed using a RP-HPLC (Agilent 1100 series; HyperSil C18 BDS column: 250 mm length, 5 μm particle size, 3 mm diameter) and fluorescence detector, using a modified method outlined by *Kaufman and Manley (1998)*. Column temperature was controlled at 25°C and a tertiary solvent system containing methanol, acetonitrile, and sodium

acetate buffer (23 mM sodium acetate trihydrate, 1.3 µM ethylenediaminetetraacetic acid (EDTA), 1.5 mM sodium azide, adjusted to pH 6.00 ± 0.01 using 10% acetic acid and sodium hydroxide) was used. Some replicates of the samples underwent an additional preparative method to reduce peak suppression caused by high mineral content (*Dickinson et al., 2019*). This involved salt removal by adding 60 µL of 1M HCl to ~2 mg of powdered sample in a 0.5 mL Eppendorf tube, sonicating for 2 mins to dissolve the powder, adding 80 µL of 1M KOH to produce a gel suspension, centrifuging for 5 mins, separating a clear solution from the gel, drying the clear supernatant by centrifugal evaporation, and finally, rehydration in 20 µL LhArg.

Principal component analysis of amino acid concentration data was run on R using the prcomp() command.

## Radiocarbon AMS

Radiocarbon analyses were performed at the BRAMS facility at the University of Bristol in order to assess the age of the organic carbon within the samples. Fossil bone samples were surface cleaned using an autoclaved razorblade to scrape their exterior surface. All samples were powdered by mortar and pestle cleaned through autoclaving and rinsing with 70% ethanol. Samples were transferred into pre-combusted (450℃, 5 hr) culture tubes and 10 mL of 0.5 M HCl were added to eliminate any carbonates. The HCl solution was replaced as necessary until $CO_2$ effervescence ceased. Samples were rinsed with three washes of 10 mL MilliQ ultrapure water before freeze-drying. Samples were weighed into aluminium capsules to obtain ~1 mg C before being combusted in an Elementar Microcube elemental analyser (also obtaining the % C by mass of the demineralised samples) and graphitised using an IonPlus AGE3 graphitisation system. The resulting graphite samples were pressed into Al cathodes and analysed using a MICADAS accelerator mass spectrometer (Laboratory of Ion Beam Physics, ETH, Zurich). All samples were blank subtracted using a bone sample known to be radiocarbon 'dead', the Yarnton sample from *Cook et al. (2012)*.

## DNA extraction, 16S rRNA gene amplicon sequencing, and epifluorescence microscopy

DNA extraction and quantification (Qubit fluorometer) in order to quantify microbial inhabitation in samples, epifluoresence microscopy (SYTO 9/propidium iodide (PI) dual staining) in order to visualize microbial cells, and 16S rRNA gene amplicon sequencing in order to characterize the microbial community composition in samples were conducted at Princeton University. The bone and adjacent mudstone were processed inside a laminar flow hood after its interior had been illuminated with UV for 30 min. Specifically, the bone fragments were carefully picked out and surfaces of the fossil bones were scraped off with an autoclaved razor blade. The bone, the scrapings, and mudstone were powdered separately, with a sterile mortar and pestle that had been autoclaved and UV-treated. The powder from fossil bone samples were either demineralised in 0.5 M EDTA (pH = 8) or not demineralised. The EDTA demineralised bone was stained with SYTO 9 dye and *propidium iodide* (LIVE/DEAD BacLight Bacterial Viability Kit, Molecular Probes, USA) in the dark for 15 min. These are fluorescent dyes that intercalate between the base pairs of DNA. The stained samples were analysed using a fluorescence microscope (Olympus BX60, Japan). Live cells are stained as green whereas membrane-compromised cells fluoresce red.

Powdered samples (0.25 g) were used to extract DNA from the bone, the scrapings, and mudstone by using Power Viral Environmental RNA/DNA Isolation kit (MOBIO Laboratories, Carlsbad, CA, USA). However, the DNA yield from the mudstone was below detection (detection limit of 0.01 ng/µL). Therefore, a further attempt was made to extract DNA from a large amount of powder (5 g bone and 10 g mudstone) using DNeasy PowerMax Soil Kit (QIAGEN, Germany) according to the manufacturer's instruction. Additionally, the slurry from the EDTA demineralised bone was subjected to DNA extraction using the same large scale kit.

Extracted DNA was then quantified by dsDNA HS Assay kit (Life Technologies, Carlsbad, USA) and the fluorescence was measured using a Qubit fluorometer (Invitrogen, Carlsbad, USA).

To prepare the library for 16S rRNA gene amplicon sequencing, DNA was used as PCR template to amplify the 16S rRNA gene V4 region using bacterial/archaeal primer 515F/806R (*Caporaso et al., 2010*). The PCR reaction condition was as follows: initial denaturation at 94°C for 3 min; 25 or 30 cycles of denaturation at 94°C for 45 s, annealing at 50°C for 1 min, extension at 72°C for 90 s and a final extension at 72°C for 10 min. PCR product (5 μL) was loaded onto a gel to confirm the amplification by running agarose gel electrophoresis. The amplicon products were pooled to make the library and sequenced for a 150 bp paired-end reads on Illumina Hiseq 2500 housed in the Genomics Core Facility at Princeton University. The raw sequences were quality-filtered with a minimum Phred score of 30 and analysed by QIIME (Quantitative Insights Into Microbial Ecology) software package (*Caporaso et al., 2010*). The 16S amplicon sequencing data were deposited in the Sequence Read Archive (SRA) of NCBI under the accession number of SRR7947417.

## Hydrolysis estimates

Assuming no exogenous water (i.e., closed, anhydrous system).

**Appendix 1—table 1.** Average human body composition and hydrolysis calculations. Source is *Janaway et al., 2009*.

| Substance | % of body mass | Average adult body mass (kg) | Mass of substance (kg) | Mass of substance (Da) | Typical amino acid (Da) | Number of amino acids | Maximum number of peptide bonds | Number of bonds surviving |
|---|---|---|---|---|---|---|---|---|
| Protein | 20 | 62 | 12.4 | 7.46728E + 27 | 110 | 6.78844E + 25 | 6.78844E + 25 | −1.25963E + 27 |
| | | | | | H₂O (Da) | Number of H₂O molecules | | % H₂O molecules used up in hydrolysis |
| Water | 64 | 62 | 39.68 | 2.38953E + 28 | 18 | 1.32752E + 27 | | 5.113636364 |

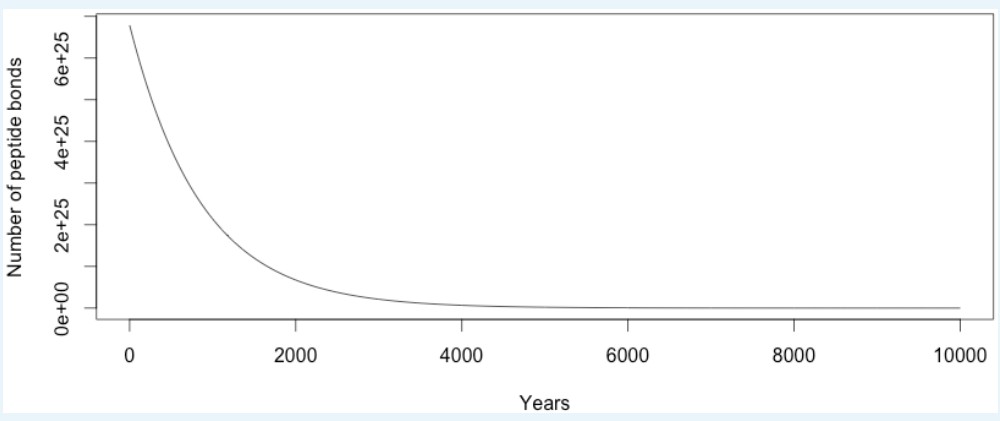

**Appendix 1—figure 1.** Decay curve of average human proteome based on a half life of 600 years as described by the equation in the text.

## Reanalysis of *Armstrong et al. (1983)*

Reanalysis of amino acid composition from the appendices of *Armstrong et al. (1983)* and data digitised from the figures therein was done using the prcomp() function in R (scale set to 'TRUE' to normalise data since some of the samples do not sum to 1000 despite being recorded as parts per thousand [‰]). PCA shows that only the Holocene and Upper

Pleistocene samples cluster near the modern bone samples (*Appendix 1—figure 2*). The questionable Jurassic sauropod lies away from the modern protein samples while its sediment is nearer to them, suggesting input of exogenous, modern protein (e.g., collagen from fungi or lab contamination). Some of the samples are keratin rather than bone so this is a 'general protein' analysis. The one solid black circle that lies away from the rest is keratin from a turtle whose data was digitised from a figure, rather than taken directly from a table so this point is not very comparable to the bone samples. The rest show high variation and are often closer to bacteria. Most of the variation (PC1) tends to separate recent–Upper Pleistocene samples from the older samples. Most of the contamination (bacteria or sediment) tend to lie nearer the more ancient samples with respect to PC1. The PC loadings biplot (*Appendix 1—figure 3*) shows the changes in the amino acids responsible for the different positions on the PC space (e.g., a loss of PRO, VAL, HYP, GLY, GLU, and ALA and increase in PHE, HIS, LEU, TYR, ILE, and 'unknown' from modern protein to ancient samples and contamination). Variation, along PC2, in the other amino acids is apparent in the contaminated or diagenetically altered samples but not so much in the modern samples. The Upper Devonian fossil samples all plot closely to the Upper Devonian sediment, suggesting that the amino acids are present throughout and represent contamination. One question is whether there are two bacteria samples reported in *Armstrong et al. (1983)*. Both figures were digitised and the two points lie close together (open black circles). Regardless, it might be proof of consistency in data digitisation if this sample is indeed doubled, and their similarity would be expected to only very minimally alter the PCA. The table in *Armstrong et al. (1983)* lists 'hand contamination' that is presented here as a bacterial sample (open black circle that lies farther from the other two bacteria points).

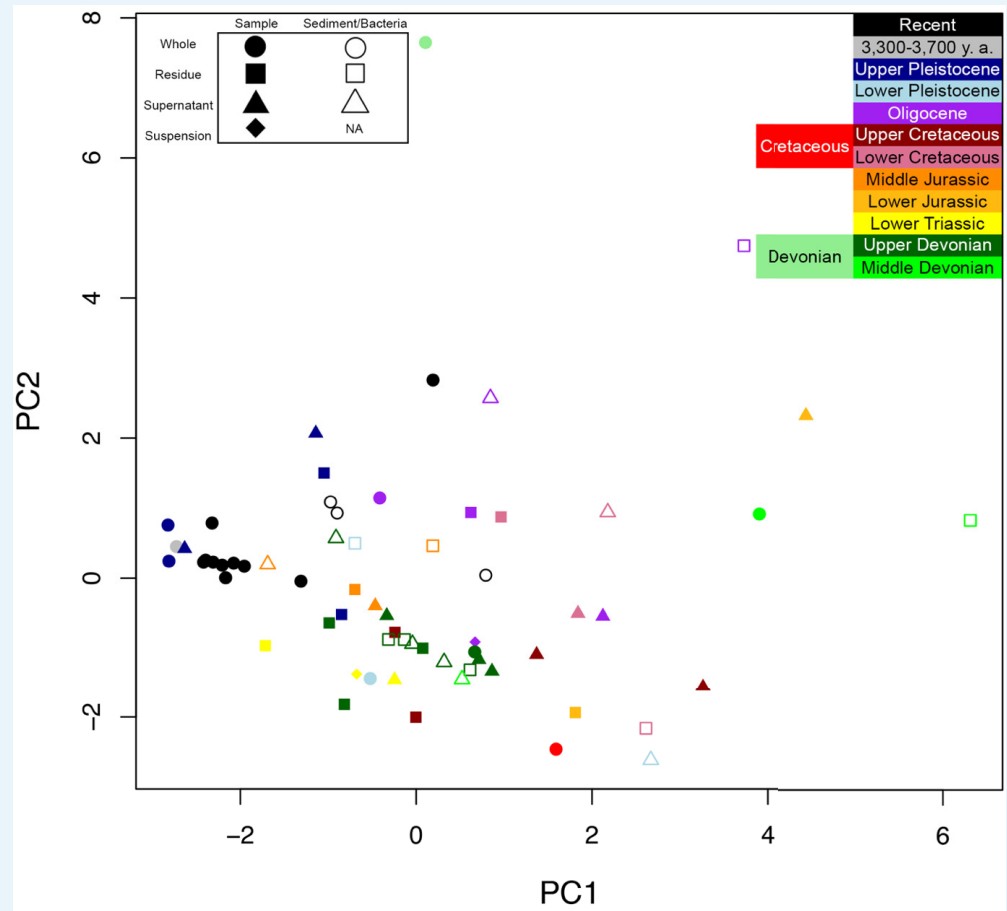

**Appendix 1—figure 2.** PCA of data from *Armstrong et al. (1983)*.

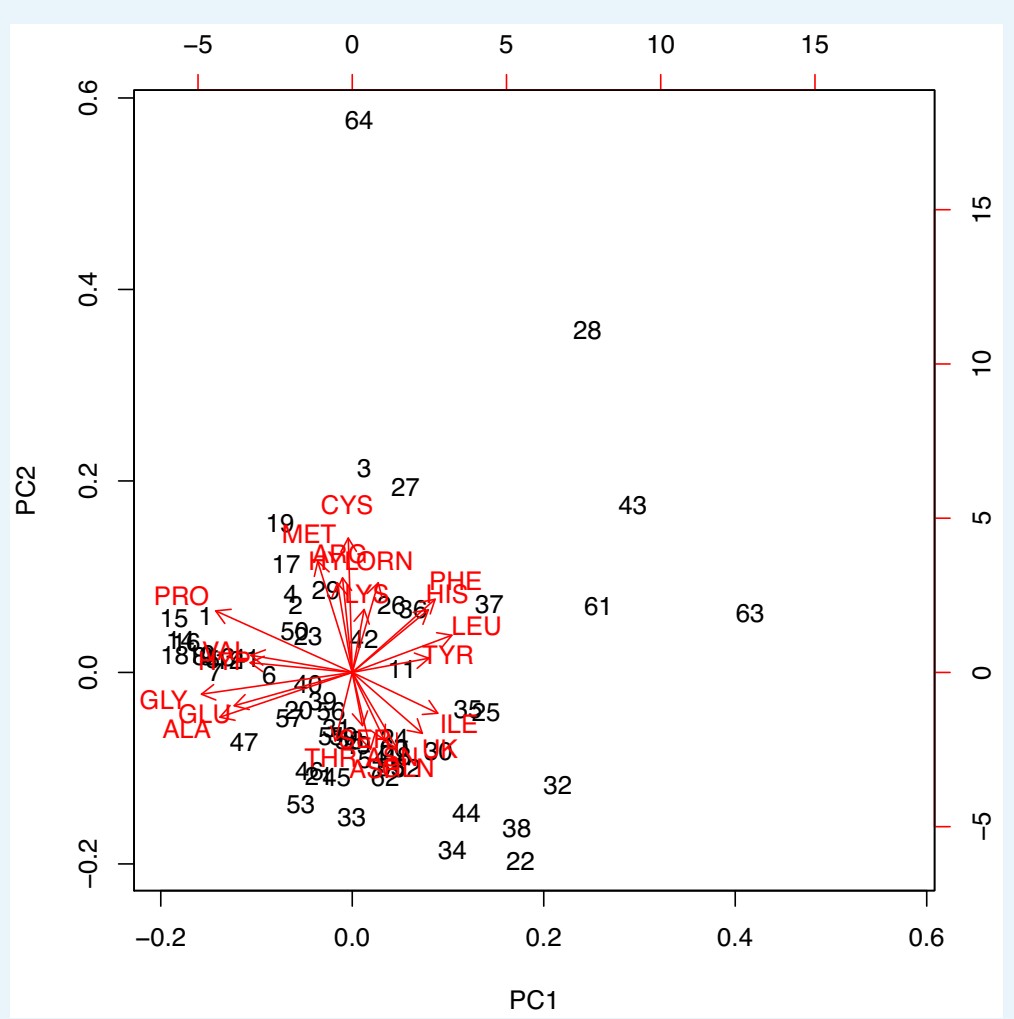

**Appendix 1—figure 3.** PCA biplot of data from *Armstrong et al. (1983)*.

PCA was also run on only the samples themselves, excluding the bacteria and sediment (*Appendix 1—figure 4*). This is essentially assuming that contamination does not occur and examining the amino acids in the samples as if they must be endogenous. Even with this major caveat, one still sees the same pattern as before. Only the Holocene and Upper Pleistocene samples cluster near the recent samples. Most of the variation in the samples' amino acid concentrations (PC1) is a result of differences between recent–Upper Pleistocene samples versus all of the other samples. Amino acid changes associated with PC1 can be seen in the biplot (*Appendix 1—figure 5*). Recent and Upper Pleistocene samples have higher PRO, VAL, ARG, HYP, GLY, GLU, and ALA while the older samples have higher PHE, TYR, HIS, LEU, SER, ASP, and 'unknown'.

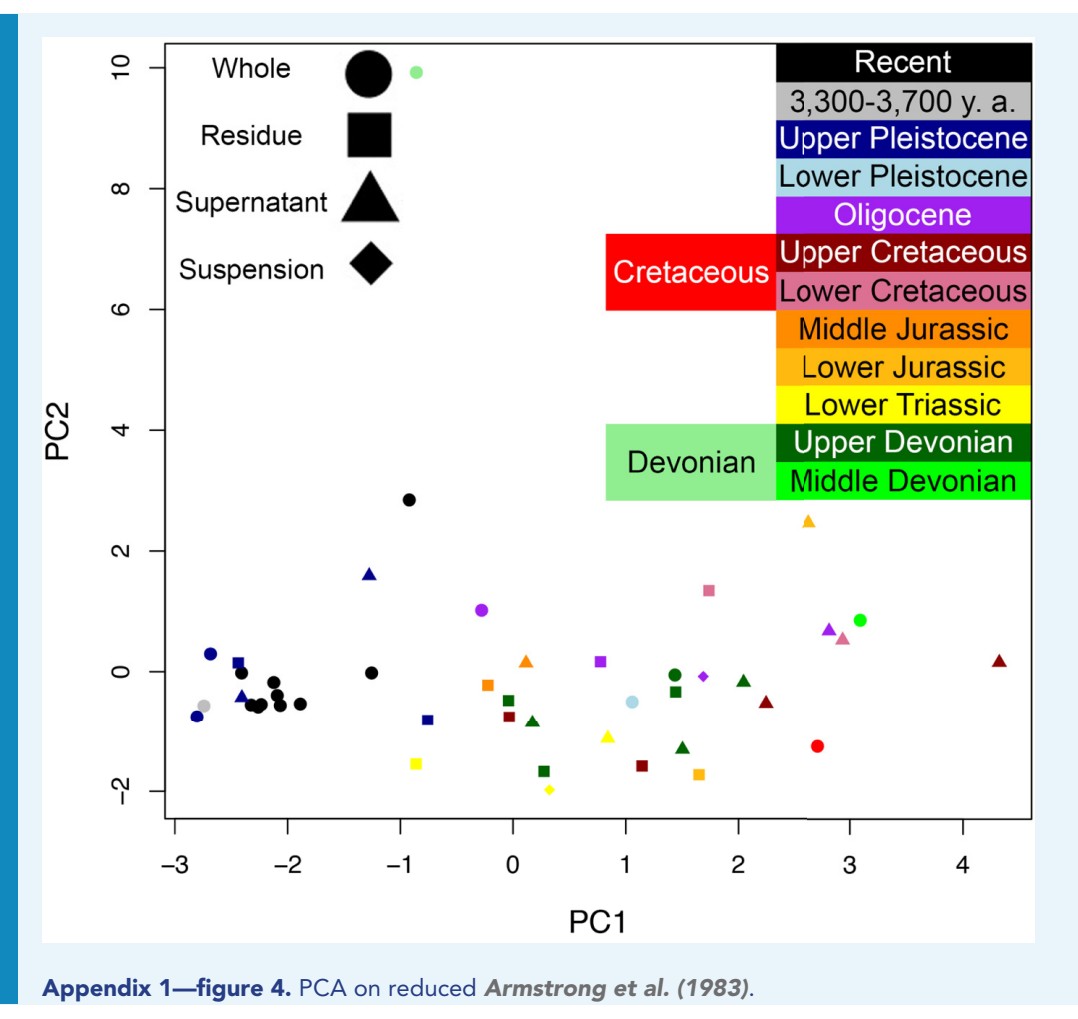

**Appendix 1—figure 4.** PCA on reduced *Armstrong et al. (1983)*.

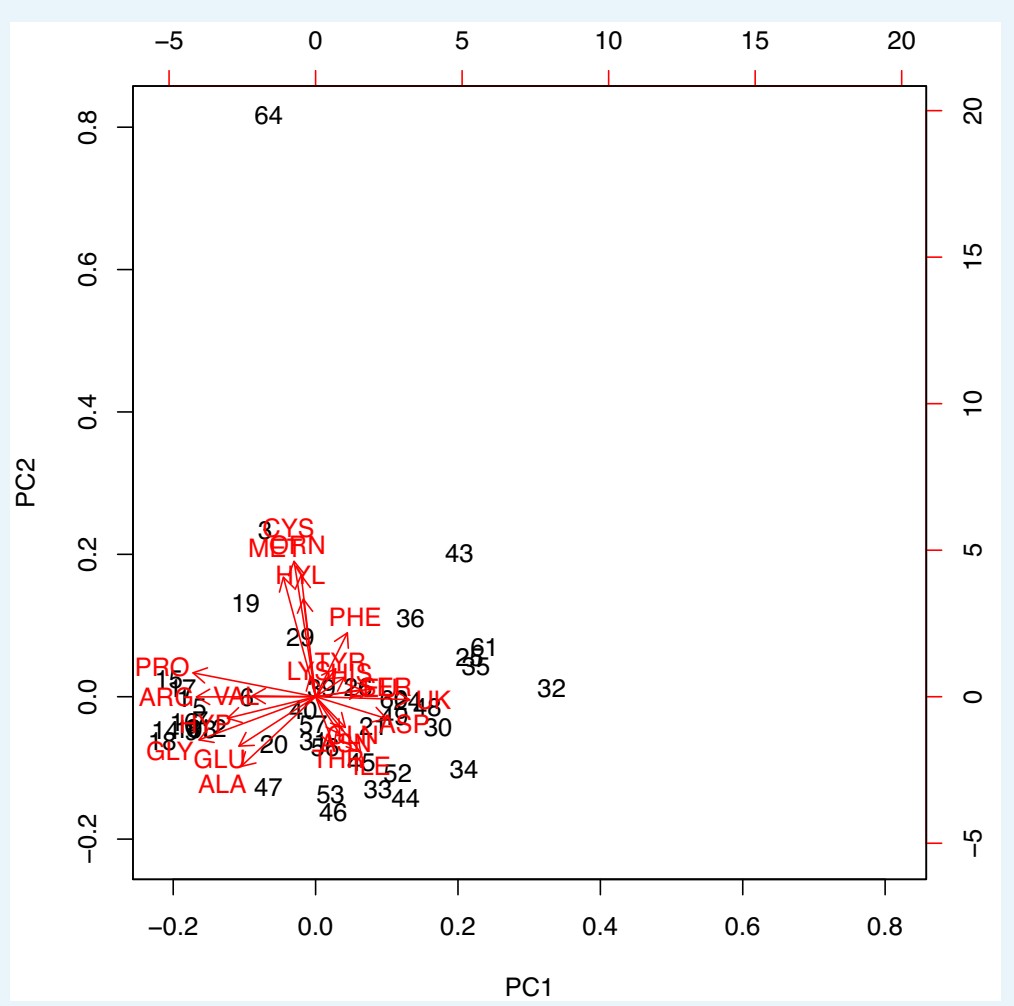

**Appendix 1—figure 5.** PCA biplot on reduced *Armstrong et al. (1983)*.

Given the presentation of the data in the appendix of *Armstrong et al. (1983)*, it appears that the recent samples did not show racemisation (marked by Armstrong et al. with a '-" representing 'absence in analysis', rather than 'n.d.' representing 'not determined'). Regardless, we expect modern proteins to be very low, if not zero, in D/L value. More ancient samples show a rapid rise in Ile epimerisation going from recent to Upper Pleistocene samples, followed by a gradual decrease towards lower A/I, reaching a minimum in the Lower Jurassic (*Appendix 1—figure 6*). Epimerisation then increases in the even older samples older. Given that samples older than the Upper Pleistocene do not have amino acid compositions similar to modern protein (as shown above), this suggests that Mesozoic samples tend to show more recent contamination, while the Palaeozoic samples tend to show ancient contamination, in agreement with the fact that Upper Devonian fossils and sediment have similar amino acid compositions. The high epimerisation in the 'hand contamination' sample (the open black circle) is peculiar and maybe exposure on the hands kills bacteria and results in conditions favorable for epimerisation (e.g., washing or exposure to chemicals).

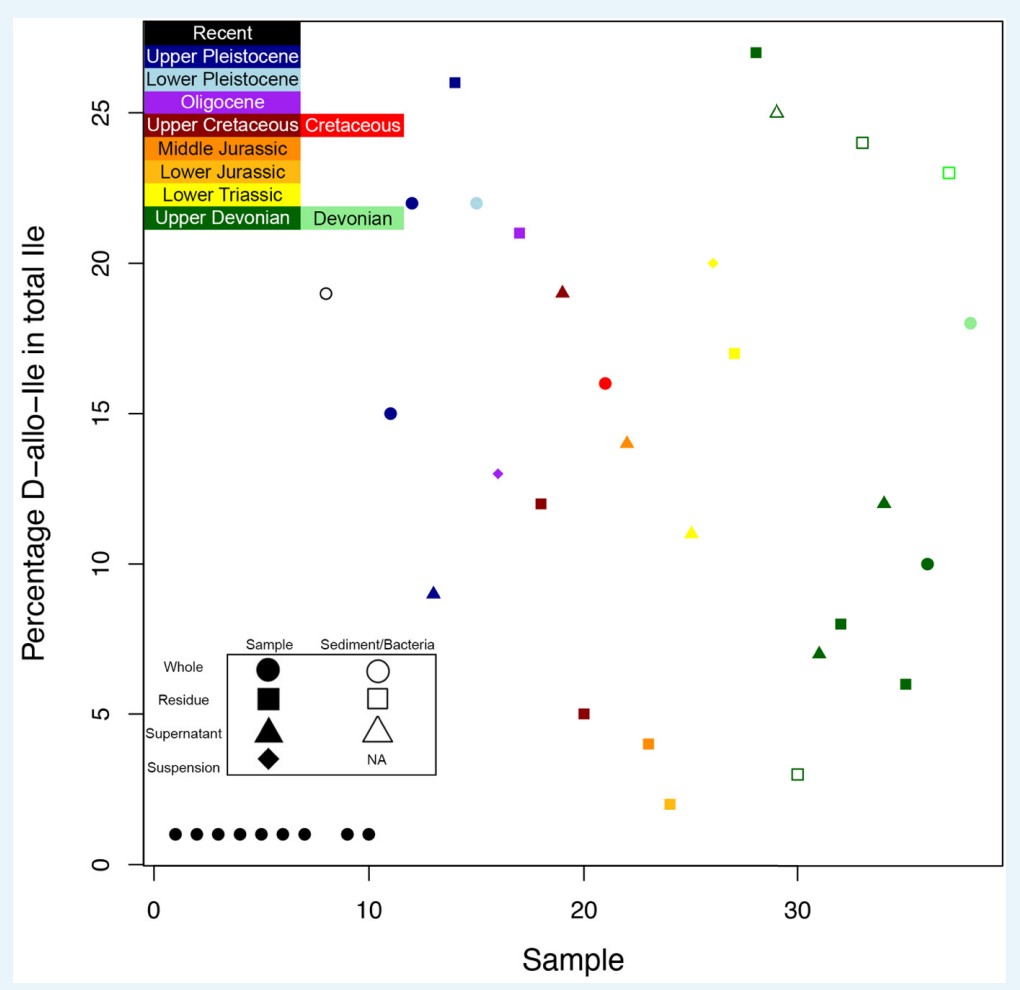

**Appendix 1—figure 6.** Amino acid epimerisation data in *Armstrong et al. (1983)*.

A similar pattern occurs in the amino acid mass as a percentage of total sample mass (*Appendix 1—figure 7*). Percent amino acid mass decreases from younger to older samples with a minimum in the Lower Triassic samples. The Devonian samples show high percent amino acid mass, suggesting that this ancient contamination might provide more amino acid mass than does recent contamination.

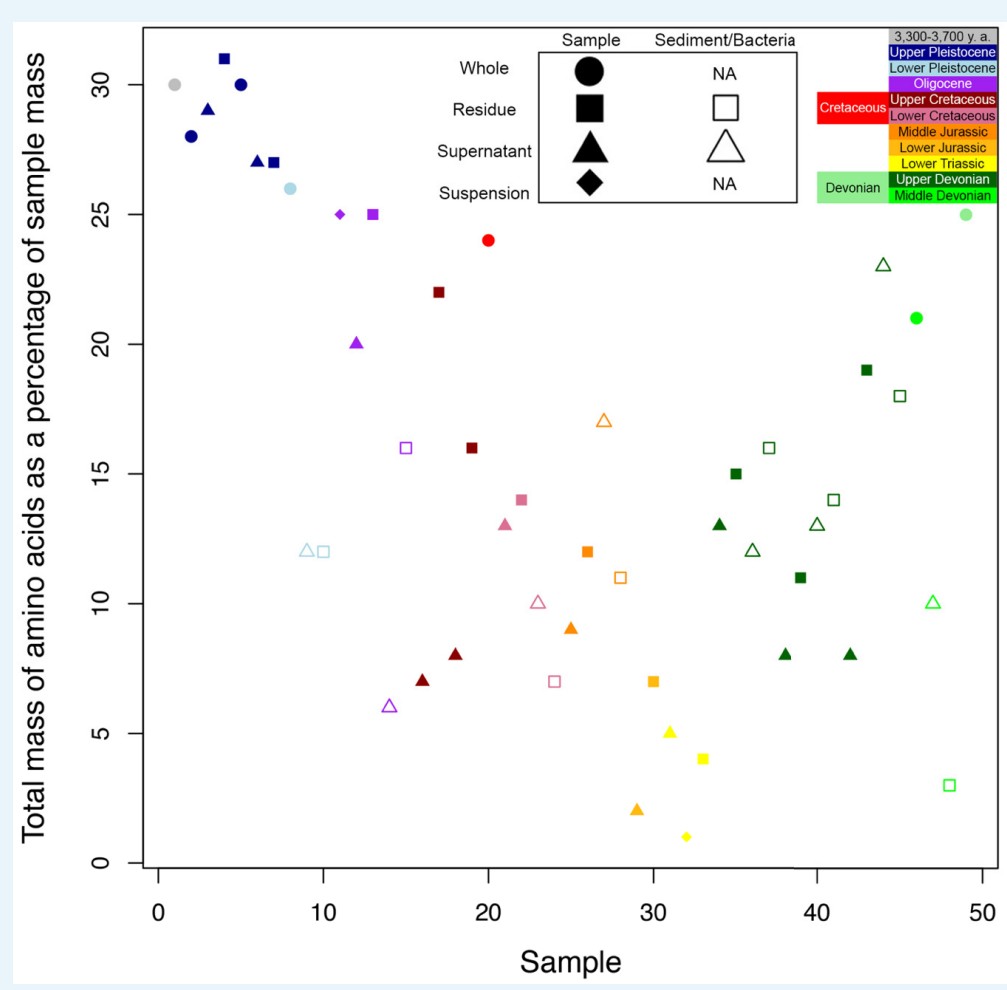

**Appendix 1—figure 7.** Amino acid concentration data in *Armstrong et al. (1983)*.

## Information on aseptically collected Dinosaur Provincial Park samples and modern chicken control

### Photographs of sample collection

**Appendix 1—table 2. Sample ID key and descriptions.**

| Sample bag # | Originally sent to | HPLC ID | Py-GC/MS ID | DNA extraction, fluorescence microscopy, 16S rRNA amplicon ID | Location | Type | Details |
|---|---|---|---|---|---|---|---|
| 1 (TMP 2016.016.0007) | Princeton | NA | NA | 1B, 1S, 1F | Dinosaur Provincial Park, Alberta, Canada | Bone | *Centrosaurus* rib aseptically collected within immediately surrounding sediment. One end of the rib was first exposed. Aseptic protocol was then implemented to expose more of the rib. A rib section was isolated with the surrounding sediment kept in situ. Foil was placed on top of this section. The bone was sawed on its ends and then flipped by prying underneath with an awl. Mudstone tended to fracture during flipping. More foil was added after flipping to encase the whole sample. |
| 1 (TMP 2016.016.0007) | Princeton | NA | NA | 1M | Dinosaur Provincial Park, Alberta, Canada | Mudstone matrix | Mudstone matrix surrounding and collected with the matrix-surrounded *Centrosaurus* rib sample. Had tendency to fracture when manipulated. |
| 2 (TMP 2016.016.0007) | Princeton | NA | NA | 2B, 2S | Dinosaur Provincial Park, Alberta, Canada | Bone | Uncovered *Centrosaurus* rib section immediately adjacent to matrix-surrounded section of sample bag #1. |
| 6 (TMP 2016.016.0013) | Princeton | NA | NA | 6B, 6S | Dinosaur Provincial Park, Alberta, Canada | Bone | Surface bone eroded out of BB180, either excavated in past years and left or naturally eroded. About eight steps away from quarry cliff-face. 667 m elevation. |
| 8 (TMP 2016.016.0014) | Princeton | NA | NA | 8M | Dinosaur Provincial Park, Alberta, Canada | Mudstone sediment | Sediment from BB180 at bone producing layer. Sampled ∼30 cm away from the sampled rib and tibia (sample bags #1–4). 670 m elevation. |
| 10 (TMP 2016.016.0015) | Princeton | NA | NA | 10, 10T | Dinosaur Provincial Park, Alberta, Canada | Topsoil | Topsoil from same ridge as BB180. 734 m elevation. |
| 11 (TMP 2016.016.0016) | Princeton | NA | NA | 11M | Dinosaur Provincial Park, Alberta, Canada | Mudstone sediment | Sediment on same ridge as BB180 from 709 m elevation. Outcrop was dug into by several cm before sampling. |
| 13 (TMP 2016.016.0017) | Princeton | NA | NA | 13, 13M | Dinosaur Provincial Park, Alberta, Canada | Mudstone sediment | Sediment on same ridge as BB180 from 693 m elevation. Outcrop was dug into by several cm before sampling. |
| 16 (TMP 2016.016.0018) | Princeton | NA | NA | 16B, 16S | Dinosaur Provincial Park, Alberta, Canada | Bone | Surface bone eroded out of same ridge as BB180 but at 691 m elevation. Unknown taxon. Near sample bags #13–14. |
| NA | Bristol | 1 | 1 | NA | Sainsbury's Bristol, UK | Bone | Chicken bone purchased from grocery store with meat removed. |

*Appendix 1—table 2 continued on next page*

Appendix 1—table 2 continued

| Sample bag # | Originally sent to | HPLC ID | Py-GC/MS ID | DNA extraction, fluorescence microscopy, 16S rRNA amplicon ID | Location | Type | Details |
|---|---|---|---|---|---|---|---|
| 3 (TMP 2016.016.0008) | Bristol | 2 | 2 | NA | Dinosaur Provincial Park, Alberta, Canada | Bone | *Centrosaurus* tibia aseptically collected within immediately surrounding sediment. One end of the tibia was first exposed. Aseptic protocol was then implemented to expose more of the tibia. A tibia section was isolated with the surrounding sediment kept in situ. Foil was placed on top of this section. The bone was sawed on its ends and then flipped by prying underneath with an awl. Mudstone tended to fracture during flipping. More foil was added after flipping to encase the whole sample. |
| 3 (TMP 2016.016.0008) | Bristol | 3 | 3 | NA | Dinosaur Provincial Park, Alberta, Canada | Mudstone matrix | Mudstone matrix surrounding and collected with the matrix-surrounded *Centrosaurus* tibia. Had tendency to fracture when manipulated. |
| 4 (TMP 2016.016.0008) | Bristol | 4 | 4 | NA | Dinosaur Provincial Park, Alberta, Canada | Bone | Uncovered *Centrosaurus* tibia region immediately adjacent to matrix-surrounded section of sample bag #3. |
| 5 (TMP 2016.016.0013) | Bristol | 6 | 6 | NA | Dinosaur Provincial Park, Alberta, Canada | Bone | Surface bone eroded out of BB180, either excavated in past years and left or naturally eroded. About eight steps away from quarry cliff-face. 667 m elevation. |
| 7 (TMP 2016.016.0014) | Bristol | 5 | 5 | NA | Dinosaur Provincial Park, Alberta, Canada | Mudstone sediment | Sediment from BB180 at bone producing layer. Sampled ~30 cm away from the sampled rib and tibia (sample bags #1–4). 670 m elevation. |
| 9 (TMP 2016.016.0015) | Bristol | 7 | 7 | NA | Dinosaur Provincial Park, Alberta, Canada | Topsoil | Topsoil from same ridge as BB180. 734 m elevation. |
| 12 (TMP 2016.016.0016) | Bristol | 8 | 8 | NA | Dinosaur Provincial Park, Alberta, Canada | Mudstone sediment | Sediment on same ridge as BB180 from 709 m elevation. Outcrop was dug into by several cm before sampling. |
| 14 (TMP 2016.016.0017) | Bristol | 9 | 9 | NA | Dinosaur Provincial Park, Alberta, Canada | Sediment | Sediment on same ridge as BB180 from 693 m elevation. Outcrop was dug into by several cm before sampling. |
| 15 (TMP 2016.016.0018) | Bristol | 10 | 10 | NA | Dinosaur Provincial Park, Alberta, Canada | Bone | Surface bone eroded out of same ridge as BB180 but at 691 m elevation. Unknown taxon. Near sample bags #13–14. |

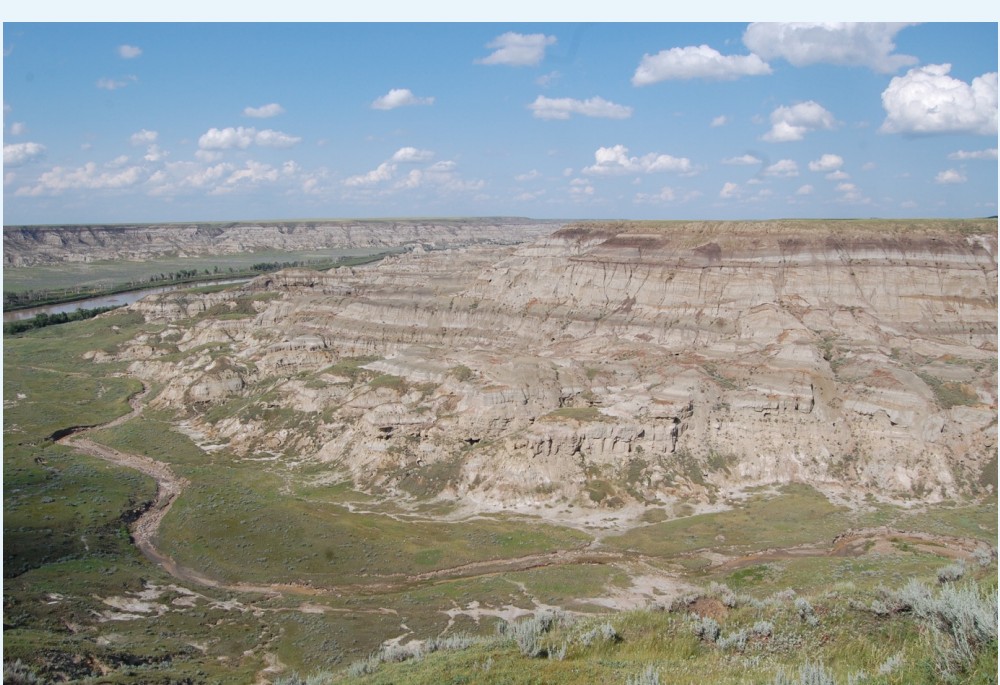

**Appendix 1—figure 8.** Ridge on which BB180 is located. View looking east at mouth of Jackson Coulee, Dinosaur Provincial Park, AB.

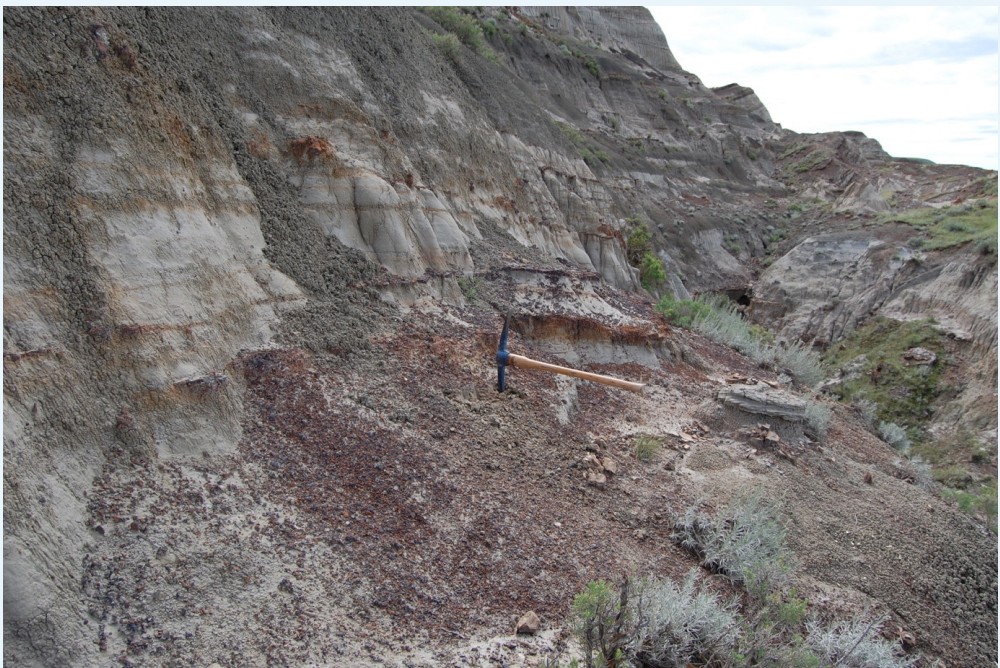

**Appendix 1—figure 9.** Region of BB180 sampled prior to removal of overburden.

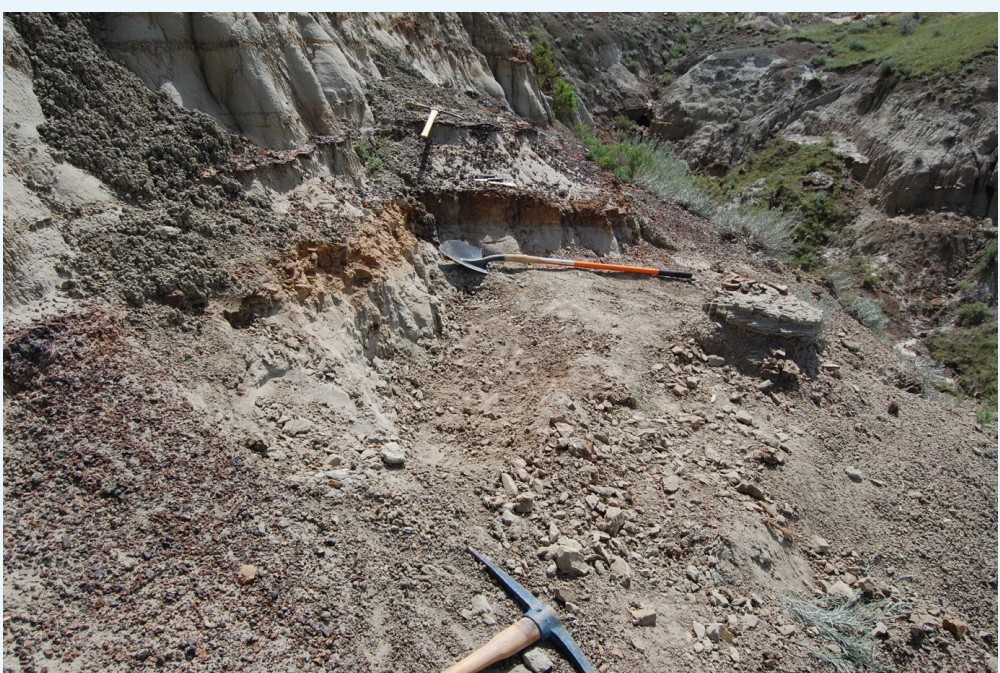

**Appendix 1—figure 10.** Region of BB180 sampled after removal of overburden.

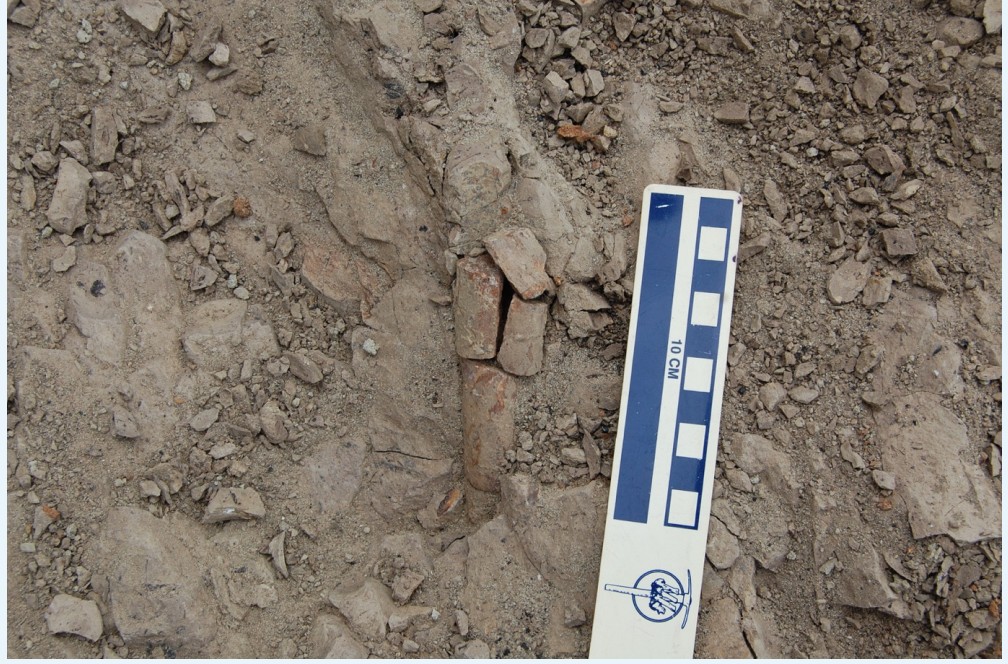

**Appendix 1—figure 11.** Exposed end of Centrosaurus rib upon initial discovery.

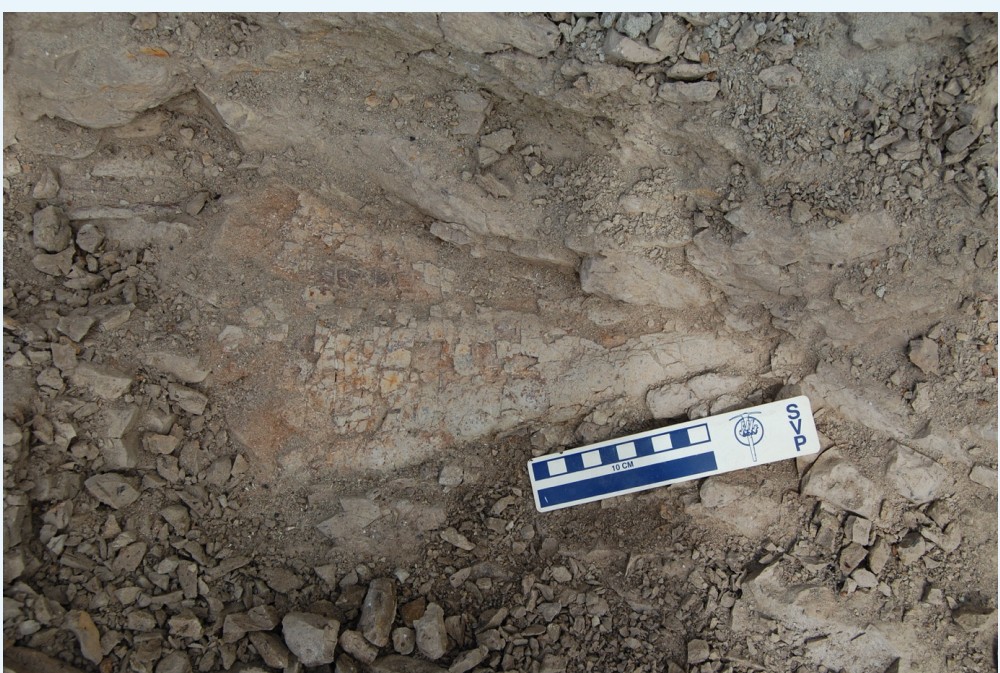

**Appendix 1—figure 12.** Exposed end of Centrosaurus tibia upon initial discovery.

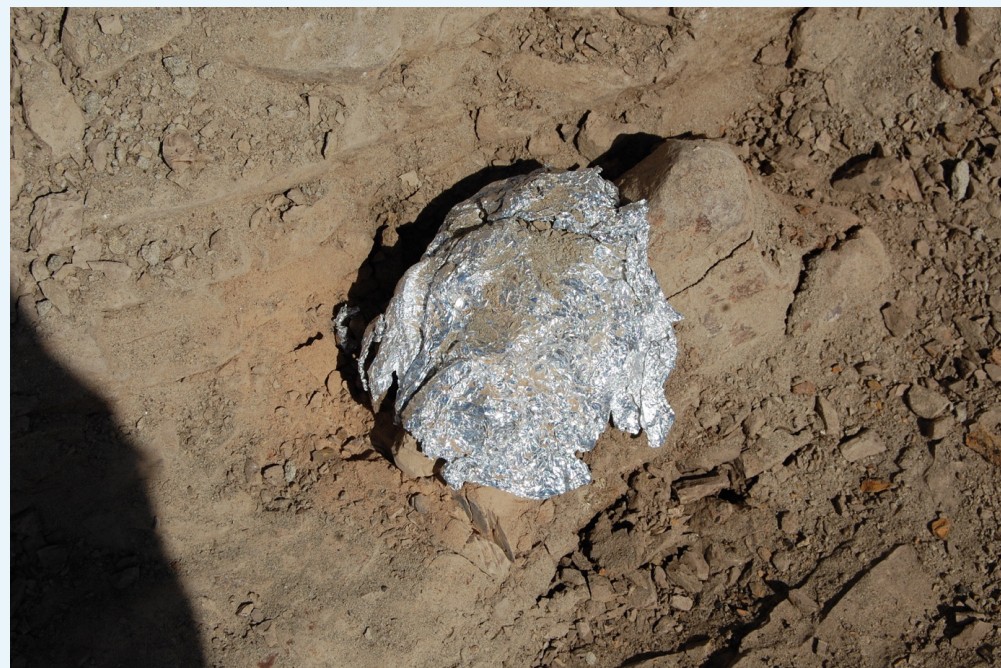

**Appendix 1—figure 13.** Foil placed on top of sediment and matrix-surrounded Centrosaurus tibia portion prior to flipping with an awl. Uncovered distal end of tibia is visible to the right of foil.

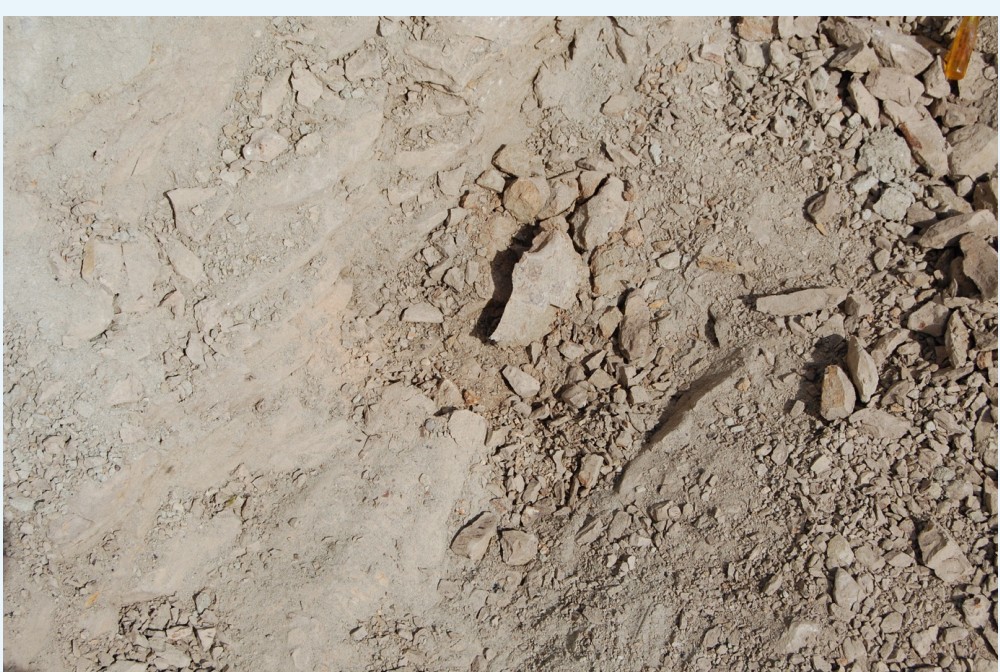

**Appendix 1—figure 14.** Centrosaurus tibia after matrix-surrounded sample and uncovered distal end were collected.

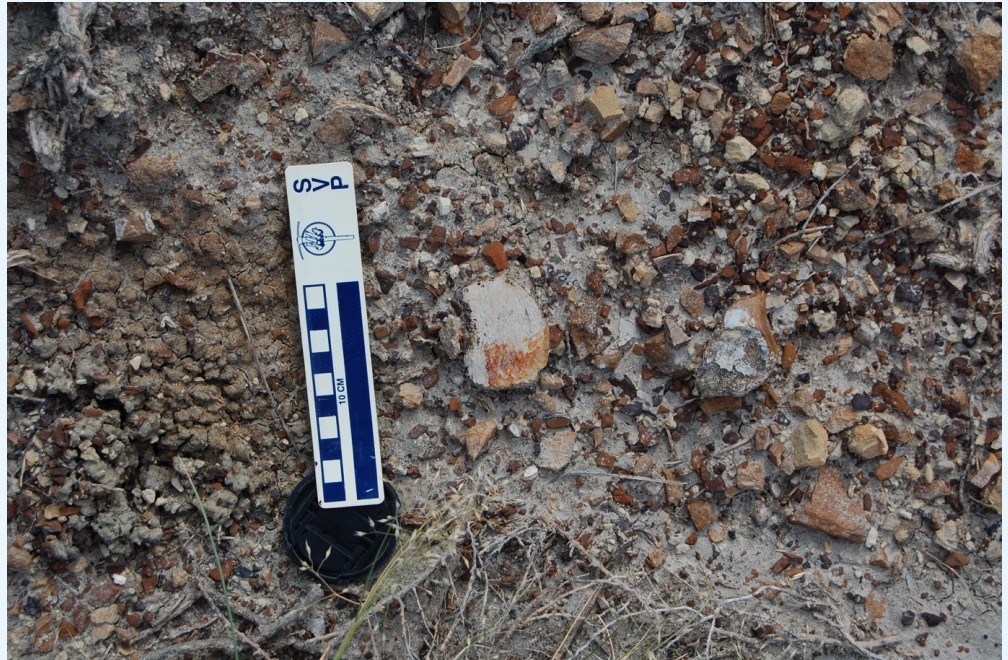

**Appendix 1—figure 15.** Surface eroded bone fragments from BB180 as they were found.

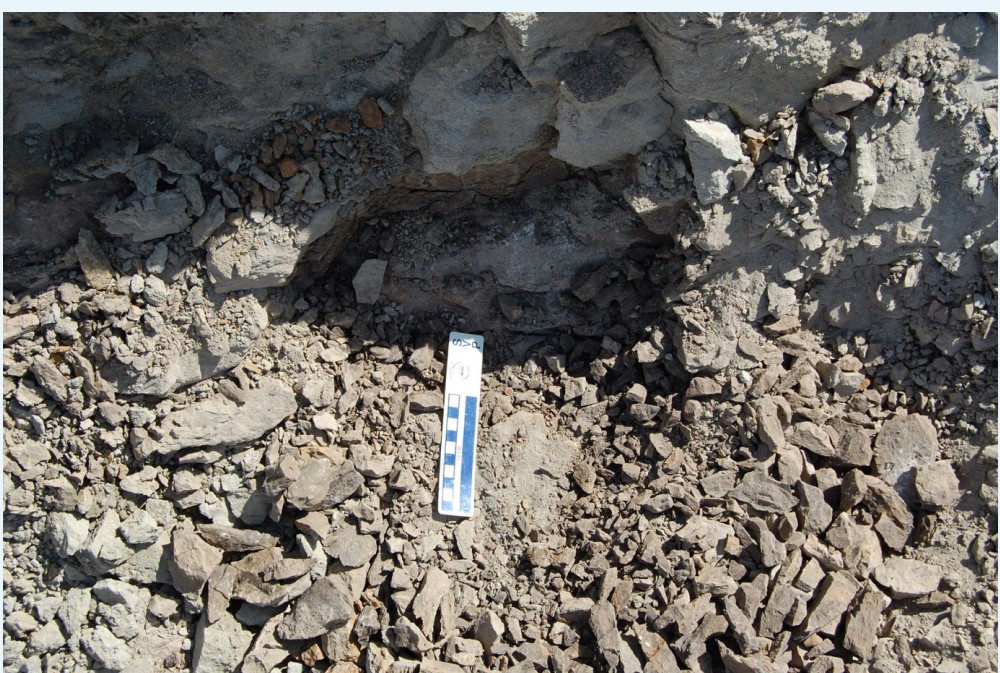

**Appendix 1—figure 16.** Mudstone from overburden-removed area of BB180 after sampling.

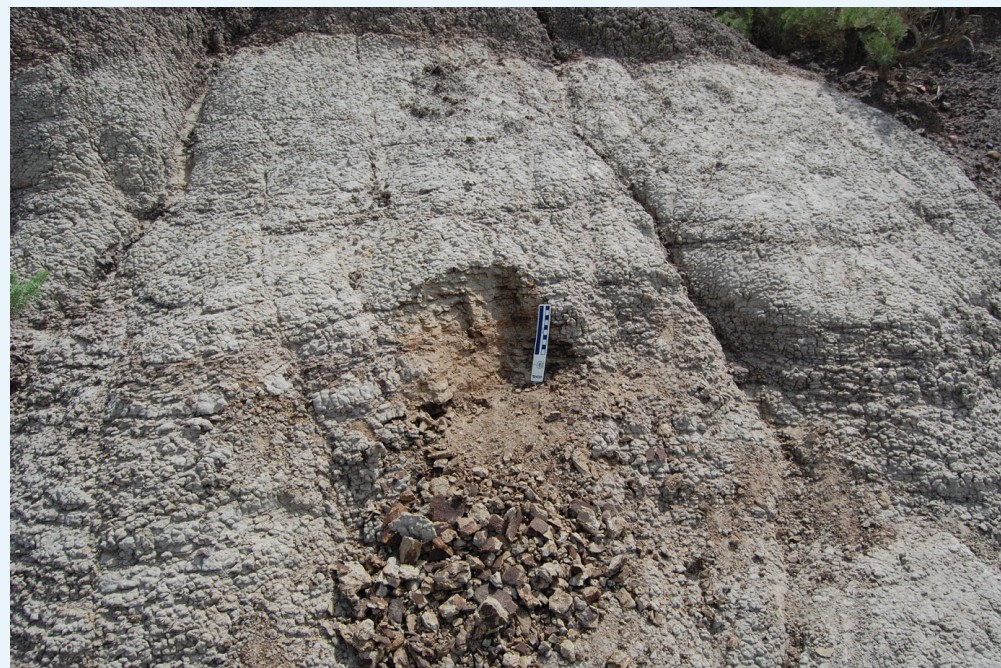

**Appendix 1—figure 17.** Mudstone collected from 709 m elevation after sampling.

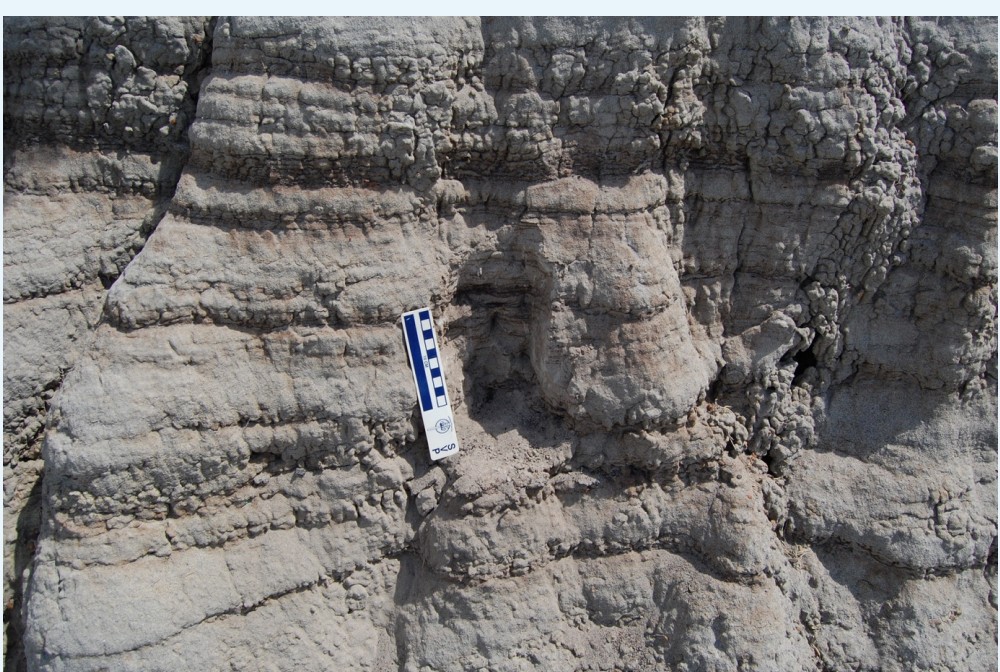

**Appendix 1—figure 18.** Mudstone from 693 m elevation after sampling.

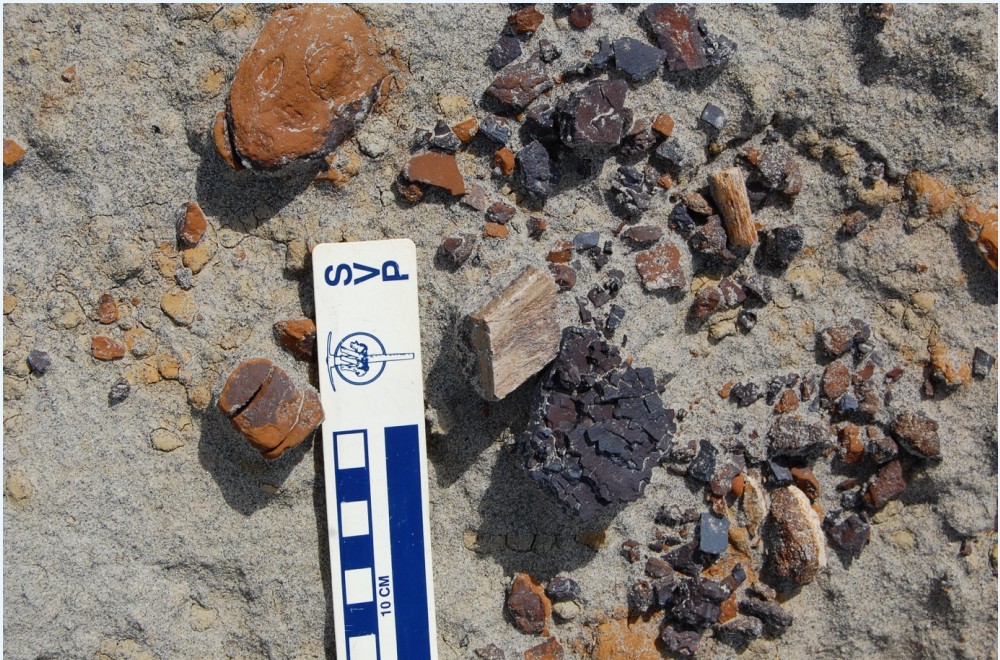

**Appendix 1—figure 19.** Surface eroded bone fragments from 691 m elevation as they were found.

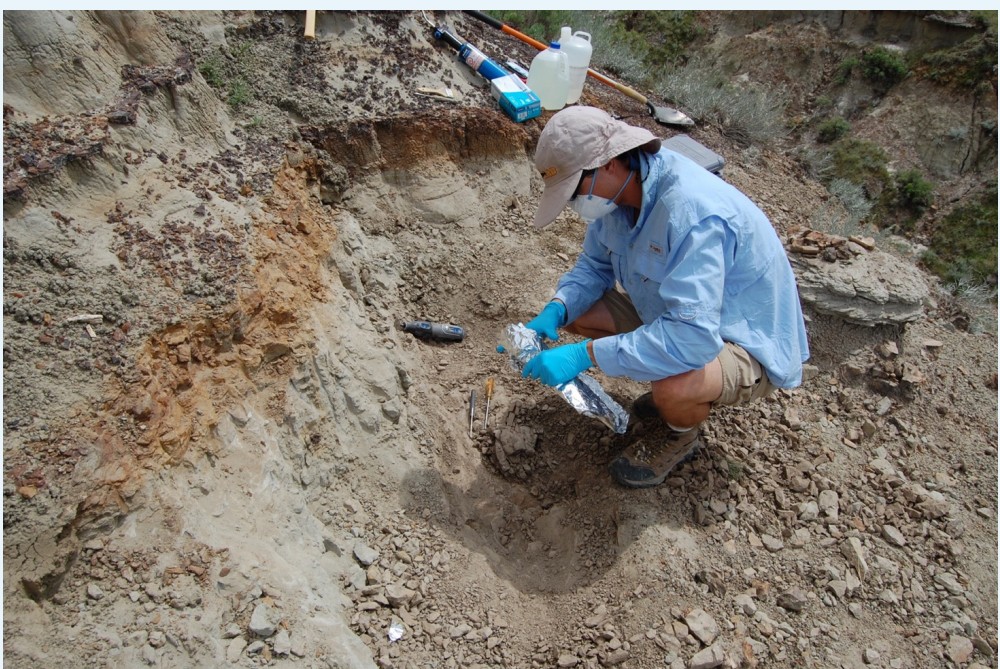

**Appendix 1—figure 20.** Aseptic collection of fossil samples in BB180 on July 8, 2016. Only skin of the face, wrists, and shins was exposed above the fossils. Shins remained uncovered to act as a thermal window for health and safety reasons to avoid overheating and to reduce the likelihood of contamination by lowering body temperature to reduce perspiration that might fall onto the samples. The body was positioned downhill of the bones at all times to compensate. Photograph by Kentaro Chiba.

## Powdering protocol details for Bristol replicates:

Samples included fossil bone and controls (fresh bone, mudstone, topsoil).

Materials:
- laminar flow hood
- 5% bleach
- 70% ethanol
- Gloves
- Petri dishes
- Large tray
- Foil
- Hydrolysis vials and Teflon liners (provided by University of York)
- Pen
- Mortar and pestle x 2
- Scoop to transfer powder for weighing x 2
- Lab balance
- Tubes to store excess powder
- Styrofoam/insulated box to transport to University of York
- Cold packs to transport to University of York

Method:
1. Clean surface of laminar flow bench with 5% bleach followed by 70% ethanol
2. Ethanol-rinsed replicates
    a. Rinse one replicate fragment of each sample/control in 70% ethanol
    b. Place fragments on petri dishes
    c. Place petri dishes in larger tray
    d. Cover larger tray with foil
    e. Place in fume hood for 3 days
3. Powder and hydrolysis tube loading

 a. Label hydrolysis vials
 b. Weigh vials on balance with caps and labels
 c. Rinse mortar, pestle, and scoop with 70% ethanol
 d. Place a foil 'placemat' under area where you will mortar and pestle
 e. Place fragment in pestle and grind into powder
 f. Scoop powder into hydrolysis vial, put excess in extra tubesRinse mortar, pestle, and scoop with 70% and foil 'placemat' between samples
 g. Rinse mortar, pestle, and scoop with 70 % ethanol and replace gloves and foil 'placemat' between samples
 h. Repeat for all non-treated and dried ethanol rinsed samples,
 i. Use separate mortar, pestle, and scoop for fresh chicken bone
 i. Reweigh vials on balance and pour out (into excess tubes) until <7 mg of powder is left
 i. Fresh chicken bone sample was handled after the others

4. Storage and travel
 a. Place loaded hydrolysis vials in box
 b. Place box in fridge until traveling to University of York (and in between previous steps when needed)
 c. Place box in Styrofoam insulated container with cold packs and tape shut to bring to University of York

5. Clean surface of laminar flow bench with 5% bleach followed by 70% ethanol

## HPLC amino acid analysis

### Pilot tests on Mesozoic fossil bone

Samples of non-aseptically collected surface eroded North American Mesozoic bone were cut using a non-sterilised, water-lubricated saw prior to powdering. In some cases, the saw was used to cut exterior and interior portions of the original fragment to analyse separately. Non-aseptically collected Mongolian bone fragments was powdered as they were.

**Appendix 1—table 3.** Pilot samples of fossil Mesozoic bone THAA composition. Concentrations in picomoles/mg.

| Taxon | Details | Location | Approximate age | Notes | [Asx] | [Glx] | [Ser] | [L-Thr] | [L-His] | [Gly] | [L-Arg] | [Ala] | [Tyr] | [Val] | [Phe] | [Leu] | [Ile] | [Total] |
|---|---|---|---|---|---|---|---|---|---|---|---|---|---|---|---|---|---|---|
| hadrosaur? | unrinsed, exterior, Dinosaur Park Fm | Dinosaur Provincial Park, Alberta, Canada | Late Cretaceous | signal suppression | 81 | 126 | 153 | 0 | 81.61 | 433 | 24 | 119 | 1.0 | 30 | 65 | 61 | 0 | 1174 |
| hadrosaur? | unrinsed, exterior, Dinosaur Park Fm | Dinosaur Provincial Park, Alberta, Canada | Late Cretaceous | signal suppression | 140 | 175 | 230 | 67 | 0.00 | 568 | 0 | 108 | 2.0 | 52 | 76 | 0 | 0 | 1417 |
| hadrosaur? | unrinsed, interior, Dinosaur Park Fm | Dinosaur Provincial Park, Alberta, Canada | Late Cretaceous | signal suppression | 57 | 97 | 135 | 28 | 0.00 | 432 | 30 | 73 | 3.0 | 26 | 38 | 0 | 0 | 919 |
| hadrosaur? | unrinsed, interior, Dinosaur Park Fm | Dinosaur Provincial Park, Alberta, Canada | Late Cretaceous | signal suppression | 58 | 102 | 123 | 31 | 0.00 | 354 | 30 | 84 | 4.0 | 30 | 32 | 31 | 0 | 902 |
| hadrosaur? | gelated, unrinsed, interior, Dinosaur Park Fm | Dinosaur Provincial Park, Alberta, Canada | Late Cretaceous | weaker signal suppression | 46 | 84 | 104 | 25 | 0.00 | 356 | 25 | 72 | 12.0 | 23 | 39 | 27 | 18 | 831 |
| sauropod? | unrinsed,, high in siderite, Morrison Fm | near Grass Range, Montana, USA | Late Jurassic | strong signal suppression | 0 | 127 | 0 | 0 | 0.00 | 482 | 0 | 149 | 13.0 | 0 | 0 | 0 | 0 | 772 |

*Appendix 1—table 3 continued on next page*

*Appendix 1—table 3 continued*

| Taxon | Details | Location | Approximate age | Notes | [Asx] | [Glx] | [Ser] | [L-Thr] | [L-His] | [Gly] | [L-Arg] | [Ala] | [Tyr] | [Val] | [Phe] | [Leu] | [Ile] | [Total] |
|---|---|---|---|---|---|---|---|---|---|---|---|---|---|---|---|---|---|---|
| hadrosaur? | ethanol rinsed, exterior, Dinosaur Park Fm | Dinosaur Provincial Park, Alberta, Canada | Late Cretaceous | strong signal suppression | 84 | 119 | 164 | 0 | 0.00 | 465 | 95 | 0 | 6.0 | 0 | 59 | 0 | 0 | 991 |
| hadrosaur? | ethanol rinsed, exterior, Dinosaur Park Fm | Dinosaur Provincial Park, Alberta, Canada | Late Cretaceous | strong signal suppression | 68 | 105 | 145 | 39 | 0.00 | 348 | 0 | 76 | 7.0 | 0 | 0 | 0 | 0 | 789 |
| hadrosaur? | gelated. ethanol rinsed, exterior, Dinosaur Park Fm | Dinosaur Provincial Park, Alberta, Canada | Late Cretaceous | weaker signal suppression | 56 | 93 | 132 | 35 | 0.00 | 294 | 0 | 62 | 14.0 | 37 | 40 | 32 | 21 | 816 |
| hadrosaur? | ethanol rinsed, interior, Dinosaur Park Fm | Dinosaur Provincial Park, Alberta, Canada | Late Cretaceous | signal suppression | 60 | 116 | 174 | 30 | 0.00 | 315 | 0 | 64 | 8.0 | 28 | 38 | 0 | 0 | 832 |
| hadrosaur? | ethanol rinsed, interior, Dinosaur Park Fm | Dinosaur Provincial Park, Alberta, Canada | Late Cretaceous | signal suppression | 57 | 104 | 164 | 38 | 73.67 | 286 | 0 | 56 | 9.0 | 28 | 33 | 30 | 16 | 895 |
| hadrosaur? | gelated, ethanol rinsed, interior, Dinosaur Park Fm | Dinosaur Provincial Park, Alberta, Canada | Late Cretaceous | weaker signal suppression | 50 | 98 | 149 | 32 | 42.43 | 280 | 13 | 58 | 15.0 | 26 | 39 | 30 | 18 | 848 |

*Appendix 1—table 3 continued on next page*

Appendix 1—table 3 continued

| Taxon | Details | Location | Approximate age | Notes | [Asx] | [Glx] | [Ser] | [L-Thr] | [L-His] | [Gly] | [L-Arg] | [Ala] | [Tyr] | [Val] | [Phe] | [Leu] | [Ile] | [Total] |
|---|---|---|---|---|---|---|---|---|---|---|---|---|---|---|---|---|---|---|
| sauropod? | gelated, ethanol rinsed, high in siderite, Morrison Fm | near Grass Range, Montana, USA | Late Jurassic | still strong signal suppression | 0 | 98 | 241 | 129 | 0.00 | 997 | 0 | 0 | 16.0 | 0 | 0 | 0 | 0 | 1481 |
| hadrosaur? | unrinsed, exterior, Dinosaur Park Fm | Dinosaur Provincial Park, Alberta, Canada | Late Cretaceous | no signal suppression | 9 | 3 | 7 | 1 | 0.00 | 21 | 1 | 12 | 12.0 | 6 | 4 | 1 | 5 | 82 |
| hadrosaur? | unrinsed, exterior, Dinosaur Park Fm | Dinosaur Provincial Park, Alberta, Canada | Late Cretaceous | no signal suppression | 8 | 4 | 40 | 8 | 5.98 | 37 | 1 | 15 | 13.0 | 6 | 5 | 4 | 4 | 152 |
| Alioramus | loaned from American Museum of Natural History | Mongolia | Late Cretaceous | poor amino acid profile | 0 | 0 | 0 | 0 | 0.00 | 0 | 0 | 0 | 17.0 | 0 | 0 | 0 | 0 | 17 |
| Alioramus | ethanol rinsed, loaned from American Museum of Natural History | Mongolia | Late Cretaceous | decent amino acid profile | 29 | 40 | 89 | 0 | 0.00 | 98 | 0 | 61 | 19.2 | 0 | 0 | 0 | 0 | 336 |

*Appendix 1—table 3 continued*

| Taxon | Details | Location | Approximate age | Notes | [Asx] | [Glx] | [Ser] | [L-Thr] | [L-His] | [Gly] | [L-Arg] | [Ala] | [Tyr] | [Val] | [Phe] | [Leu] | [Ile] | [Total] |
|---|---|---|---|---|---|---|---|---|---|---|---|---|---|---|---|---|---|---|
| *Alioramus* | gelated, ethanol rinsed, loaned from American Museum of Natural History | Mongolia | Late Cretaceous | poor amino acid profile | 0 | 0 | 0 | 0 | 0.00 | 0 | 0 | 0 | 21.4 | 0 | 0 | 0 | 0 | 21 |

**Appendix 1—table 4.** Pilot samples of fossil Mesozoic bone THAA D/L values.

| Taxon | Details | Location | Approximate age | Notes | Asx D/L | Glx D/L | Ser D/L | Arg D/L | Ala D/L | Val D/L | Phe D/L | Leu D/L | Ile D/L | Tyr D/L |
|---|---|---|---|---|---|---|---|---|---|---|---|---|---|---|
| hadrosaur? | unrinsed, exterior, Dinosaur Park Fm | Dinosaur Provincial Park, Alberta, Canada | Late Cretaceous | signal suppression | 0.185 | 0.121 | 0.000 | 0.000 | 0.552 | 0.000 | 0.000 | 1.071 | NA | NA |
| hadrosaur? | unrinsed, exterior, Dinosaur Park Fm | Dinosaur Provincial Park, Alberta, Canada | Late Cretaceous | signal suppression | 0.185 | 0.000 | 0.000 | NA | 0.000 | 0.000 | 0.000 | NA | NA | NA |
| hadrosaur? | unrinsed, interior, Dinosaur Park Fm | Dinosaur Provincial Park, Alberta, Canada | Late Cretaceous | signal suppression | 0.000 | 0.000 | 0.000 | 0.000 | 0.000 | 0.000 | 0.000 | NA | NA | 0.000 |
| hadrosaur? | unrinsed, interior, Dinosaur Park Fm | Dinosaur Provincial Park, Alberta, Canada | Late Cretaceous | signal suppression | 0.000 | 0.000 | 0.000 | 0.000 | 0.000 | 0.000 | 0.000 | 0.000 | 0.000 | NA |
| hadrosaur? | gelated, unrinsed, interior, Dinosaur Park Fm | Dinosaur Provincial Park, Alberta, Canada | Late Cretaceous | weaker signal suppression | 0.000 | 0.000 | 0.000 | 0.000 | 0.000 | 0.000 | 0.000 | 0.000 | 0.000 | NA |
| sauropod? | unrinsed, high in siderite, Morrison Fm | near Grass Range, Montana, USA | Late Jurassic | strong signal suppression | NA | 0.000 | NA | NA | 0.000 | NA | NA | NA | NA | NA |
| hadrosaur? | ethanol rinsed, exterior, Dinosaur Park Fm | Dinosaur Provincial Park, Alberta, Canada | Late Cretaceous | strong signal suppression | 0.000 | 0.000 | 0.000 | 0.000 | NA | NA | 0.000 | NA | NA | 0.000 |

Appendix 1—table 4 continued

| Taxon | Details | Location | Approximate age | Notes | Asx D/L | Glx D/L | Ser D/L | Arg D/L | Ala D/L | Val D/L | Phe D/L | Leu D/L | Ile D/L | Tyr D/L |
|---|---|---|---|---|---|---|---|---|---|---|---|---|---|---|
| hadrosaur? | ethanol rinsed, exterior, Dinosaur Park Fm | Dinosaur Provincial Park, Alberta, Canada | Late Cretaceous | strong signal suppression | 0.000 | 0.000 | 0.000 | NA | 0.000 | NA | NA | NA | NA | NA |
| hadrosaur? | gelated, ethanol rinsed, exterior, Dinosaur Park Fm | Dinosaur Provincial Park, Alberta, Canada | Late Cretaceous | weaker signal suppression | 0.000 | 0.000 | 0.000 | NA | 0.000 | 0.000 | 0.000 | 0.000 | 0.000 | NA |
| hadrosaur? | ethanol rinsed, interior, Dinosaur Park Fm | Dinosaur Provincial Park, Alberta, Canada | Late Cretaceous | signal suppression | 0.000 | 0.000 | 0.000 | NA | 0.000 | 0.000 | 0.000 | NA | NA | 0.000 |
| hadrosaur? | ethanol rinsed, interior, Dinosaur Park Fm | Dinosaur Provincial Park, Alberta, Canada | Late Cretaceous | signal suppression | 0.000 | 0.000 | 0.000 | NA | 0.000 | 0.000 | 0.000 | 0.000 | 0.000 | NA |
| hadrosaur? | gelated, ethanol rinsed, interior, Dinosaur Park Fm | Dinosaur Provincial Park, Alberta, Canada | Late Cretaceous | weaker signal suppression | 0.000 | 0.000 | 0.000 | 1.842 | 0.000 | 0.000 | 0.000 | 0.000 | 0.000 | NA |
| sauropod? | gelated, ethanol rinsed, high in siderite, Morrison Fm | near Grass Range, Montana, USA | Late Jurassic | still strong signal suppression | NA | 0.000 | 0.000 | NA | NA | NA | NA | NA | NA | NA |

Appendix 1—table 4 continued

| Taxon | Details | Location | Approximate age | Notes | Asx D/L | Glx D/L | Ser D/L | Arg D/L | Ala D/L | Val D/L | Phe D/L | Leu D/L | Ile D/L | Tyr D/L |
|---|---|---|---|---|---|---|---|---|---|---|---|---|---|---|
| hadrosaur? | unrinsed, exterior, Dinosaur Park Fm | Dinosaur Provincial Park, Alberta, Canada | Late Cretaceous | no signal suppression | 0.000 | 0.426 | 0.895 | 0.000 | 0.000 | 0.000 | 0.428 | 0.000 | 0.000 | 0.000 |
| hadrosaur? | unrinsed, exterior, Dinosaur Park Fm | Dinosaur Provincial Park, Alberta, Canada | Late Cretaceous | no signal suppression | 0.084 | 0.163 | 0.014 | 1.524 | 0.089 | 0.108 | 0.245 | 0.000 | 0.000 | 0.134 |
| Alioramus | loaned from American Museum of Natural History | Mongolia | Late Cretaceous | poor amino acid profile | NA | NA | NA | NA | NA | NA | NA | NA | NA | NA |
| Alioramus | ethanol rinsed, loaned from American Museum of Natural History | Mongolia | Late Cretaceous | decent amino acid profile | 0.000 | 0.000 | 0.000 | NA | 0.000 | NA | NA | NA | NA | NA |
| Alioramus | gelated, ethanol rinsed, loaned from American Museum of Natural History | Mongolia | Late Cretaceous | poor amino acid profile | NA | NA | NA | NA | NA | NA | NA | NA | NA | NA |

**Appendix 1—table 5.** Shark teeth THAA composition. Concentrations in picomoles/mg.

| Species | Details | Location | Approximate age | Notes | [Asx] | [Glx] | [Ser] | [L-Thr] | [L-His] | [Gly] | [L-Arg] | [Ala] | [Tyr] | [Val] | [Phe] | [Leu] | [Ile] | [Total] |
|---|---|---|---|---|---|---|---|---|---|---|---|---|---|---|---|---|---|---|
| *Carcharias taurus* | uninsed, unnamed Pleistocene-Holocene sediments | Ponte Vedra Beach, Florida, USA | Quaternary | high concentration | 52656 | 53337 | 29149 | 17064 | 4643.25 | 266160 | 31884 | 76481 | 5.0 | 16577 | 9366 | 20462 | 19071 | 596855 |
| *Carcharias taurus* | ethanol rinsed, unnamed Pleistocene-Holocene sediments | Ponte Vedra Beach, Florida, USA | Quaternary | weak signal suppression | 478 | 471 | 104 | 65 | 58.31 | 2309 | 188 | 430 | 10.0 | 194 | 93 | 229 | 195 | 4824 |
| *Carcharias taurus* | ethanol rinsed, unnamed Pleistocene-Holocene sediments | Ponte Vedra Beach, Florida, USA | Quaternary | weak signal suppression | 588 | 534 | 115 | 73 | 61.79 | 2617 | 203 | 537 | 11.0 | 261 | 120 | 272 | 255 | 5647 |

**Appendix 1—table 6.** Shark teeth THAA D/L values.

| Species | Details | Location | Approximate age | Notes | Asx D/L | Glx D/L | Ser D/L | Arg D/L | Ala D/L | Val D/L | Phe D/L | Leu D/L | Ile D/L | Tyr D/L |
|---------|---------|----------|-----------------|-------|---------|---------|---------|---------|---------|---------|---------|---------|---------|---------|
| *Carcharias taurus* | uninsed, unnamed Pleistocene-Holocene sediments | Ponte Vedra Beach, Florida, USA | Quaternary | high concentration | 0.209 | 0.039 | 0.092 | 0.047 | 0.027 | 0.011 | 0.028 | 0.026 | 0.165 | 0.114 |
| *Carcharias taurus* | ethanol rinsed, unnamed Pleistocene-Holocene sediments | Ponte Vedra Beach, Florida, USA | Quaternary | weak signal suppression | 0.512 | 0.153 | 0.301 | 0.236 | 0.155 | 0.114 | 0.000 | 0.092 | 0.107 | 0.000 |
| *Carcharias taurus* | ethanol rinsed, unnamed Pleistocene-Holocene sediments | Ponte Vedra Beach, Florida, USA | Quaternary | weak signal suppression | 0.527 | 0.154 | 0.295 | 0.358 | 0.166 | 0.112 | 0.158 | 0.094 | 0.106 | NA |

**Aseptically collected samples at Dinosaur Provincial Park and modern chicken control**

**Appendix 1—table 7.** Aseptically collected samples and chicken control THAA composition.
See **Appendix 1—table 2** for Sample ID's. e = ethanol rinsed before powdering. d = gelated. Concentrations in picomoles/mg.

| Sample ID | [Asx] | [Glx] | [Ser] | [L-Thr] | [L-His] | [Gly] | [L-Arg] | [Ala] | [Tyr] | [Val] | [Phe] | [Leu] | [Ile] | [Total] |
|---|---|---|---|---|---|---|---|---|---|---|---|---|---|---|
| 1 | 71588 | 105116 | 45564 | 33327 | 17233.81 | 385402 | 58057 | 136671 | 14.7 | 38380 | 26397 | 57808 | 24829 | 1000385 |
| 1e | 81260 | 124003 | 51060 | 36547 | 22168.25 | 445966 | 66261 | 157216 | 17.4 | 42057 | 28714 | 56845 | 27166 | 1139278 |
| 2 | 0 | 0 | 0 | 0 | 0.00 | 209 | 0 | 0 | 14.8 | 0 | 0 | 0 | 0 | 224 |
| 2 | 0 | 41 | 0 | 0 | 0.00 | 182 | 0 | 0 | 15.0 | 0 | 0 | 0 | 0 | 239 |
| 2e | 56 | 60 | 0 | 0 | 0.00 | 181 | 0 | 68 | 17.6 | 0 | 0 | 0 | 0 | 383 |
| 2ed | 0 | 0 | 0 | 0 | 0.00 | 33 | 0 | 0 | 19.8 | 0 | 0 | 0 | 0 | 52 |
| 3 | 18 | 20 | 56 | 35 | 0.00 | 245 | 0 | 97 | 15.2 | 38 | 216 | 46 | 0 | 787 |
| 3 | 35 | 32 | 99 | 45 | 0.00 | 190 | 0 | 117 | 15.4 | 49 | 105 | 45 | 33 | 765 |
| 3e | 0 | 0 | 20 | 0 | 0.00 | 111 | 0 | 76 | 17.8 | 0 | 0 | 0 | 0 | 224 |
| 3ed | 0 | 0 | 24 | 17 | 0.00 | 92 | 0 | 75 | 20.0 | 28 | 0 | 0 | 37 | 294 |
| 4 | 29 | 32 | 43 | 0 | 0.00 | 158 | 0 | 41 | 15.6 | 0 | 43 | 0 | 0 | 360 |
| 4e | 234 | 227 | 177 | 108 | 0.00 | 509 | 0 | 220 | 18.0 | 148 | 0 | 0 | 0 | 1642 |
| 4ed | 87 | 110 | 51 | 45 | 0.00 | 191 | 17 | 95 | 20.2 | 57 | 29 | 53 | 36 | 791 |
| 5 | 0 | 0 | 0 | 0 | 0.00 | 0 | 0 | 0 | 15.8 | 0 | 0 | 0 | 0 | 16 |
| 5e | 0 | 64 | 0 | 0 | 0.00 | 184 | 0 | 0 | 18.1 | 0 | 0 | 0 | 0 | 266 |
| 5ed | 15 | 40 | 16 | 15 | 0.00 | 79 | 0 | 33 | 20.3 | 16 | 0 | 19 | 0 | 255 |
| 6 | 104 | 131 | 100 | 91 | 0.00 | 533 | 66 | 164 | 15.9 | 77 | 92 | 102 | 50 | 1526 |
| 6 | 148 | 144 | 116 | 97 | 0.00 | 318 | 52 | 163 | 16.1 | 89 | 59 | 104 | 55 | 1361 |
| 6e | 367 | 528 | 305 | 284 | 66.15 | 723 | 177 | 503 | 18.3 | 253 | 160 | 332 | 162 | 3878 |
| 6ed | 57 | 89 | 68 | 63 | 0.00 | 177 | 38 | 106 | 20.5 | 64 | 32 | 72 | 38 | 825 |
| 7 | 932 | 643 | 1516 | 1157 | 904.34 | 5438 | 898 | 2528 | 16.3 | 1153 | 675 | 1226 | 651 | 17737 |
| 7e | 3541 | 3068 | 3467 | 3424 | 799.16 | 11568 | 2024 | 6377 | 18.5 | 3508 | 1808 | 3972 | 2061 | 45636 |
| 7ed | 3505 | 2717 | 1674 | 1818 | 178.64 | 4659 | 541 | 3125 | 20.7 | 1947 | 900 | 2035 | 1148 | 24268 |

*Appendix 1—table 7 continued on next page*

*Appendix 1—table 7 continued*

| Sample ID | [Asx] | [Glx] | [Ser] | [L-Thr] | [L-His] | [Gly] | [L-Arg] | [Ala] | [Tyr] | [Val] | [Phe] | [Leu] | [Ile] | [Total] |
|---|---|---|---|---|---|---|---|---|---|---|---|---|---|---|
| 8 | 393 | 306 | 262 | 282 | 0.00 | 834 | 142 | 516 | 16.5 | 323 | 175 | 415 | 217 | 3880 |
| 8e | 0 | 0 | 0 | 0 | 0.00 | 0 | 0 | 0 | 18.7 | 0 | 0 | 0 | 0 | 19 |
| 8ed | 9 | 11 | 15 | 7 | 0.00 | 24 | 0 | 17 | 20.9 | 9 | 0 | 0 | 0 | 114 |
| 9 | 0 | 0 | 0 | 0 | 0.00 | 0 | 0 | 0 | 16.7 | 0 | 0 | 0 | 0 | 17 |
| 9e | 0 | 0 | 36 | 0 | 0.00 | 0 | 0 | 34 | 18.9 | 0 | 0 | 0 | 0 | 89 |
| 9ed | 18 | 48 | 45 | 0 | 0.00 | 82 | 0 | 35 | 21.1 | 0 | 0 | 0 | 0 | 249 |
| 10 | 59 | 77 | 47 | 40 | 0.00 | 185 | 38 | 104 | 16.9 | 54 | 0 | 68 | 0 | 688 |
| 10e | 39 | 199 | 0 | 0 | 0.00 | 148 | 0 | 70 | 19.1 | 139 | 0 | 169 | 0 | 782 |
| 10ed | 15 | 86 | 23 | 0 | 0.00 | 78 | 0 | 43 | 21.3 | 69 | 15 | 65 | 45 | 462 |

**Appendix 1—table 8. Aseptically collected samples and chicken control THAA D/L values.**
See *Appendix 1—table 2* for Sample ID's. e = ethanol rinsed before powdering. d = gelated.

| Sample ID | Asx D/L | Glx D/L | Ser D/L | Arg D/L | Ala D/L | Val D/L | Phe D/L | Leu D/L | Ile D/L | Tyr D/L |
|---|---|---|---|---|---|---|---|---|---|---|
| 1 | 0.053 | 0.027 | 0.000 | 0.096 | 0.015 | 0.000 | 0.019 | 0.022 | 0.000 | 0.000 |
| 1e | 0.055 | 0.029 | 0.000 | 0.081 | 0.016 | 0.000 | 0.000 | 0.000 | 0.000 | 0.000 |
| 2 | NA | NA | NA | NA | NA | NA | NA | NA | NA | NA |
| 2 | NA | 0.000 | NA | NA | NA | NA | NA | NA | NA | NA |
| 2e | 0.000 | 0.000 | NA | NA | 0.000 | NA | NA | NA | NA | NA |
| 2ed | NA | NA | NA | NA | NA | NA | NA | NA | NA | NA |
| 3 | 0.000 | 0.000 | 0.000 | NA | 0.000 | 0.000 | 0.000 | 0.000 | NA | NA |
| 3 | 0.000 | 0.000 | 0.000 | NA | 0.000 | 0.000 | 0.000 | 0.000 | 0.000 | NA |
| 3e | NA | NA | 0.000 | NA | 0.338 | NA | NA | NA | NA | NA |
| 3ed | NA | NA | 0.000 | NA | 0.299 | 0.000 | NA | NA | 0.000 | NA |
| 4 | 0.000 | 0.000 | 0.000 | NA | 0.000 | NA | 0.000 | NA | NA | NA |
| 4e | 0.000 | 0.000 | 0.000 | NA | 0.000 | 0.000 | NA | NA | NA | NA |
| 4ed | 0.214 | 0.550 | 0.000 | 0.989 | 0.207 | 0.000 | 0.000 | 0.000 | 0.000 | NA |
| 5 | NA | NA | NA | NA | NA | NA | NA | NA | NA | NA |
| 5e | NA | 0.000 | NA | NA | NA | NA | NA | NA | NA | NA |
| 5ed | 0.000 | 0.727 | 0.000 | NA | 0.000 | 0.000 | NA | 0.000 | NA | NA |
| 6 | 0.000 | 0.000 | 0.000 | 1.996 | 0.000 | 0.000 | 0.000 | 0.000 | 0.000 | NA |
| 6 | 0.110 | 0.000 | 0.000 | 1.645 | 0.000 | 0.000 | 0.000 | 0.000 | 0.000 | NA |
| 6e | 0.082 | 0.084 | 0.000 | 0.889 | 0.078 | 0.000 | 0.000 | 0.000 | 0.000 | NA |
| 6ed | 0.000 | 0.000 | 0.000 | 1.338 | 0.000 | 0.000 | 0.000 | 0.000 | 0.000 | NA |
| 7 | 0.418 | 0.378 | 0.020 | 0.887 | 0.071 | 0.000 | 0.000 | 0.056 | 0.000 | 0.000 |
| 7e | 0.130 | 0.123 | 0.038 | 0.856 | 0.080 | 0.038 | 0.048 | 0.070 | 0.000 | 0.000 |
| 7ed | 0.135 | 0.139 | 0.045 | 0.639 | 0.089 | 0.040 | 0.049 | 0.063 | 0.000 | NA |
| 8 | 0.125 | 0.000 | 0.000 | 1.148 | 0.000 | 0.000 | 0.000 | 0.000 | 0.000 | 0.000 |
| 8e | NA | NA | NA | NA | NA | NA | NA | NA | NA | NA |
| 8ed | 0.000 | 0.000 | 0.000 | NA | 0.000 | 0.000 | NA | NA | NA | NA |
| 9 | NA | NA | NA | NA | NA | NA | NA | NA | NA | NA |
| 9e | NA | NA | 0.000 | NA | 0.000 | NA | NA | NA | NA | NA |
| 9ed | 0.000 | 1.055 | 0.000 | NA | 0.000 | NA | NA | NA | NA | NA |
| 10 | 0.000 | 0.000 | 0.000 | 0.000 | 0.000 | 0.000 | NA | 0.000 | NA | NA |
| 10e | 0.000 | 0.538 | NA | NA | 0.000 | 0.750 | NA | 0.675 | NA | NA |
| 10ed | 0.000 | 0.951 | 0.000 | NA | 0.323 | 0.901 | 0.000 | 0.751 | 0.677 | NA |

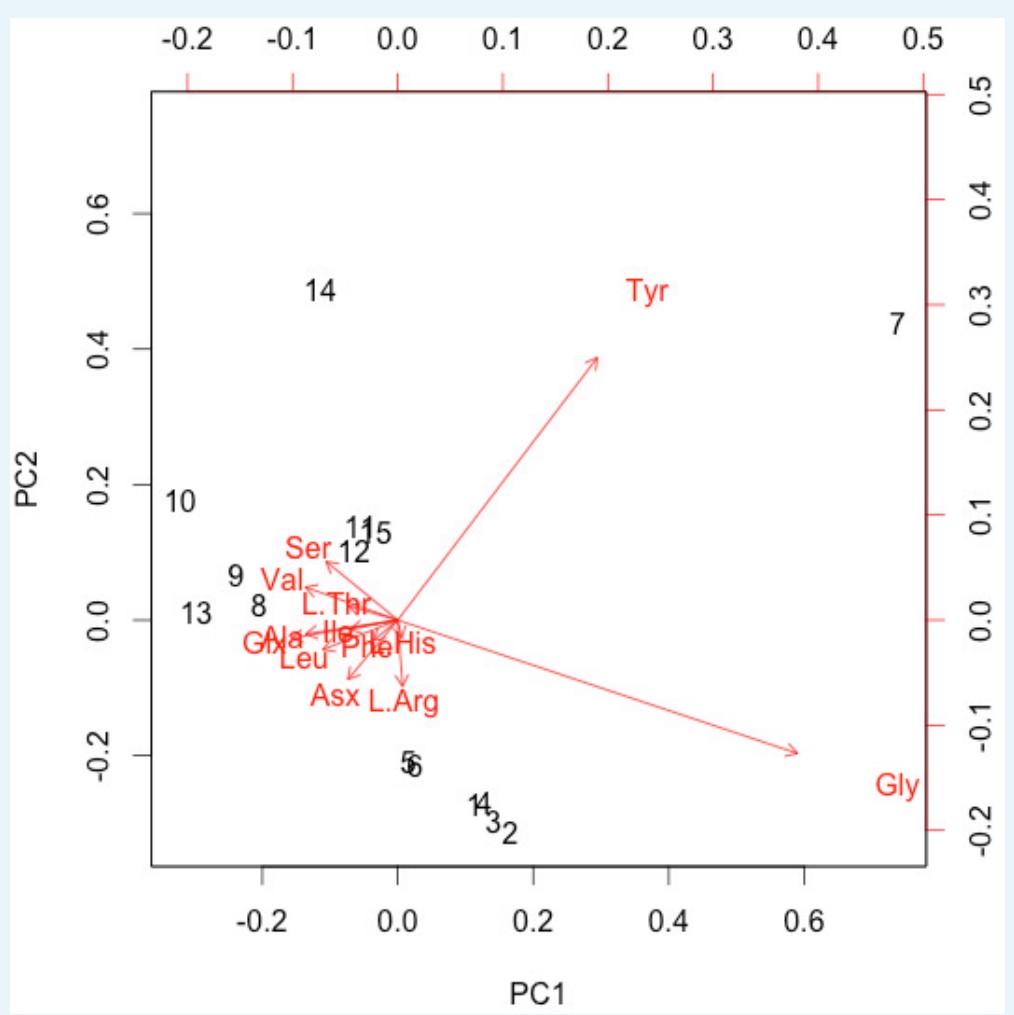

**Appendix 1—figure 21.** Non-normalised PCA biplot of THAA composition of only the sufficiently treated samples associated with the plot in *Figure 6F*.

**Appendix 1—table 9.** Summary of variable loadings for the non-normalised PCA of THAA composition of only the sufficiently treated samples associated with *Figure 6F*. Proportion of variance for each principal component in parentheses.

|  | PC1 (55.04%) | PC2 (22.66%) |
| --- | --- | --- |
| Asx | −0.100509897 | −0.18614726 |
| Glx | −0.214376490 | −0.05665852 |
| Ser | −0.144286512 | 0.18384010 |
| L. Thr | −0.098805431 | 0.04356834 |
| L. His | 0.007992613 | −0.05584073 |
| Gly | 0.805294439 | −0.41882274 |
| L. Arg | 0.009652324 | −0.20786057 |
| Alo | −0.185328976 | −0.04346374 |
| Tyr | 0.401827150 | 0.82375170 |
| Val | −0.186019066 | 0.10343138 |
| Phe | −0.048918108 | −0.06420351 |
| Leu | −0.150726139 | −0.09272073 |
| Ile | −0.095795908 | −0.02887370 |

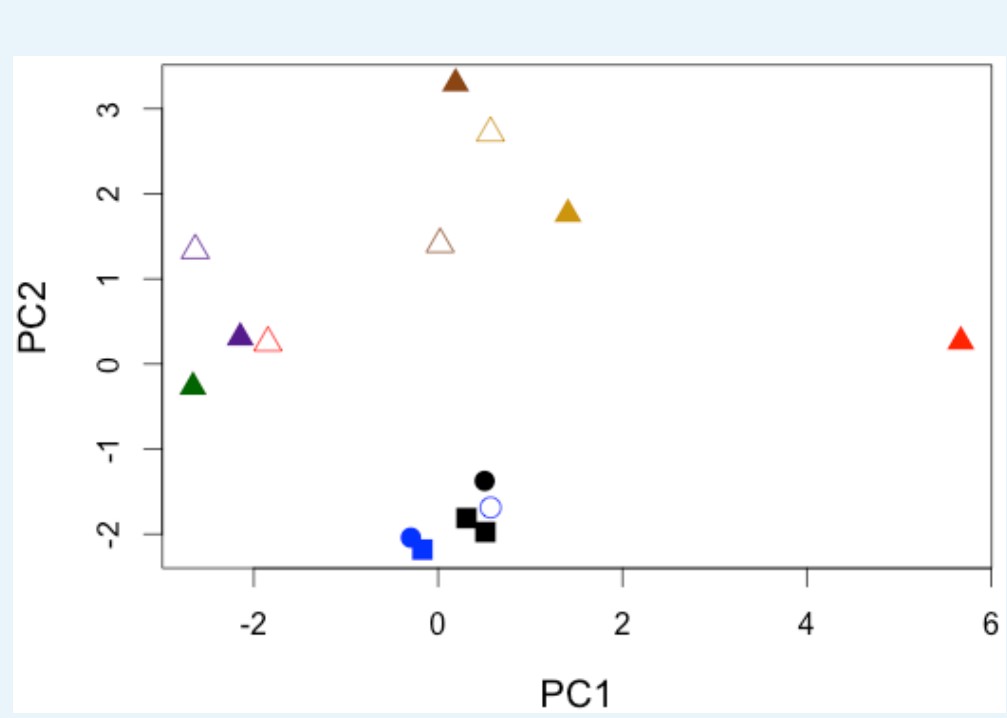

**Appendix 1—figure 22.** Normalised PCA of THAA composition of only the sufficiently treated samples. prcomp() function in R scale set to 'TRUE'. Color and shape coding identical to that in *Appendix 1—figure 24*.

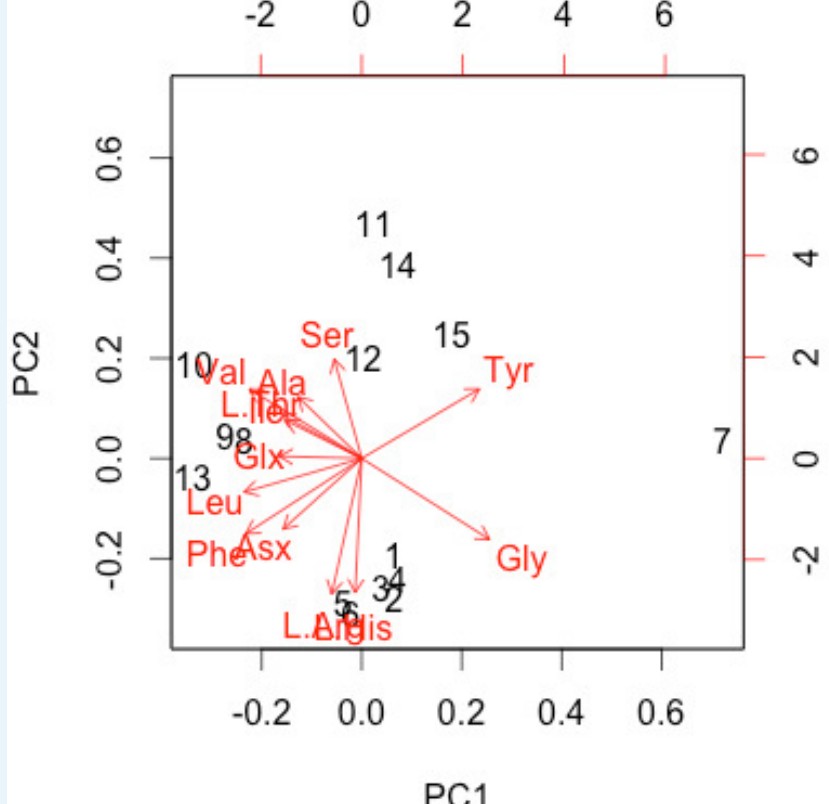

**Appendix 1—figure 23.** Normalised PCA biplot of THAA composition of only the sufficiently

treated samples associated with *Appendix 1—figure 22*.

**Appendix 1—table 10.** Summary of variable loadings for the normalised PCA of THAA composition of only the sufficiently treated samples associated with *Appendix 1—figure 22*. Proportion of variance for each principal component in parentheses.

|  | PC1 (31.64%) | PC2 (25.42%) |
|---|---|---|
| Asx | −0.24729982 | −0.247493801 |
| Glx | −0.25925017 | 0.007025074 |
| Ser | −0.08672307 | 0.347552010 |
| L. Thr | −0.25282104 | 0.153203703 |
| L. His | −0.02073369 | −0.470955522 |
| Gly | 0.40229882 | −0.284333925 |
| L. Arg | −0.09443713 | −0.475273301 |
| Alo | −0.20019839 | 0.215195078 |
| Tyr | 0.37012033 | 0.243369947 |
| Val | −0.35372396 | 0.243119598 |
| Phe | −0.36466490 | −0.264010157 |
| Leu | −0.36975651 | −0.118102708 |
| Ile | −0.23897848 | 0.132700762 |

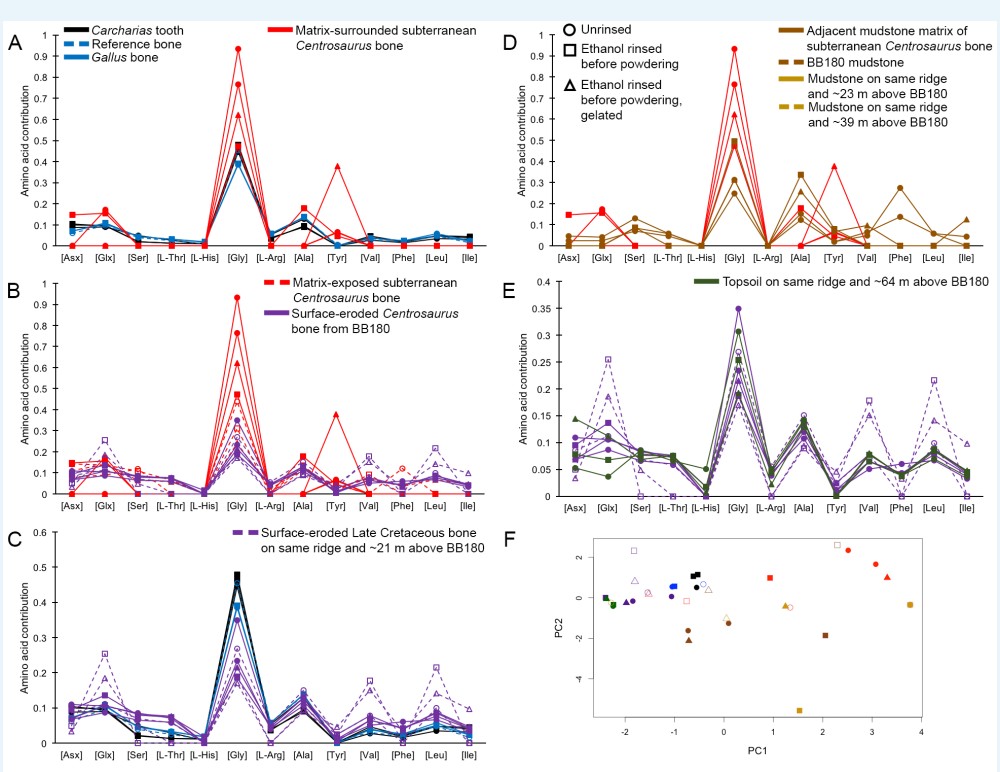

**Appendix 1—figure 24.** THAA compositional profiles of the samples based on amino acid percentages. (**A**) Late Cretaceous subterranean bone (red) compared to non-aseptically collected Pleistocene-Holocene teeth (black) and modern bone (blue). (**B**) Late Cretaceous subterranean bone (red) compared to surface-eroded Late Cretaceous bone from the same outcrop (purple). (**C**) Surface-eroded Late Cretaceous bone (purple) compared to Pleistocene-Holocene teeth (black) and modern bone (blue). (**D**) Late Cretaceous

subterranean bone aseptically collected (red) compared to the adjacent mudstone matrix (brown). (**E**) Surface-eroded Late Cretaceous bone (purple) compared to topsoil at higher elevation (i.e., prairie level) on the same ridge (green). (**F**) PCA on normalised amino acid percentages (see A–E legends). See Appendix 1 (*Appendix 1—figure 25*; *Appendix 1—table 11*) for PCA summary. Color and symbol coding is constant throughout.

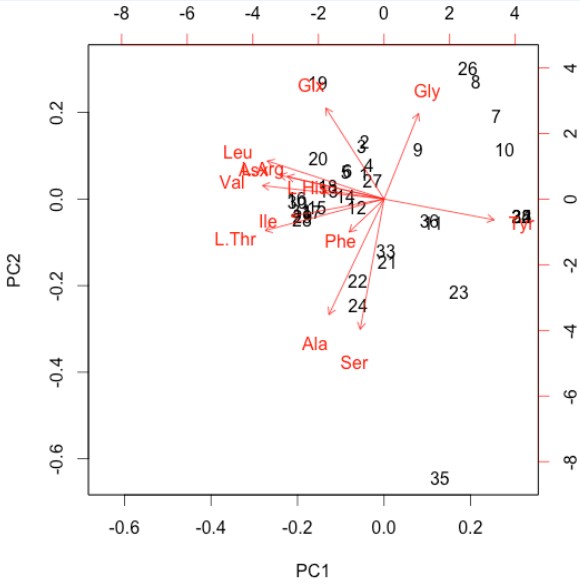

**Appendix 1—figure 25.** Normalised PCA biplot of THAA composition associated with the plot in *Appendix 1—figure 24F*.

**Appendix 1—table 11.** Summary of variable loadings for the normalised PCA of THAA composition associated with the plot in *Appendix 1—figure 24F*. Proportion of variance for each principal component in parentheses.

|  | PC1 (30.01%) | PC2 (15.97%) |
| --- | --- | --- |
| Asx | −0.3356269 | 0.10557119 |
| Glx | −0.18736532 | 0.40091324 |
| Ser | −0.07675319 | −0.57396909 |
| L. Thr | −0.38235418 | −0.13868623 |
| L. His | −0.19702491 | 0.04051298 |
| Gly | 0.11068434 | 0.37647168 |
| L. Arg | −0.31147285 | 0.09964743 |
| Alo | −0.17744465 | −0.50906983 |
| Tyr | 0.35387327 | −0.09153454 |
| Val | −0.39069929 | 0.05903456 |
| Phe | −0.11192988 | −0.14530964 |
| Leu | −0.37568445 | 0.16792075 |
| Ile | −0.29841571 | −0.07628272 |

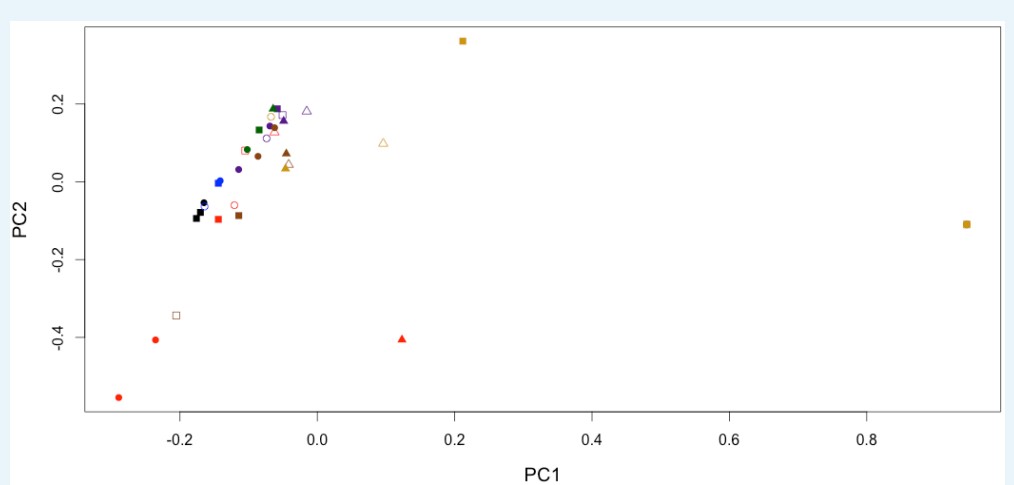

**Appendix 1—figure 26.** Non-normalised PCA of THAA composition. prcomp() function in R scale set to 'FALSE'. Color and shape coding identical to that in *Appendix 1—figure 24*.

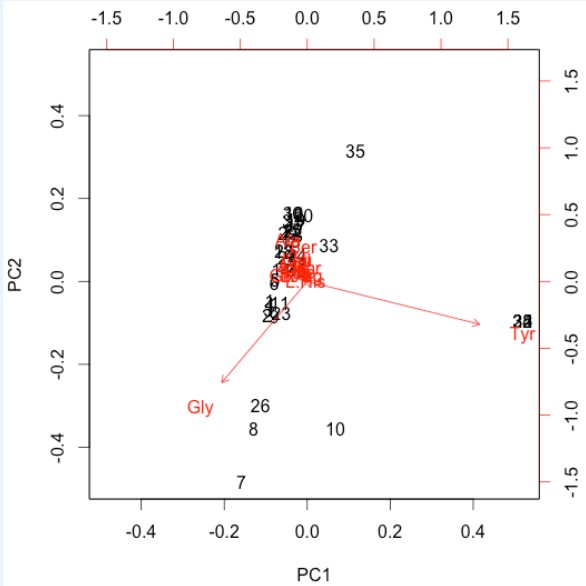

**Appendix 1—figure 27.** Non-normalised PCA biplot of THAA composition associated with *Appendix 1—figure 26*. prcomp() function in R scale set to 'FALSE'.

**Appendix 1—table 12.** Summary of variable loadings for the non-normalised PCA of THAA composition associated with *Appendix 1—figure 26*. Proportion of variance for each principal component in parentheses.

| | PC1 (62.09%) | PC2 (24.49%) |
|---|---|---|
| Asx | −0.061334126 | 0.101860246 |
| Glx | −0.101689512 | 0.042538223 |
| Ser | −0.017584775 | 0.227710833 |
| L. Thr | −0.026788291 | 0.091854076 |
| L. His | −0.006334389 | 0.006290775 |
| Gly | −0.438716137 | −0.826790362 |
| L. Arg | −0.024360271 | 0.030568893 |
| Alo | −0.079762251 | 0.26975096 |

*Appendix 1—table 12 continued on next page*

*Appendix 1—table 12 continued*

|  | PC1 (62.09%) | PC2 (24.49%) |
|---|---|---|
| Tyr | 0.883822618 | −0.350122096 |
| Val | −0.035750618 | 0.142845467 |
| Phe | −0.029804014 | 0.062677703 |
| Leu | −0.040620928 | 0.139913899 |
| Ile | −0.021077307 | 0.060901383 |

**Appendix 1—figure 28.** THAA concentrations (summed total of all amino acids measured) of the samples. (**A**) Logarithmic scale comparison of modern Gallus bone (blue), matrix-surrounded subterranean Centrosaurus bone (red), Pleistocene-Holocene surface-eroded shark teeth (black, with a repeated measurement for the ethanol rinsed sample), and topsoil on same ridge and ~64 m above BB180 (green). (**B**) Comparison between fossil Late Cretaceous bone and mudstone. Matrix-surrounded subterranean Centrosaurus bone (solid red), adjacent mudstone matrix of subterranean Centrosaurus bone (solid brown), uncovered subterranean Centrosaurus bone (open red), BB180 mudstone (open brown), surface-eroded Centrosaurus bone from BB180 (solid purple), mudstone on same ridge and ~39 m above BB180 (open tan), mudstone on same ridge and ~23 m above BB180 (solid tan), and surface-eroded Late Cretaceous bone on same ridge and ~21 m above BB180 (open purple). Gelated replicates likely provide the most accurate measurements given the peak reduction present in the non-gelated replicates.

**Appendix 1—table 13.** Comparison of Late Cretaceous, Pleistocene-Holocene, and modern amino acid racemisation values. NA indicates that amino acid concentration was below detection limit.

| Sample treatment | Asx D/L | Glx D/L | Ser D/L | Ala D/L | Val D/L |
|---|---|---|---|---|---|
| Matrix-surrounded subterranean *Centrosaurus* bone | | | | | |
| Unrinsed | NA | NA | NA | NA | NA |
| Unrinsed | NA | 0 | NA | NA | NA |
| Ethanol rinsed before powdering | 0 | 0 | NA | 0 | NA |
| Ethanol rinsed before powdering, gelated | NA | NA | NA | NA | NA |
| Subterranean *Centrosaurus* bone uncovered from matrix before collection | | | | | |
| Unrinsed | 0 | 0 | 0 | 0 | NA |
| Ethanol rinsed before powdering | 0 | 0 | 0 | 0 | 0 |
| Ethanol rinsed before powdering, gelated | 0.214 | 0.550 | 0 | 0.207 | 0 |
| Adjacent mudstone matrix of subterranean *Centrosaurus* bone | | | | | |
| Unrinsed | 0 | 0 | 0 | 0 | 0 |
| Unrinsed | 0 | 0 | 0 | 0 | 0 |
| Ethanol rinsed before powdering | NA | NA | 0 | 0.338 | NA |
| Ethanol rinsed before powdering, gelated | NA | NA | 0 | 0.299 | 0 |
| Surface-eroded *Centrosaurus* bone from BB180 | | | | | |
| Unrinsed | 0 | 0 | 0 | 0 | 0 |
| Unrinsed | 0.110 | 0 | 0 | 0 | 0 |
| Ethanol rinsed before powdering | 0.082 | 0.084 | 0 | 0.078 | 0 |
| Ethanol rinsed before powdering, gelated | 0 | 0 | 0 | 0 | 0 |
| Surface-eroded Late Cretaceous bone on same ridge and ~ 21 m above BB180 | | | | | |
| Unrinsed | 0 | 0 | 0 | 0 | 0 |
| Ethanol rinsed before powdering | 0 | 0.538 | NA | 0 | 0.750 |
| Ethanol rinsed before powdering, gelated | 0 | 0.951 | 0 | 0.323 | 0.901 |
| Topsoil on same ridge and ~ 64 m above BB180 | | | | | |
| Unrinsed | 0.418 | 0.378 | 0.020 | 0.071 | 0 |
| Ethanol rinsed before powdering | 0.130 | 0.123 | 0.038 | 0.080 | 0.038 |
| Ethanol rinsed before powdering, gelated | 0.135 | 0.139 | 0.045 | 0.089 | 0.040 |
| Pleistocene-Holocene surface-eroded *Carcharias* teeth | | | | | |
| Unrinsed | 0.209 | 0.039 | 0.092 | 0.027 | 0.011 |
| Ethanol rinsed before powdering | 0.512 | 0.153 | 0.301 | 0.155 | 0.114 |
| Ethanol rinsed before powdering | 0.527 | 0.154 | 0.295 | 0.166 | 0.112 |
| Modern *Gallus* bone | | | | | |
| Unrinsed | 0.053 | 0.027 | 0 | 0.015 | 0 |
| Ethanol rinsed before powdering | 0.055 | 0.029 | 0 | 0.016 | 0 |

## ATR FTIR

CSV files containing the raw spectral data of demineralised samples are available through the Field Museum's collections database: multimedia record containing raw files (GUID: 60c79cec-4da1-4bea-8535-a332c70ae4c9, URI: https://mm.fieldmuseum.org/60c79cec-4da1-

4bea-8535-a332c70ae4c9), event record with surrounding information about the project (GUID: 34e15532-2c46-47cf-aac0-5d29cc5a2c22, URI: https://pj.fieldmuseum.org/event/34e15532-2c46-47cf-aac0-5d29cc5a2c22) and *Source data 1*.

## Py-GC-MS

CDF and RAW files containing the raw chromatogram and spectrometry data are available through the Field Museum's collections database: multimedia record containing raw files (GUID: 60c79cec-4da1-4bea-8535-a332c70ae4c9, URI: https://mm.fieldmuseum.org/60c79cec-4da1-4bea-8535-a332c70ae4c9), event record with surrounding information about the project (GUID: 34e15532-2c46-47cf-aac0-5d29cc5a2c22, URI: https://pj.fieldmuseum.org/event/34e15532-2c46-47cf-aac0-5d29cc5a2c22). See *Appendix 1—table 2* for sample ID's. Data for one of the pilot *Alioramus* bone samples is also included in *Source data 1*. All samples ethanol rinsed before powdering.

## VPSEM

Demineralised modern chicken bone

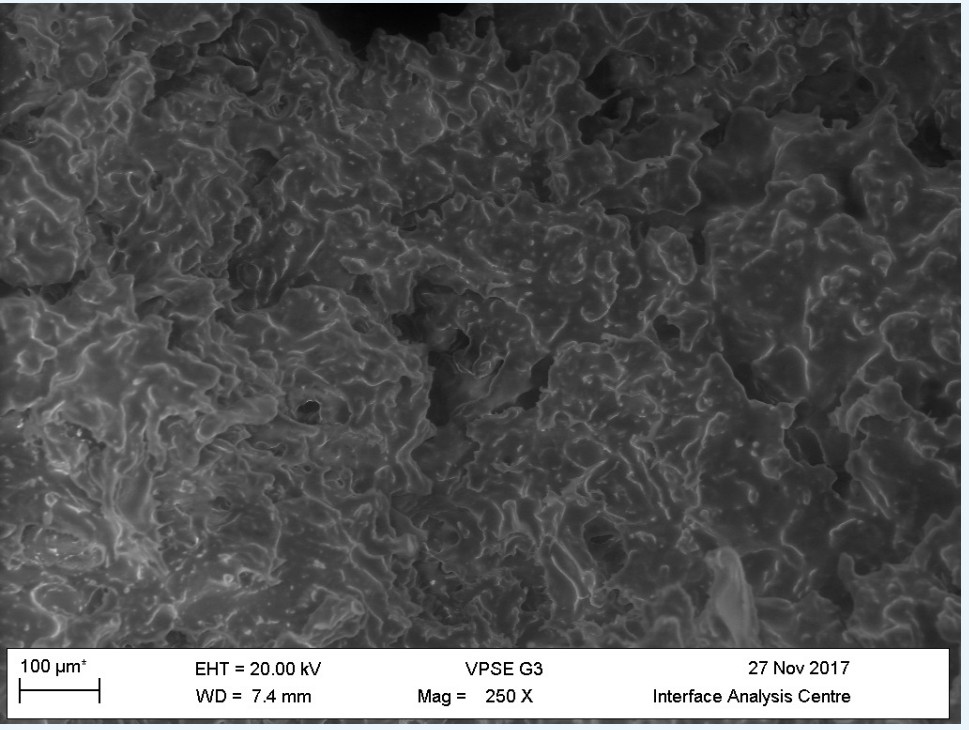

**Appendix 1—figure 29.** Demineralised modern chicken bone viewed at a distance.

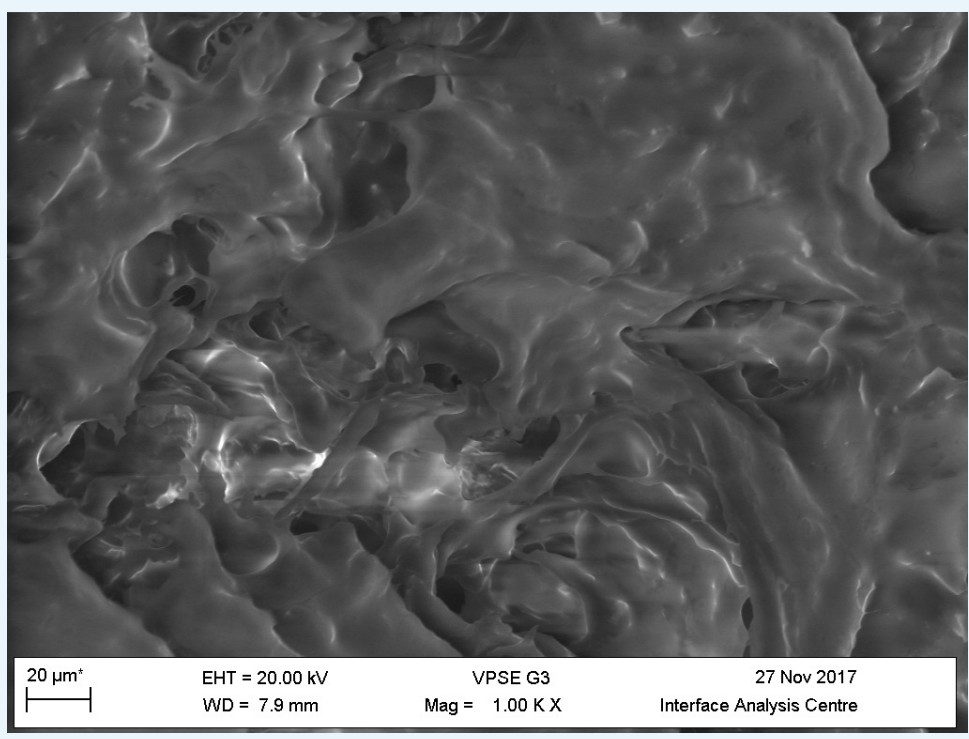

**Appendix 1—figure 30.** Demineralised modern chicken bone viewed close up (image 1).

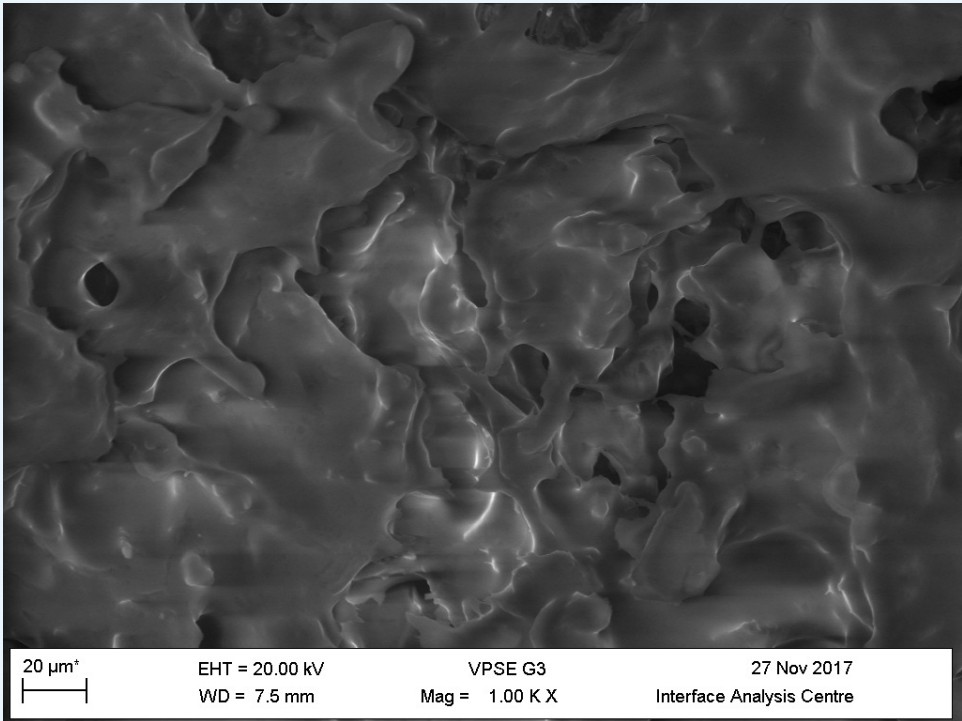

**Appendix 1—figure 31.** Demineralised modern chicken bone viewed close up (image 2).
## Demineralised Pleistocene-Holocene shark tooth

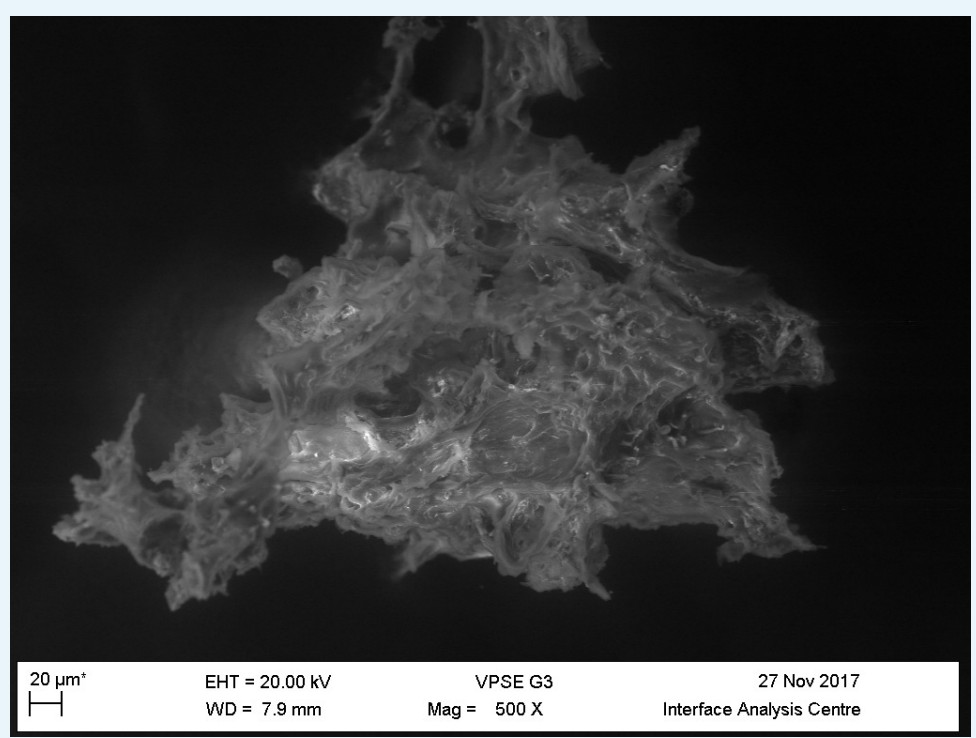

| 20 µm* | EHT = 20.00 kV | VPSE G3 | 27 Nov 2017 |
|---|---|---|---|
| | WD = 7.9 mm | Mag = 500 X | Interface Analysis Centre |

**Appendix 1—figure 32.** Demineralised Pleistocene-Holocene shark tooth (image 1).

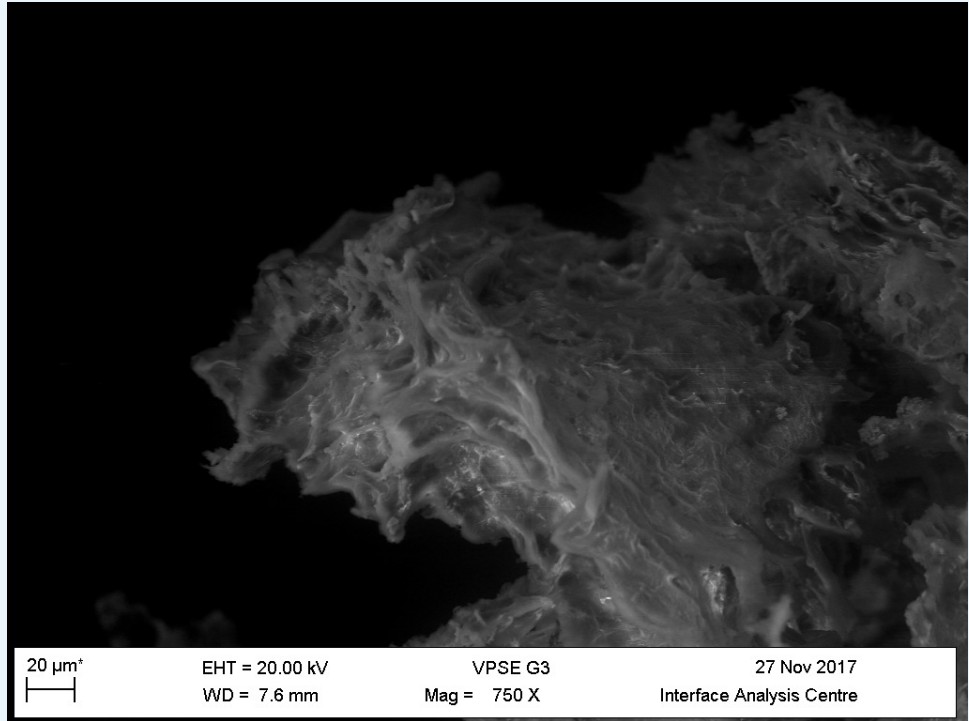

| 20 µm* | EHT = 20.00 kV | VPSE G3 | 27 Nov 2017 |
|---|---|---|---|
| | WD = 7.6 mm | Mag = 750 X | Interface Analysis Centre |

**Appendix 1—figure 33.** Demineralised Pleistocene-Holocene shark tooth (image 2).

## Demineralised *Centrosaurus* bone

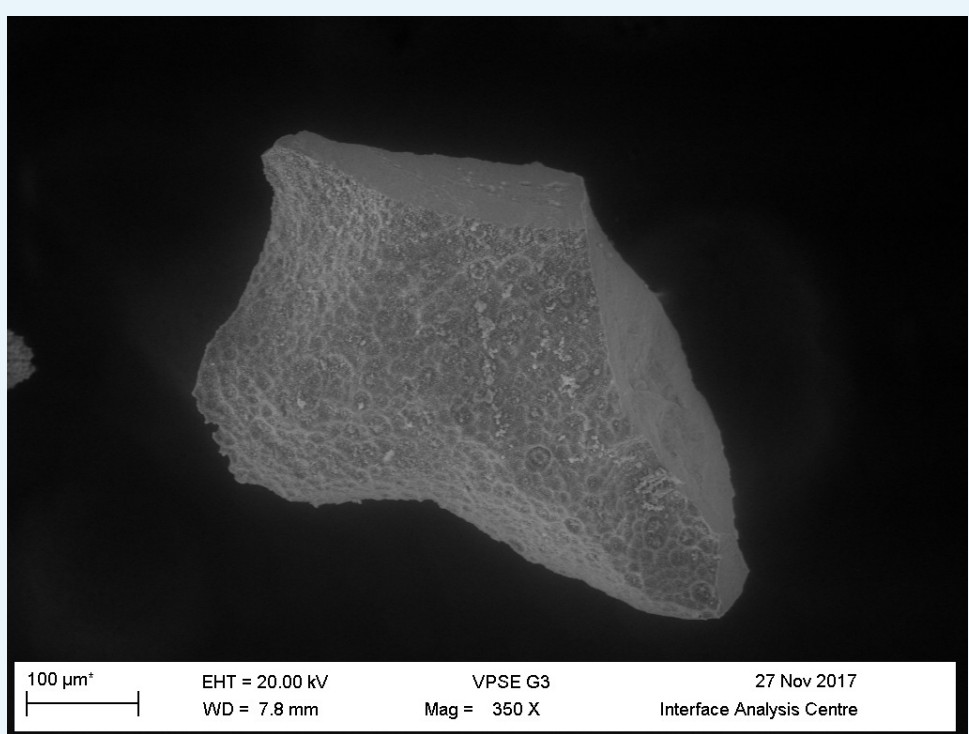

| 100 µm* | EHT = 20.00 kV | VPSE G3 | 27 Nov 2017 |
|---|---|---|---|
| | WD = 7.8 mm | Mag = 350 X | Interface Analysis Centre |

**Appendix 1—figure 34.** Mineral grain from demineralised Late Cretaceous Centrosaurus bone.

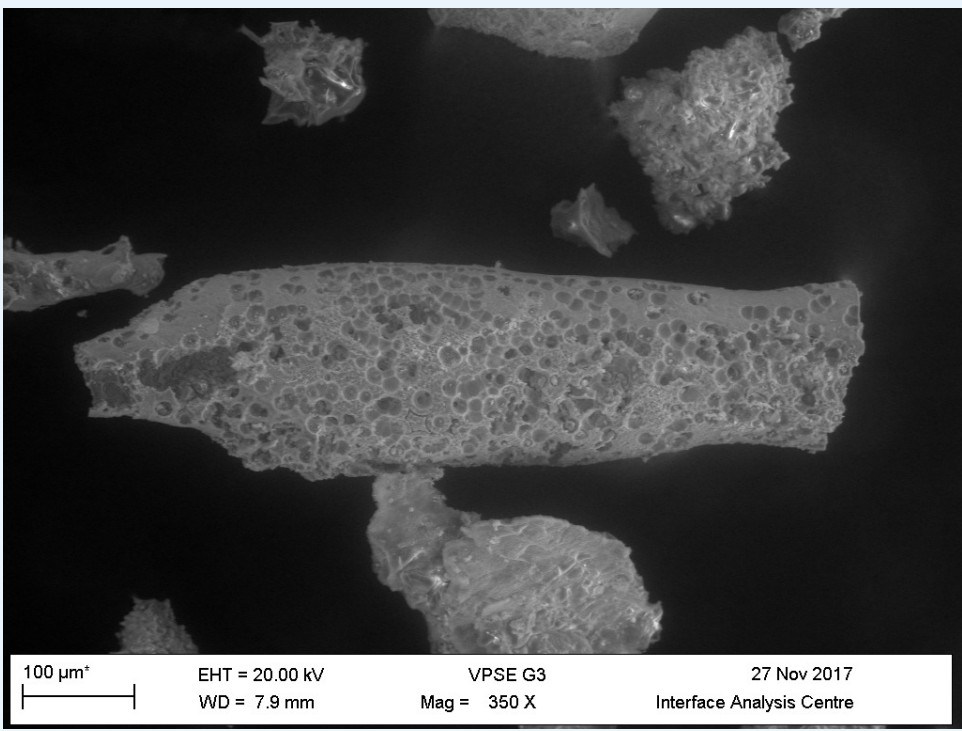

| 100 µm* | EHT = 20.00 kV | VPSE G3 | 27 Nov 2017 |
|---|---|---|---|
| | WD = 7.9 mm | Mag = 350 X | Interface Analysis Centre |

**Appendix 1—figure 35.** Vessel from demineralised Late Cretaceous Centrosaurus bone (image 1).

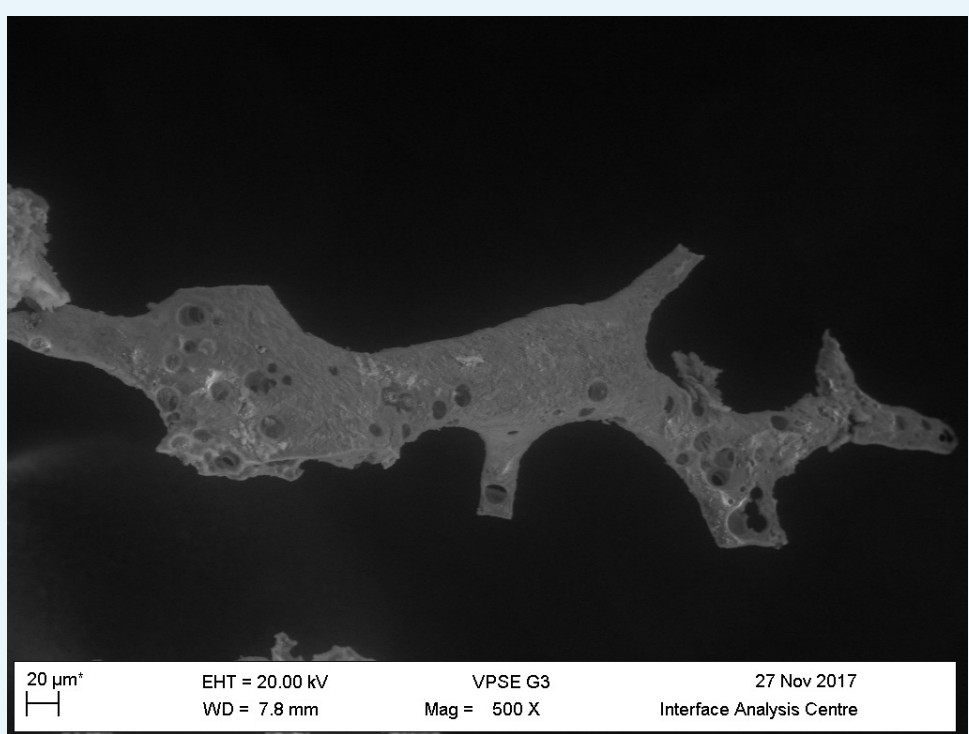

| 20 μm* | EHT = 20.00 kV | VPSE G3 | 27 Nov 2017 |
| | WD = 7.8 mm | Mag = 500 X | Interface Analysis Centre |

**Appendix 1—figure 36.** Vessel from demineralised Late Cretaceous Centrosaurus bone (image 2).

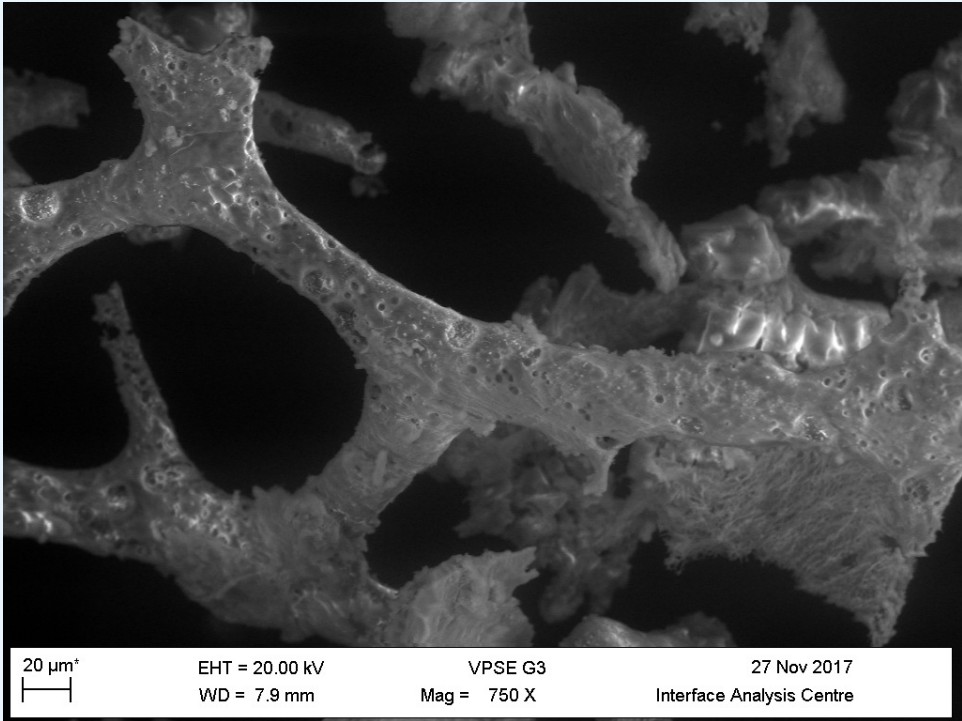

| 20 μm* | EHT = 20.00 kV | VPSE G3 | 27 Nov 2017 |
| | WD = 7.9 mm | Mag = 750 X | Interface Analysis Centre |

**Appendix 1—figure 37.** Vessel from demineralised Late Cretaceous Centrosaurus bone (image 3).

## EDS

CSV files containing raw spectral data of demineralised samples are available through the Field Museum's collections database: multimedia record containing raw files (GUID: 60c79cec-4da1-4bea-8535-a332c70ae4c9, URI: https://mm.fieldmuseum.org/60c79cec-4da1-4bea-8535-a332c70ae4c9), event record with surrounding information about the project (GUID: 34e15532-2c46-47cf-aac0-5d29cc5a2c22, URI: https://pj.fieldmuseum.org/event/34e15532-2c46-47cf-aac0-5d29cc5a2c22) and *Source data 1*.

EDS eZAF Smart Quant: Error % – error on the previous % values; K ratio – how much of this quantification is done on the K-shell emission; Z, R, A, F – all correspond to the peak fitting that the software performs (ZAF and eZAF are the common peak fitting/quantification algorithms).

EDS spectra: KV – accelerating voltage; Mag – magnification; Take-off – angle of detector from the horizontal (a specific of the system); Live Time(s) – how long we collected a spectrum for (in seconds); Amp Time(µs) – system dead-time for processing the previous incident x-rays.

### Demineralised modern chicken bone

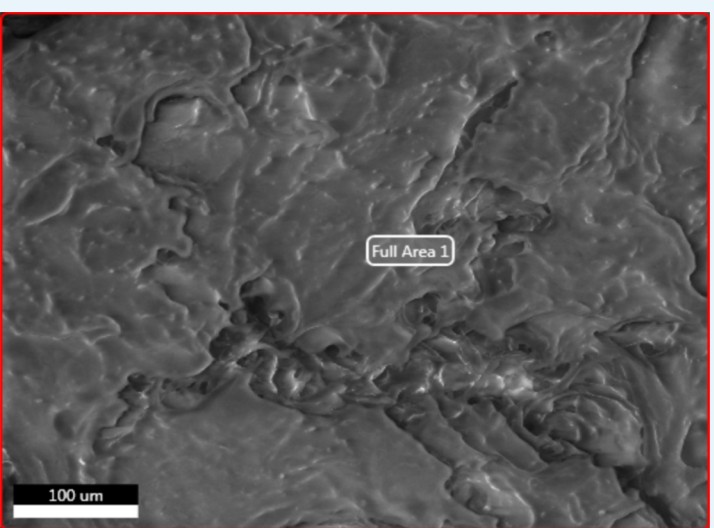

**Appendix 1—figure 38.** Electron image of modern demineralised chicken bone full area 1 EDS analysis.

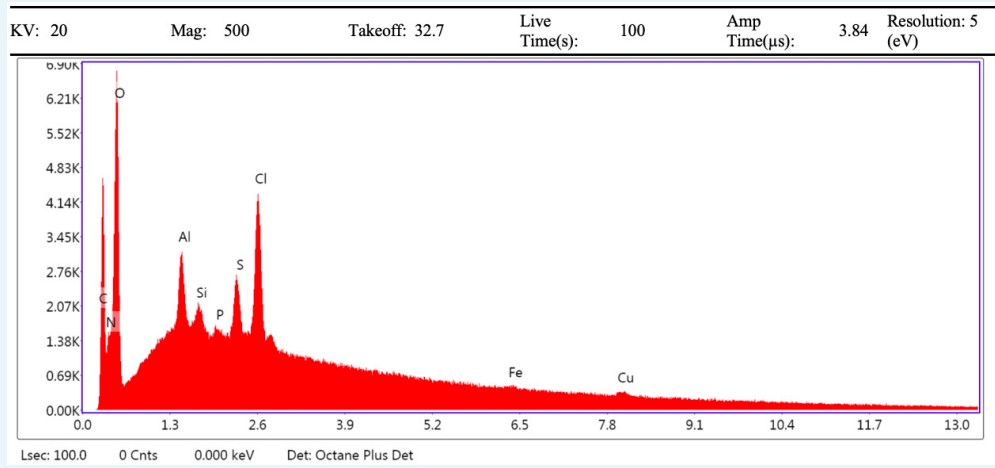

**Appendix 1—figure 39.** EDS spectrum of demineralised modern chicken bone full area 1.

**Appendix 1—table 14.** EDS eZAF Smart Quant results of demineralised modern chicken bone full area 1.

| Element | Weight % | Atomic % | Net int. | Error % | Kratio | Z | R | A | F |
|---|---|---|---|---|---|---|---|---|---|
| C K | 21.88 | 27.49 | 191.17 | 8.98 | 0.0666 | 1.0413 | 0.9803 | 0.2921 | 1.0000 |
| N K | 24.85 | 26.76 | 138.36 | 10.57 | 0.0451 | 1.0169 | 0.9906 | 0.1786 | 1.0000 |
| O K | 44.17 | 41.65 | 471.57 | 9.96 | 0.0679 | 0.9957 | 0.9999 | 0.1543 | 1.0000 |
| AlK | 1.77 | 0.99 | 134.20 | 7.25 | 0.0103 | 0.8846 | 1.0363 | 0.6545 | 1.0066 |
| SiK | 0.61 | 0.33 | 54.63 | 9.51 | 0.0043 | 0.9038 | 1.0422 | 0.7658 | 1.0100 |
| P K | 0.21 | 0.10 | 16.93 | 23.70 | 0.0016 | 0.8679 | 1.0477 | 0.8576 | 1.0160 |
| S K | 1.53 | 0.72 | 132.12 | 4.25 | 0.0127 | 0.8848 | 1.0529 | 0.9237 | 1.0199 |
| ClK | 4.10 | 1.74 | 322.09 | 2.47 | 0.0334 | 0.8416 | 1.0578 | 0.9553 | 1.0138 |
| FeK | 0.24 | 0.07 | 7.47 | 55.87 | 0.0022 | 0.7593 | 1.0862 | 1.0212 | 1.1884 |
| CuK | 0.63 | 0.15 | 12.85 | 30.91 | 0.0061 | 0.7264 | 1.0858 | 1.0175 | 1.2983 |

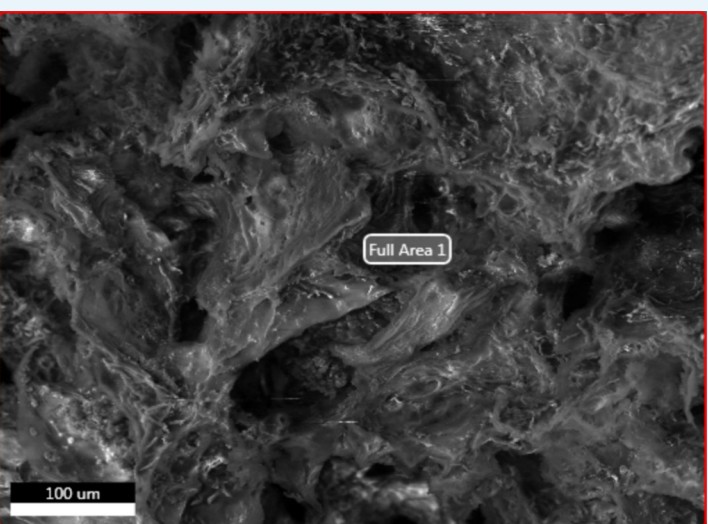

**Appendix 1—figure 40.** Electron image of demineralised Pleistocene-Holocene shark tooth full area 1 EDS analysis.

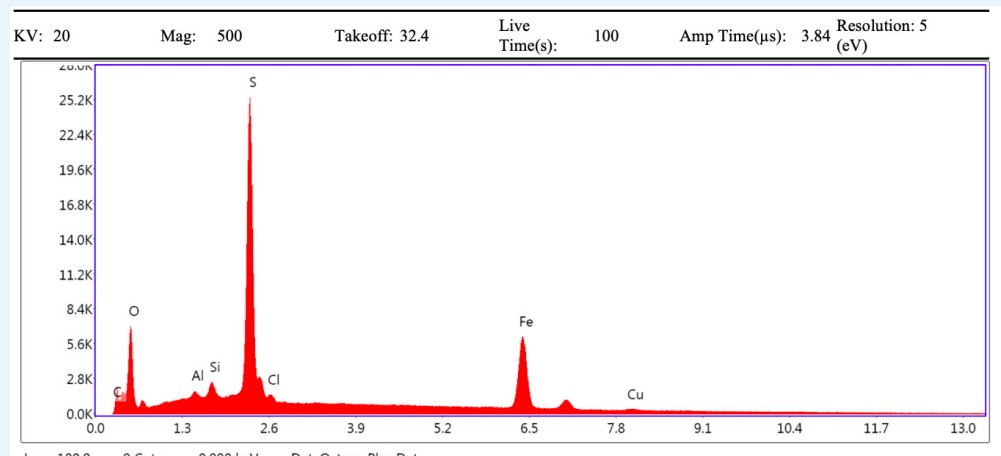

**Appendix 1—figure 41.** EDS spectrum of demineralised Pleistocene-Holocene shark tooth full area 1.

**Appendix 1—table 15.** EDS eZAF Smart Quant results of demineralised Pleistocene-Holocene shark tooth full area 1.

| Element | Weight % | Atomic % | Net int. | Error % | Kratio | Z | R | A | F |
|---|---|---|---|---|---|---|---|---|---|
| C K | 25.65 | 42.51 | 150.16 | 10.58 | 0.0401 | 1.1074 | 0.9367 | 0.1411 | 1.0000 |
| O K | 27.79 | 34.58 | 542.95 | 9.50 | 0.0597 | 1.0620 | 0.9592 | 0.2024 | 1.0000 |
| AlK | 0.39 | 0.29 | 35.97 | 12.82 | 0.0021 | 0.9479 | 1.0018 | 0.5587 | 1.0102 |
| SiK | 0.88 | 0.62 | 99.39 | 7.89 | 0.0060 | 0.9691 | 1.0088 | 0.6901 | 1.0169 |
| S K | 21.63 | 13.43 | 2464.07 | 2.58 | 0.1807 | 0.9499 | 1.0217 | 0.8695 | 1.0112 |
| ClK | 0.76 | 0.43 | 68.59 | 10.21 | 0.0054 | 0.9039 | 1.0277 | 0.7737 | 1.0166 |
| FeK | 22.45 | 8.00 | 847.32 | 2.19 | 0.1927 | 0.8189 | 1.0658 | 1.0039 | 1.0442 |
| CuK | 0.44 | 0.14 | 10.24 | 41.09 | 0.0037 | 0.7847 | 1.0692 | 0.9849 | 1.0840 |

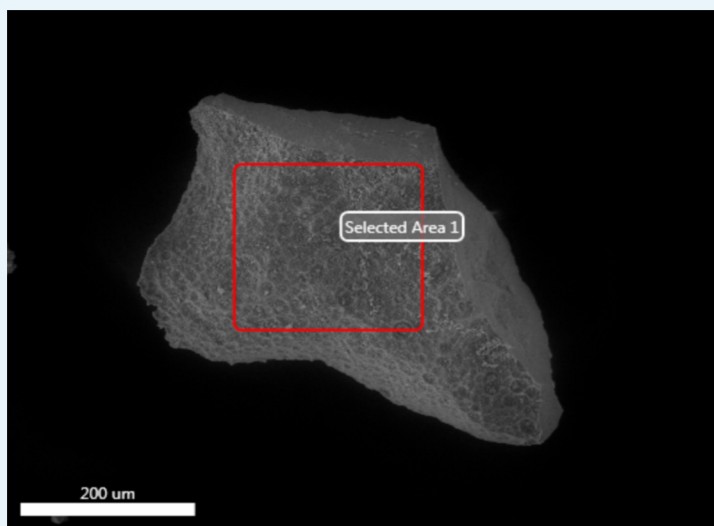

**Appendix 1—figure 42.** Electron image of mineral grain from demineralised Late Cretaceous Centrosaurus bone area A.

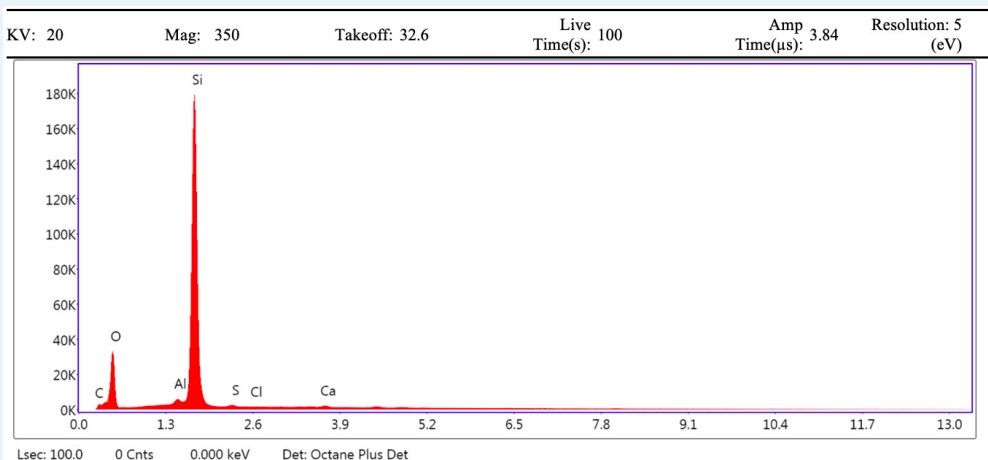

**Appendix 1—figure 43.** EDS spectrum of mineral grain from demineralised Late Cretaceous Centrosaurus bone area A.

**Appendix 1—table 16.** EDS eZAF Smart Quant results of mineral grain from demineralised Late Cretaceous *Centrosaurus* bone area A.

| Element | Weight % | Atomic % | Net int. | Error % | Kratio | Z | R | A | F |
|---|---|---|---|---|---|---|---|---|---|
| C K | 8.68 | 14.27 | 101.93 | 99.99 | 0.0107 | 1.0874 | 0.9545 | 0.1133 | 1.0000 |
| O K | 40.70 | 50.27 | 2568.99 | 8.50 | 0.1113 | 1.0415 | 0.9759 | 0.2625 | 1.0000 |
| AlK | 1.06 | 0.78 | 326.72 | 5.09 | 0.0075 | 0.9274 | 1.0163 | 0.7418 | 1.0360 |
| SiK | 48.46 | 34.09 | 16392.38 | 2.65 | 0.3872 | 0.9479 | 1.0229 | 0.8410 | 1.0025 |
| S K | 0.40 | 0.25 | 80.67 | 9.76 | 0.0023 | 0.9284 | 1.0350 | 0.6279 | 1.0056 |
| ClK | 0.01 | 0.01 | 2.31 | 60.70 | 0.0001 | 0.8832 | 1.0405 | 0.7259 | 1.0084 |
| CaK | 0.69 | 0.34 | 125.27 | 8.44 | 0.0057 | 0.8950 | 1.0552 | 0.9103 | 1.0202 |

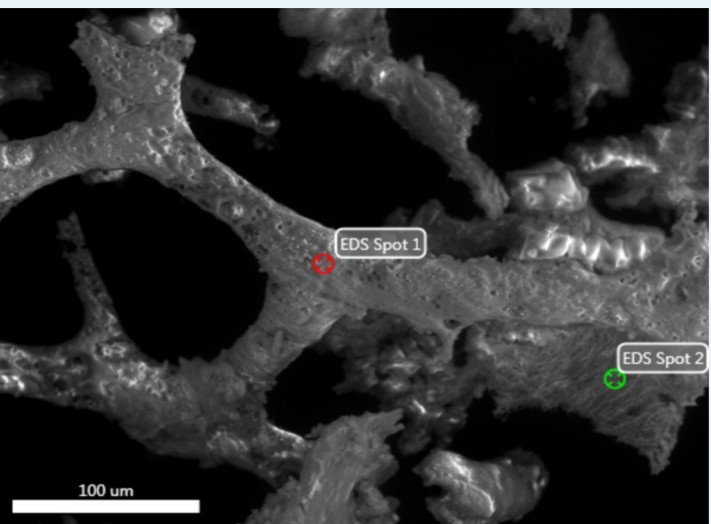

**Appendix 1—figure 44.** Electron image of vessel (spot 1) and fibrous mass (spot 2) from demineralised Late Cretaceous Centrosaurus bone area B.

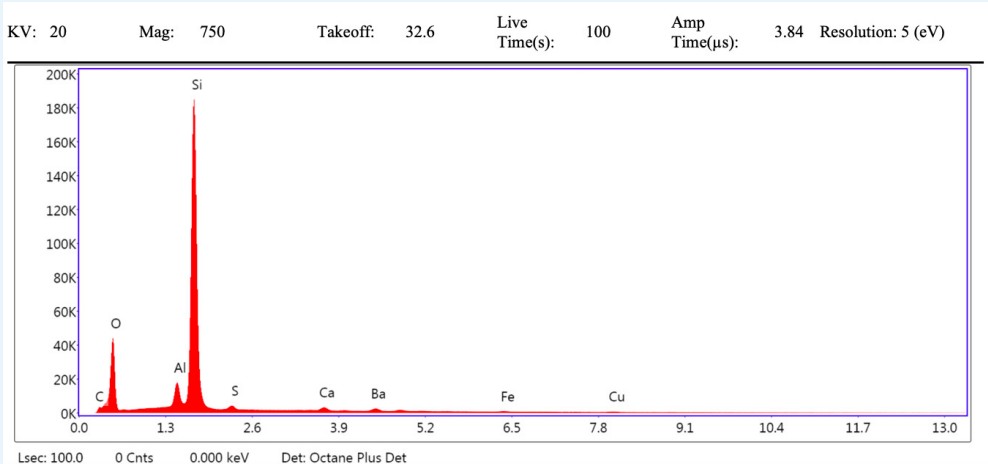

**Appendix 1—figure 45.** EDS spectrum of vessel (spot 1) from demineralised Late Cretaceous Centrosaurus bone area B.

**Appendix 1—table 17.** EDS eZAF Smart Quant results of vessel (spot 1) from demineralised Late Cretaceous *Centrosaurus* bone area B.

| Element | Weight % | Atomic % | Net int. | Error % | Kratio | Z | R | A | F |
|---|---|---|---|---|---|---|---|---|---|
| C K | 5.61 | 9.66 | 81.05 | 99.99 | 0.0071 | 1.1005 | 0.9464 | 0.1152 | 1.0000 |
| O K | 40.95 | 52.94 | 3343.32 | 8.29 | 0.1212 | 1.0546 | 0.9683 | 0.2807 | 1.0000 |
| AlK | 3.75 | 2.88 | 1306.45 | 4.48 | 0.0253 | 0.9400 | 1.0097 | 0.6987 | 1.0256 |
| SiK | 44.51 | 32.78 | 16780.97 | 3.33 | 0.3320 | 0.9609 | 1.0165 | 0.7739 | 1.0031 |
| S K | 0.87 | 0.56 | 212.94 | 7.26 | 0.0052 | 0.9414 | 1.0290 | 0.6248 | 1.0067 |
| CaK | 0.96 | 0.50 | 211.92 | 6.87 | 0.0081 | 0.9079 | 1.0501 | 0.9059 | 1.0253 |
| BaL | 2.46 | 0.37 | 177.23 | 8.15 | 0.0184 | 0.6537 | 1.2526 | 1.0822 | 1.0599 |
| FeK | 0.40 | 0.15 | 46.67 | 13.64 | 0.0035 | 0.8100 | 1.0708 | 0.9944 | 1.0873 |
| CuK | 0.49 | 0.16 | 36.93 | 17.79 | 0.0044 | 0.7756 | 1.0733 | 1.0044 | 1.1609 |

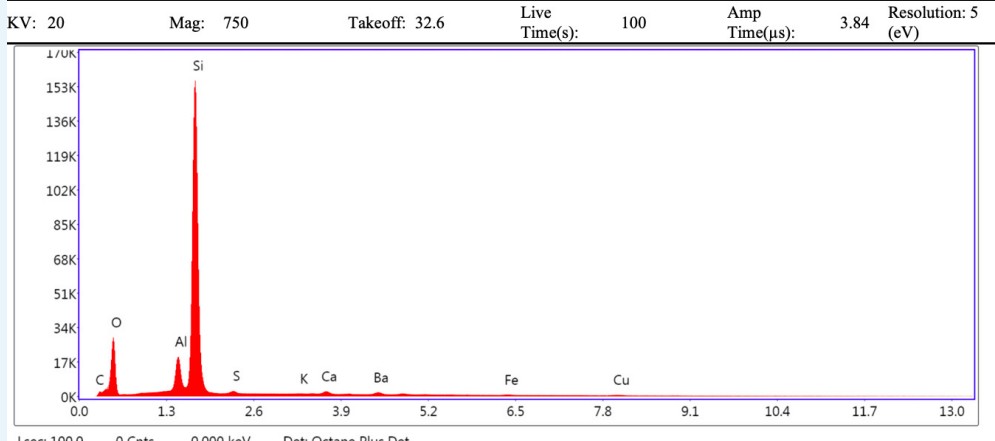

**Appendix 1—figure 46.** EDS spectrum of fibrous mass (spot 2) from demineralised Late Cretaceous. Centrosaurus bone area B.

**Appendix 1—table 18.** EDS eZAF Smart Quant results of fibrous mass (spot 2) from demineralised Late Cretaceous *Centrosaurus* bone area B.

| Element | Weight % | Atomic % | Net int. | Error % | Kratio | Z | R | A | F |
|---|---|---|---|---|---|---|---|---|---|
| C K | 5.94 | 10.38 | 62.73 | 99.99 | 0.0070 | 1.1021 | 0.9464 | 0.1076 | 1.0000 |
| O K | 36.53 | 47.94 | 2200.10 | 8.49 | 0.1021 | 1.0562 | 0.9684 | 0.2646 | 1.0000 |
| AlK | 5.47 | 4.26 | 1553.47 | 4.17 | 0.0385 | 0.9413 | 1.0098 | 0.7276 | 1.0272 |
| SiK | 48.16 | 36.01 | 14261.52 | 3.30 | 0.3611 | 0.9622 | 1.0166 | 0.7771 | 1.0027 |
| S K | 0.57 | 0.37 | 105.79 | 8.71 | 0.0033 | 0.9426 | 1.0291 | 0.6074 | 1.0061 |
| K K | 0.06 | 0.03 | 12.05 | 55.96 | 0.0005 | 0.8927 | 1.0453 | 0.8536 | 1.0188 |
| CaK | 0.79 | 0.42 | 135.54 | 6.59 | 0.0066 | 0.9091 | 1.0501 | 0.8985 | 1.0229 |
| BaL | 1.43 | 0.22 | 80.41 | 12.01 | 0.0107 | 0.6546 | 1.2527 | 1.0784 | 1.0639 |
| FeK | 0.34 | 0.13 | 31.14 | 16.61 | 0.0030 | 0.8109 | 1.0708 | 0.9951 | 1.0943 |
| CuK | 0.71 | 0.24 | 42.62 | 16.37 | 0.0065 | 0.7765 | 1.0733 | 1.0049 | 1.1673 |

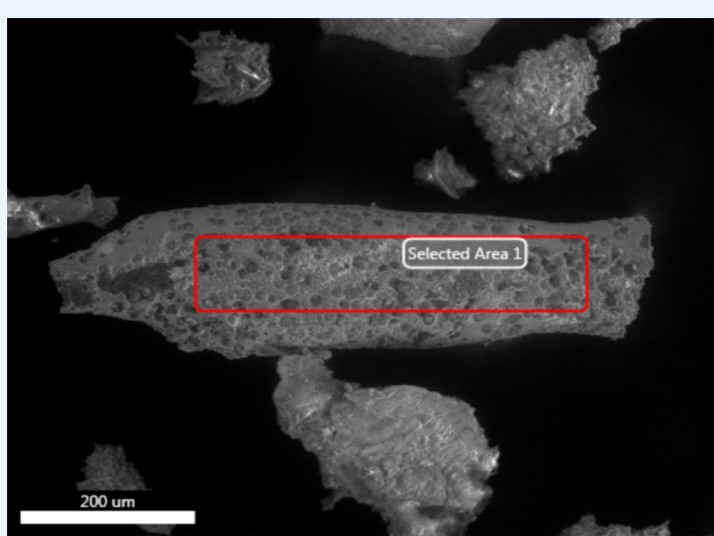

**Appendix 1—figure 47.** Electron image of another vessel from demineralised Late Cretaceous Centrosaurus bone area C.

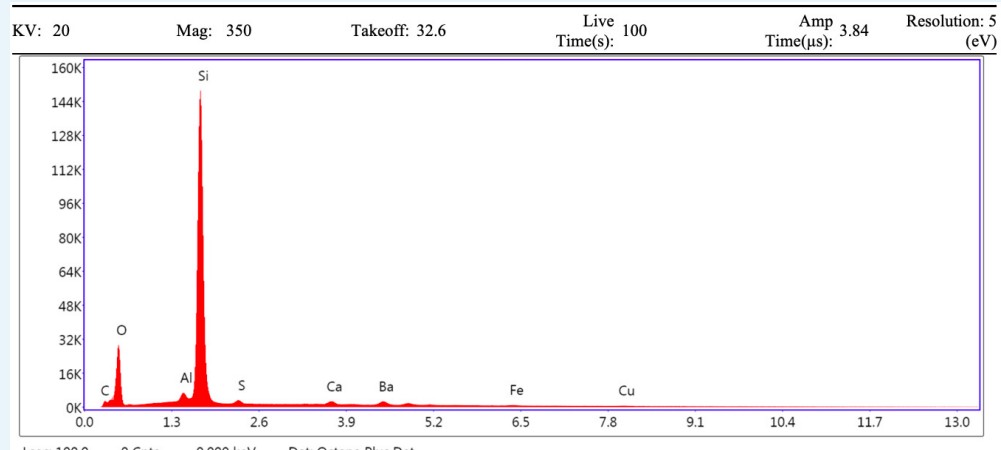

**Appendix 1—figure 48.** EDS spectrum of vessel from demineralised Late Cretaceous Centrosaurus bone area C.

**Appendix 1—table 19.** EDS eZAF Smart Quant results of vessel from demineralised Late Cretaceous *Centrosaurus* bone area C.

| Element | Weight % | Atomic % | Net int. | Error % | Kratio | Z | R | A | F |
|---------|----------|----------|----------|---------|--------|------|------|------|------|
| C K | 7.78 | 13.45 | 85.90 | 99.99 | 0.0100 | 1.1029 | 0.9436 | 0.1160 | 1.0000 |
| O K | 37.94 | 49.22 | 2221.65 | 8.49 | 0.1064 | 1.0572 | 0.9657 | 0.2653 | 1.0000 |
| AlK | 1.56 | 1.20 | 407.62 | 5.19 | 0.0104 | 0.9426 | 1.0074 | 0.6911 | 1.0273 |
| SiK | 46.30 | 34.22 | 13551.97 | 3.16 | 0.3543 | 0.9636 | 1.0143 | 0.7913 | 1.0032 |
| S K | 0.84 | 0.54 | 154.40 | 7.87 | 0.0049 | 0.9442 | 1.0269 | 0.6217 | 1.0071 |
| CaK | 1.01 | 0.52 | 169.40 | 6.80 | 0.0086 | 0.9108 | 1.0482 | 0.9041 | 1.0273 |
| BaL | 3.73 | 0.56 | 203.06 | 7.22 | 0.0279 | 0.6559 | 1.2507 | 1.0806 | 1.0545 |
| FeK | 0.41 | 0.15 | 35.85 | 17.08 | 0.0036 | 0.8128 | 1.0694 | 0.9912 | 1.0797 |
| CuK | 0.43 | 0.14 | 24.38 | 21.06 | 0.0038 | 0.7784 | 1.0721 | 1.0027 | 1.1492 |

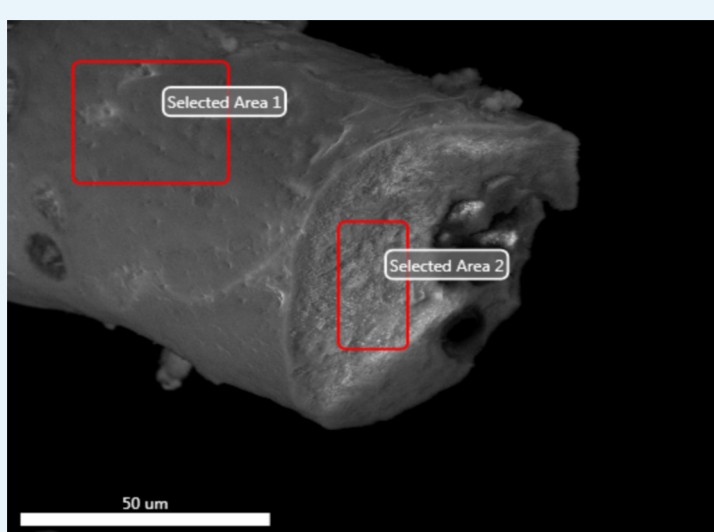

**Appendix 1—figure 49.** Electron image of a different vessel exterior (region 1) and interior (region 2) from demineralised Late Cretaceous Centrosaurus bone area D.

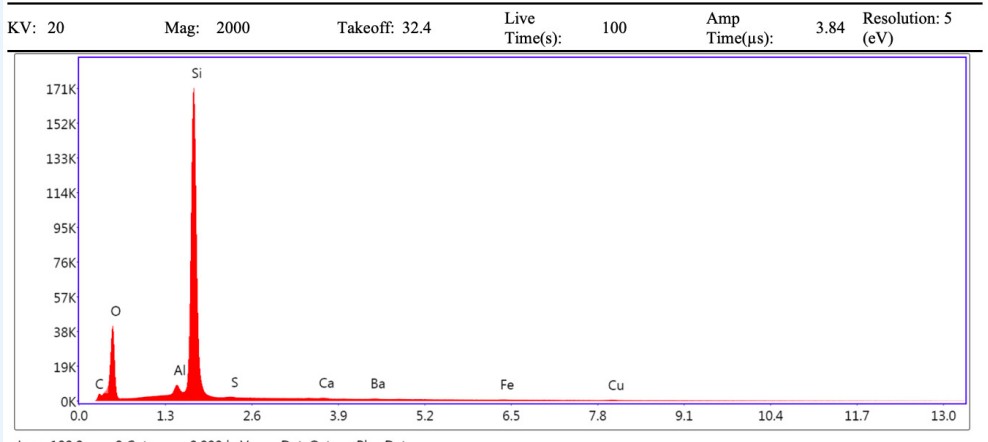

**Appendix 1—figure 50.** EDS spectrum of vessel exterior (region 1) from demineralised Late Cretaceous Centrosaurus bone area D.

**Appendix 1—table 20.** EDS eZAF Smart Quant results of vessel exterior (region 1) from demineralised Late Cretaceous *Centrosaurus* bone area D.

| Element | Weight % | Atomic % | Net int. | Error % | Kratio | Z | R | A | F |
|---------|----------|----------|----------|---------|--------|--------|--------|--------|--------|
| C K | 10.99 | 17.49 | 159.50 | 99.99 | 0.0148 | 1.0832 | 0.9565 | 0.1245 | 1.0000 |
| O K | 43.55 | 52.05 | 3194.43 | 8.39 | 0.1224 | 1.0374 | 0.9778 | 0.2708 | 1.0000 |
| AlK | 1.62 | 1.15 | 541.46 | 4.85 | 0.0110 | 0.9237 | 1.0178 | 0.7166 | 1.0296 |
| SiK | 42.37 | 28.85 | 15672.00 | 2.92 | 0.3265 | 0.9440 | 1.0244 | 0.8142 | 1.0026 |
| S K | 0.16 | 0.10 | 38.75 | 15.68 | 0.0010 | 0.9246 | 1.0364 | 0.6496 | 1.0060 |
| CaK | 0.31 | 0.15 | 65.74 | 9.96 | 0.0026 | 0.8914 | 1.0564 | 0.9211 | 1.0232 |
| BaL | 0.53 | 0.07 | 36.74 | 27.53 | 0.0040 | 0.6416 | 1.2593 | 1.0954 | 1.0750 |
| FeK | 0.17 | 0.06 | 19.37 | 24.43 | 0.0015 | 0.7947 | 1.0756 | 1.0031 | 1.1088 |
| CuK | 0.29 | 0.09 | 21.53 | 21.86 | 0.0027 | 0.7607 | 1.0773 | 1.0090 | 1.2037 |

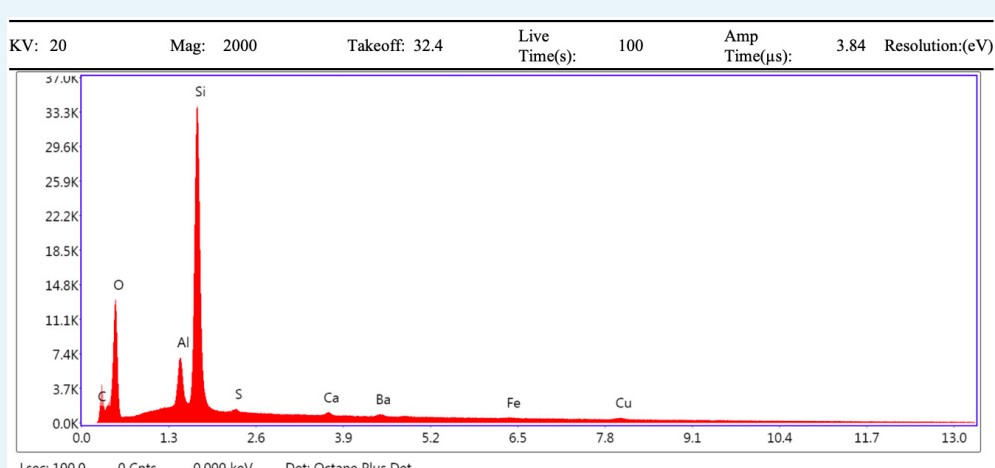

| KV: 20 | Mag: 2000 | Takeoff: 32.4 | Live Time(s): | 100 | Amp Time(μs): | 3.84 | Resolution:(eV) |

Lsec: 100.0   0 Cnts   0.000 keV   Det: Octane Plus Det

**Appendix 1—figure 51.** EDS spectrum of vessel interior (region 2) from demineralised Late Cretaceous Centrosaurus bone area D.

**Appendix 1—table 21.** EDS eZAF Smart Quant results of vessel interior (region 2) from demineralised Late Cretaceous *Centrosaurus* bone area D.

| Element | Weight % | Atomic % | Net int. | Error % | Kratio | Z | R | A | F |
|---------|----------|----------|----------|---------|--------|------|------|------|------|
| C K | 25.09 | 35.59 | 183.44 | 99.99 | 0.0467 | 1.0660 | 0.9650 | 0.1747 | 1.0000 |
| O K | 43.49 | 46.31 | 999.60 | 8.89 | 0.1049 | 1.0204 | 0.9857 | 0.2365 | 1.0000 |
| AlK | 4.18 | 2.64 | 475.03 | 4.96 | 0.0265 | 0.9080 | 1.0246 | 0.6878 | 1.0165 |
| SiK | 24.25 | 14.71 | 3016.29 | 3.57 | 0.1723 | 0.9279 | 1.0309 | 0.7630 | 1.0034 |
| S K | 0.17 | 0.09 | 16.84 | 23.85 | 0.0012 | 0.9087 | 1.0424 | 0.7368 | 1.0077 |
| CaK | 0.43 | 0.18 | 33.69 | 13.24 | 0.0037 | 0.8760 | 1.0616 | 0.9572 | 1.0302 |
| BaL | 1.18 | 0.15 | 29.89 | 27.69 | 0.0090 | 0.6305 | 1.2647 | 1.1172 | 1.0818 |
| FeK | 0.19 | 0.06 | 7.87 | 55.09 | 0.0017 | 0.7809 | 1.0796 | 1.0094 | 1.1220 |
| CuK | 1.01 | 0.27 | 26.63 | 17.38 | 0.0091 | 0.7473 | 1.0805 | 1.0120 | 1.1953 |

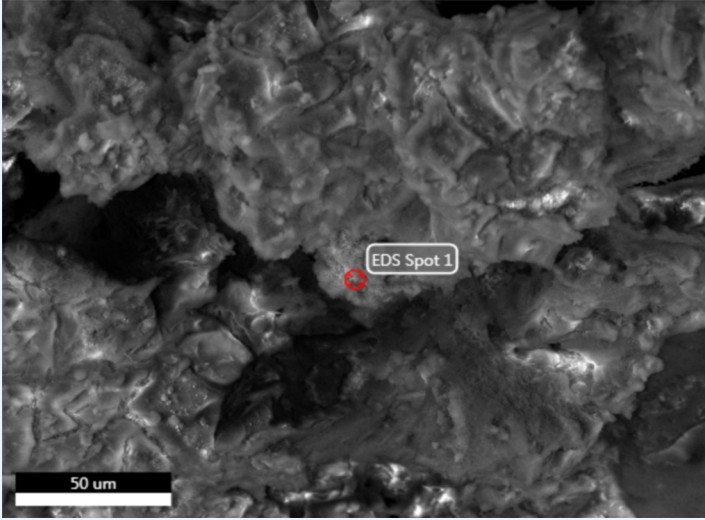

**Appendix 1—figure 52.** Electron image of mineral grain from demineralised Late Cretaceous Centrosaurus bone area E with spot one analysis shown.

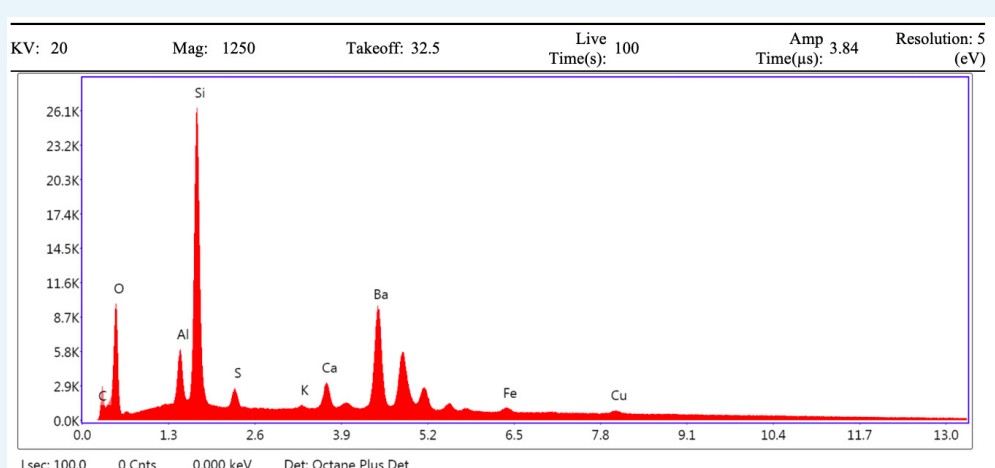

**Appendix 1—figure 53.** EDS spectrum of mineral grain from demineralised Late Cretaceous Centrosaurus bone area E, spot 1.

**Appendix 1—table 22.** EDS eZAF Smart Quant results of mineral grain from demineralised Late Cretaceous *Centrosaurus* bone area E, spot 1.

| Element | Weight % | Atomic % | Net int. | Error % | Kratio | Z | R | A | F |
|---------|----------|----------|----------|---------|--------|------|------|------|------|
| C K | 8.12 | 19.85 | 76.35 | 99.99 | 0.0181 | 1.2324 | 0.8623 | 0.1813 | 1.0000 |
| O K | 21.32 | 39.16 | 732.39 | 8.55 | 0.0718 | 1.1857 | 0.8869 | 0.2838 | 1.0000 |
| AlK | 4.94 | 5.38 | 431.02 | 7.60 | 0.0225 | 1.0649 | 0.9356 | 0.4237 | 1.0087 |
| SiK | 21.74 | 22.74 | 2311.26 | 6.07 | 0.1233 | 1.0899 | 0.9440 | 0.5169 | 1.0068 |
| S K | 1.45 | 1.33 | 146.92 | 8.19 | 0.0096 | 1.0705 | 0.9597 | 0.6086 | 1.0162 |
| K K | 0.27 | 0.20 | 27.82 | 18.41 | 0.0024 | 1.0174 | 0.9810 | 0.8363 | 1.0548 |
| CaK | 2.62 | 1.92 | 249.09 | 4.58 | 0.0257 | 1.0373 | 0.9875 | 0.8814 | 1.0698 |
| BaL | 36.01 | 7.70 | 1044.05 | 2.70 | 0.2929 | 0.7495 | 1.1854 | 1.0526 | 1.0312 |
| FeK | 1.34 | 0.71 | 59.05 | 10.93 | 0.0120 | 0.9325 | 1.0202 | 0.9145 | 1.0442 |
| CuK | 2.19 | 1.01 | 63.31 | 7.26 | 0.0203 | 0.8976 | 1.0306 | 0.9591 | 1.0793 |

## Carbon analysis

Calculating the relative contribution of C to subterranean Late Cretaceous bone using a simple 2-end-member mixing model was as follows:

$$F = M(x) + D(1 - x)$$

where $F$ = fossil bone $F^{14}C$,

$M$ = mudstone matrix $F^{14}C$,

and $D$ = endogenous dinosaur $F^{14}C = 0$.

**Appendix 1—table 23.** Non-demineralised Dinosaur Provincial Park samples.

| Sample | Mass analysed (mg) | C % |
|--------|-------------------|-----|
| Matrix-surrounded subterranean bone (not scrapped) | 3.219 | 2.3 |
| Adjacent mudstone matrix of subterranean bone | 2.112 | 1.11 |

*Appendix 1—table 23 continued on next page*

*Appendix 1—table 23 continued*

| Sample | Mass analysed (mg) | C % |
|---|---|---|
| BB180 surface bone (not scrapped) | 3.184 | 4.19 |
| Topsoil | 2.235 | 1.26 |
| Mudstone 693 m elevation | 3.413 | 1.15 |

**Appendix 1—table 24.** Pilot tests on demineralised Dinosaur Provincial Park samples (0.5 M HCl).

| Sample | Mass (mg) bulk powder before demineralisation | Mass (mg) after demineralisation | % mass surviving demineralisation | Mass (mg) analysed | C % | % change in [C] from bulk powder |
|---|---|---|---|---|---|---|
| Matrix-surrounded sub-terranean bone (not scrapped) | 299.7 | 7.536 | 2.514514515 | 4.035 | 13.48 | 486.0869565 |
| Adjacent mudstone matrix of subterranean bone | 308.5 | 253.8 | 82.26904376 | 3.499 | 1.28 | 15.31531532 |
| BB180 surface bone (not scrapped) | 307.1 | 71.99 | 23.44187561 | 10.051 | 0.13 | −96.8973747 |
| Topsoil | 307.2 | 281.5 | 91.63411458 | 6.21 | 1.43 | 13.49206349 |
| Mudstone 693 m elevation | 300.4 | 271.5 | 90.37949401 | 5.005 | 0.92 | −20 |
| Surface bone 691 m elevation (core) | 341 | 148 | 43.40175953 | 7.903 | 1.47 | NA |

# Quebit fluorometry

**Appendix 1—table 25. Quebit fluorometer test results on Dinosaur Provincial Park samples.** Sample ID's as in *Appendix 1—table 2*.

| Bag number | Type | Replicate | Sample ID | Kit | Ng of DNA / µL First read | Ng of DNA / µL Second read | Concentration | Concentrated ng of DNA / µL First read | Concentrated ng of DNA / µL Second read |
|---|---|---|---|---|---|---|---|---|---|
| NA | Blank | NA | Blank | Power Viral | Below detection | Below detection | | | |
| 10 | Topsoil | 1 | 10 | Power Viral | 0.151 | 0.133 | | | |
| 13 | Mudstone | 1 | 13 | Power Viral | Below detection | Below detection | | | |
| 16 | Scrappings | 1 | 16S | Power Viral | 0.172 | 0.164 | | | |
| 16 | Bone | 1 | 16B1 | Power Viral | 0.424 | 0.404 | | | |
| 16 | Bone | 2 | 16B2 | Power Viral | 0.592 | 0.55 | | | |
| NA | Blank | NA | Blank | Power Viral | Below detection | Below detection | | | |
| 1 | 'Float' bone in matrix | 1 | 1F1 | Power Viral | 0.0926 | 0.0908 | | | |
| 1 | Scrappings | 1 | 1S1 | Power Viral | 0.128 | 0.127 | | | |
| 1 | Scrappings | 2 | 1S2 | Power Viral | 0.0238 | 0.0236 | | | |

*Appendix 1—table 25 continued on next page*

*Appendix 1—table 25 continued*

| Bag number | Type | Replicate | Sample ID | Kit | Ng of DNA / μL | | Concentration | Concentrated ng of DNA / μL | |
|---|---|---|---|---|---|---|---|---|---|
| | | | | | First read | Second read | | First read | Second read |
| 1 | Bone | 1 | 1B1 | Power Viral | 0.0382 | 0.0376 | | | |
| 1 | Bone | 2 | 1B2 | Power Viral | 0.0544 | 0.0546 | | | |
| 1 | Mudstone | 1 | 1M1 | Power Viral | Below detection | Below detection | | | |
| 1 | Mudstone | 2 | 1M2 | Power Viral | Below detection | Below detection | | | |
| NA | Blank | NA | Blank | Dneasy PowerMax Soil | Below detection | Below detection | x 25 | Below detection | Below detection |
| 1 | Bone | 3 | 1B5g | Dneasy PowerMax Soil | 0.798 | 0.788 | x 25 | 11.1 | 10.5 |
| 1 | Mudstone | 3 | 1M10g1 | Dneasy PowerMax Soil | 0.0334 | 0.0322 | x 25 | 0.626 | 0.612 |
| 1 | Mudstone | 4 | 1M10g2 | Dneasy PowerMax Soil | 0.0624 | 0.0596 | x 25 | 0.586 | 0.586 |
| 8 | Mudstone | 1 | 8M1 | Dneasy PowerMax Soil | 0.096 | 0.0924 | x 25 | 1.64 | 1.58 |
| 8 | Mudstone | 2 | 8M2 | Dneasy PowerMax Soil | 0.144 | 0.141 | x 25 | 1.6 | 1.55 |
| 11 | Mudstone | 1 | 11M1 | Dneasy PowerMax Soil | Below detection | Below detection | x 25 | 0.0306 | 0.0292 |
| 11 | Mudstone | 2 | 11M2 | Dneasy PowerMax Soil | Below detection | Below detection | x 25 | 0.0218 | 0.021 |
| 13 | Mudstone | 2 | 13M1 | Dneasy PowerMax Soil | Below detection | Below detection | x 25 | 0.123 | 0.119 |
| 13 | Mudstone | 3 | 13M2 | Dneasy PowerMax Soil | 0.0166 | 0.016 | x 25 | 0.141 | 0.138 |
| 2 | Scrappings | 1 | 2S1 | Power Viral | 0.167 | 0.168 | | | |
| 2 | Scrappings | 2 | 2S2 | Power Viral | 0.113 | 0.109 | | | |
| 2 | Bone | 1 | 2B1 | Power Viral | 0.134 | 0.131 | | | |
| 2 | Bone | 2 | 2B2 | Power Viral | 0.118 | 0.116 | | | |
| 6 | Scrappings | 1 | 6S1 | Power Viral | Below detection | Below detection | | | |
| 6 | Scrappings | 2 | 6S2 | Power Viral | 0.0114 | Below detection | | | |
| 6 | Bone | 1 | 6B1 | Power Viral | Below detection | Below detection | | | |

*Appendix 1—table 25 continued on next page*

*Appendix 1—table 25 continued*

| Bag number | Type | Replicate | Sample ID | Kit | Ng of DNA / μL | | Concentration | Concentrated ng of DNA / μL | |
|---|---|---|---|---|---|---|---|---|---|
| | | | | | First read | Second read | | First read | Second read |
| 6 | Bone | 2 | 6B2 | Power Viral | 0.0102 | Below detection | | | |
| 1 | Float | 2 | 1F2 | Power Viral | 0.029 | 0.0278 | | | |
| 10 | Topsoil | 2 | 10T2 | Power Viral | 1.04 | 1.02 | | | |
| 6 | Scrappings | 3 | 6S3 | Power Viral | | | x 2 | 0.208 | 0.206 |
| 6 | Bone | 3 | 6B3 | Power Viral | | | x 2 | 0.0422 | 0.0424 |
| 1 | Bone (EDTA demineralised) | 1 | 1BEDTA | Dneasy PowerMax Soil | 0.148 | 0.145 | x 20 | 3.52 | 3.44 |
| 6 | Bone (EDTA demineralised) | 1 | 6BEDTA | Dneasy PowerMax Soil | 0.0144 | 0.013 | x 20 | 0.324 | 0.318 |

**Appendix 1—table 26.** Cell abundance calculations for dinosaur bone and adjacent mudstone matrix from amino acid and DNA abundance based on *Lomstein et al. (2012)*, *Onstott et al. (2014)*, and *Magnabosco et al. (2018)*.

**Amino acids**

| | Bone | Mudstone | |
|---|---|---|---|
| picomoles/mg | 50 | 300 | picomoles/mg |
| nanomoles/g of bone | 50 | 300 | nanomoles/g of mudstone |
| g/mole | 117.4 | 113.8 | g/mole |
| grams of AA/g | 8.39E-06 | 4.88E-05 | grams of AA/g |
| g of cells/g of bone | 1.68E-05 | 9.75E-05 | g of cells/g of mudstone |
| g dry wt/cell | 4.00E-14 | 4.00E-14 | g dry wt/cell |
| cells/gram | 4.19E + 08 | 2.44E + 09 | cells/gram |

**DNA**

| | Bone | Mudstone | |
|---|---|---|---|
| ng/g | 793 | 16.4 | ng/g |
| DNA g/g of bone | 7.93E-07 | 1.64E-08 | DNA g/g of mudstone |
| DNA g/cell | 3.00E-15 | 3.00E-15 | DNA g/cell |
| cells/g of bone | 2.64E + 08 | 5.47E + 06 | cells/g of mudstone |

## Fluorescence microscopy

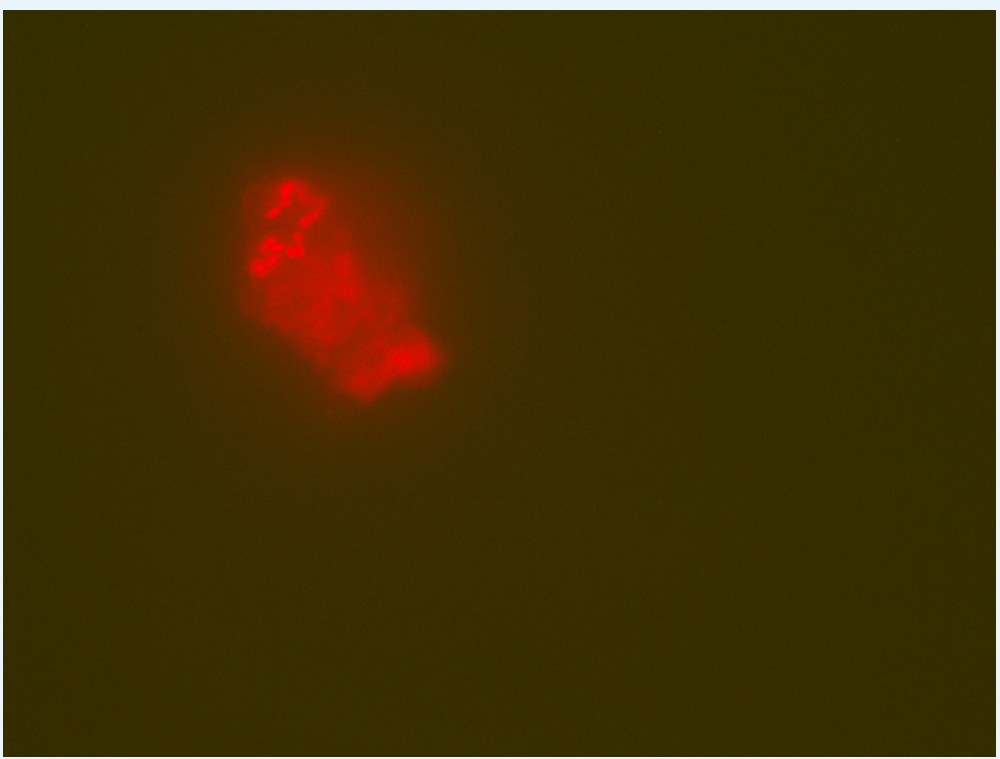

**Appendix 1—figure 54.** EDTA demineralised Late Cretaceous surface eroded fossil bone at 691 m elevation in Dinosaur Provincial Park showing possible cell clusters as might be expected in a biofilm (image 1). Sample ID as in *Appendix 1—table 2*: 16B.

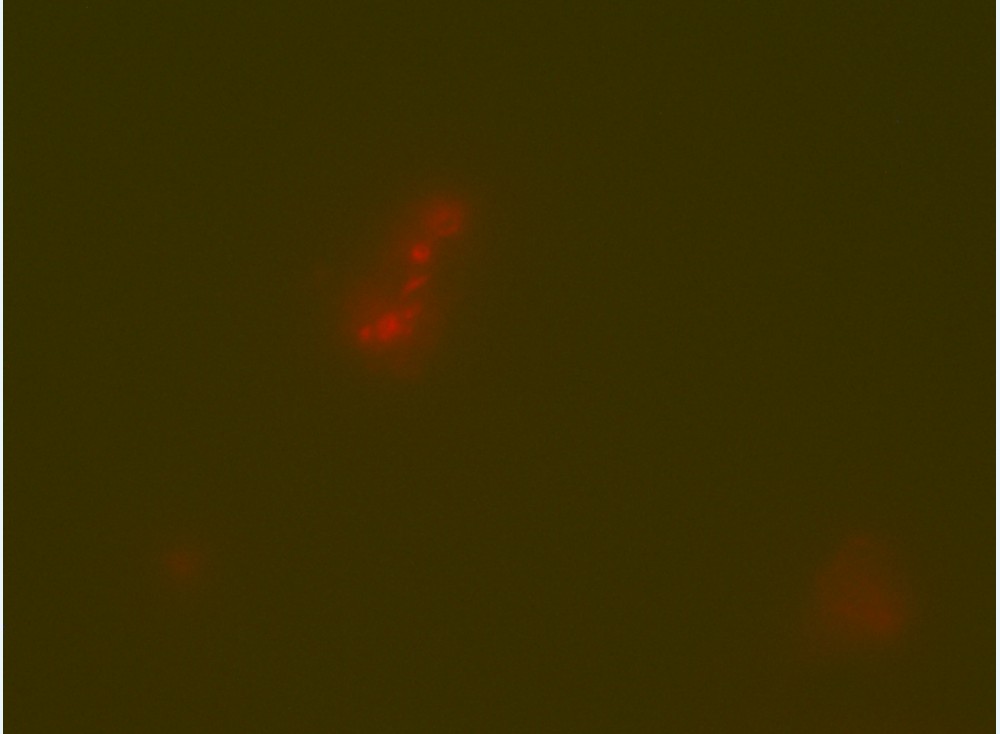

**Appendix 1—figure 55.** EDTA demineralised Late Cretaceous surface eroded fossil bone at 691 m elevation in Dinosaur Provincial Park showing possible cell clusters as might be expected in a biofilm (image 2). Sample ID as in *Appendix 1—table 2*: 16B.

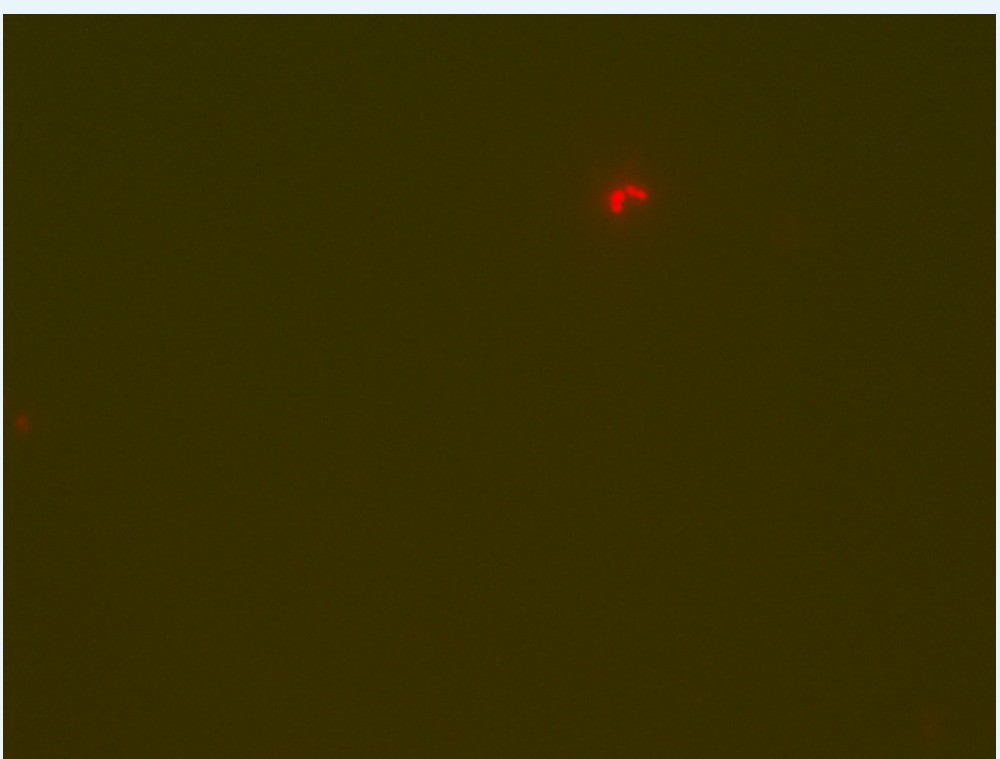

**Appendix 1—figure 56.** EDTA demineralised Late Cretaceous surface eroded fossil bone at 691 m elevation in Dinosaur Provincial Park showing possible cell clusters as might be expected in a biofilm (image 3). Sample ID as in *Appendix 1—table 2*: 16B.

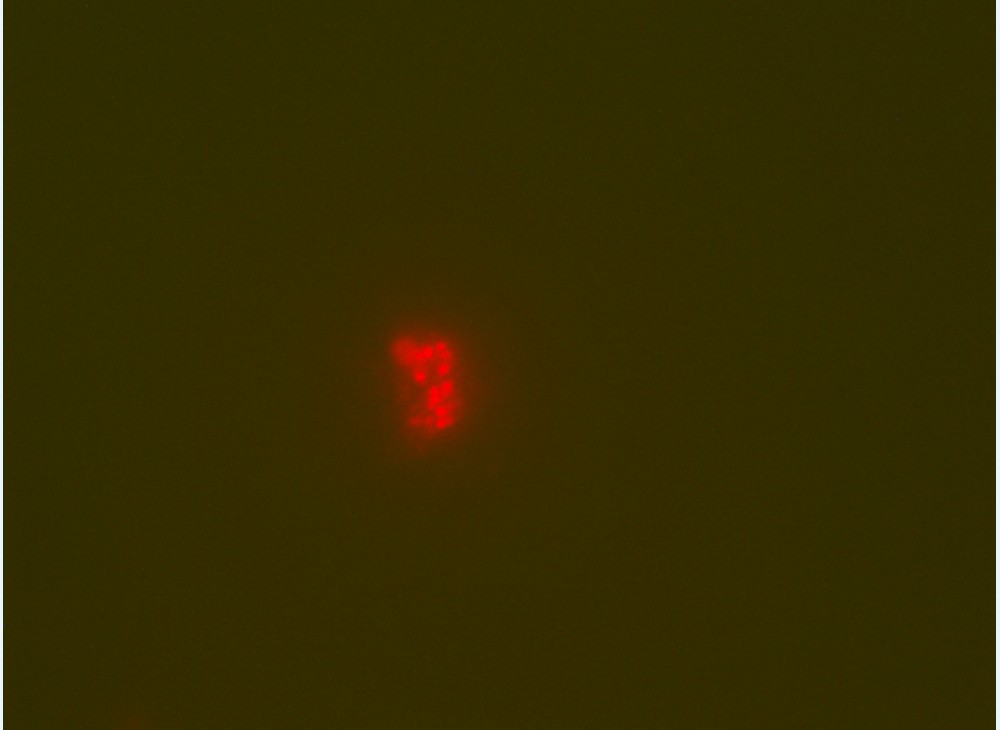

**Appendix 1—figure 57.** EDTA demineralised Late Cretaceous surface eroded fossil bone at 691 m elevation in Dinosaur Provincial Park showing possible cell clusters as might be expected in a biofilm (image 4). Sample ID as in *Appendix 1—table 2*: 16B.

# 16S rRNA amplicon sequencing

.html file containing the results summaries produced from Quantitative Insights Into Microbial Ecology (QIIME) are available through the Field Museum's collections database: multimedia record containing raw files (GUID: 60c79cec-4da1-4bea-8535-a332c70ae4c9, URI: https://mm.fieldmuseum.org/60c79cec-4da1-4bea-8535-a332c70ae4c9), event record with surrounding information about the project (GUID: 34e15532-2c46-47cf-aac0-5d29cc5a2c22, URI: https://pj.fieldmuseum.org/event/34e15532-2c46-47cf-aac0-5d29cc5a2c22) and *Source data 1*. Data is uploaded onto the Sequence Read Archive (SRA) of NCBI.

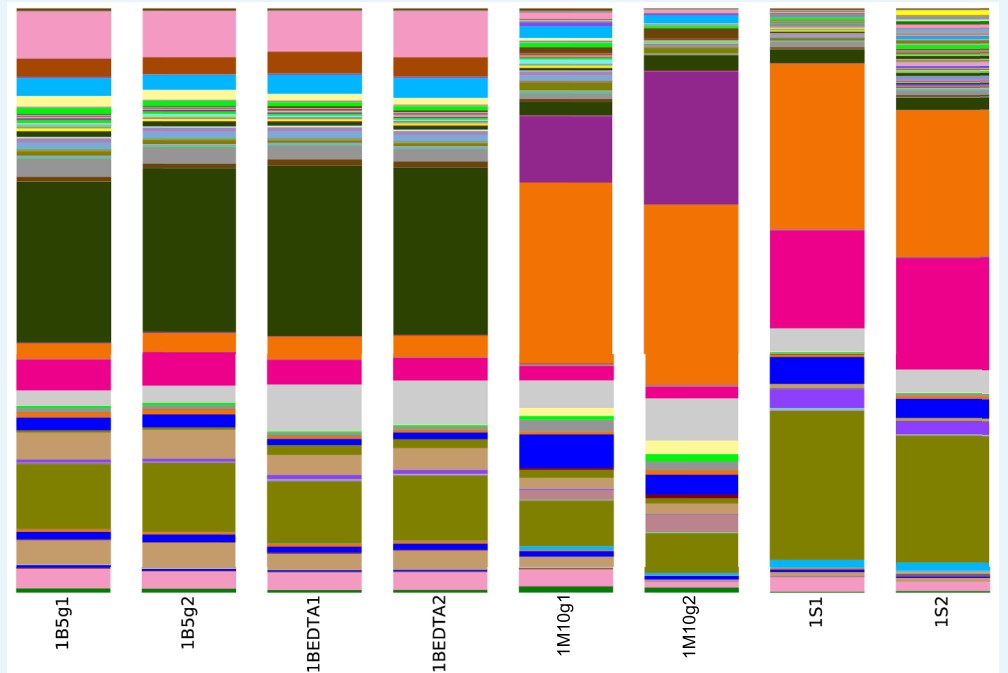

**Appendix 1—figure 58.** 16S rRNA amplicon sequence diversity at the species level of matrix-surrounded subterranean Late Cretaceous Centrosaurus bone and adjacent mudstone matrix. There are two replicates per sample. Sample ID's are as in *Appendix 1—table 2*: B = matrix surrounded subterranean *Centrosaurus* bone core (surface scraped prior to powdering; 5g indicates that this sample was analysed using the DNeasy PowerMax Soil kit), BEDTA = matrix surrounded subterranean Centrosaurus bone core (EDTA demineralised, surface scraped prior to powdering), M = adjacent mudstone matrix of subterranean *Centrosaurus* bone (10g indicates that the samples were analysed using the DNeasy PowerMax Soil kit), S = Surface scrapings from matrix-surrounded subterranean *Centrosaurus* bone. The dark green bands are sequences phylogenetically close to Euzebya. See *Source data 1* for a full listing of taxa. Both the bone core data and the EDTA demineralized bone core data are technique replicates deriving from single DNA extractions of each sample category, while both the mudstone matrix and surface scrapings data columns are from two separate DNA extractions of each sample category (*Appendix 1—table 25*).

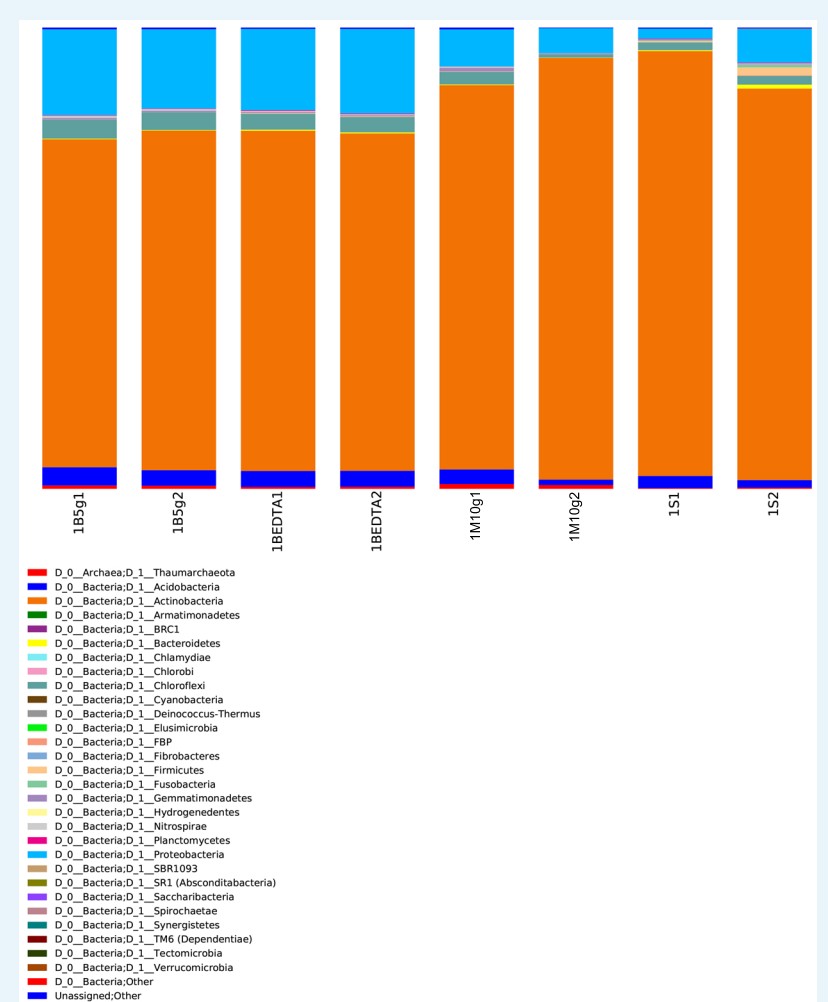

**Appendix 1—figure 59.** Comparison of microbial community (phylum level) from matrix-surrounded subterranean. Centrosaurus bone, bone scrapings, and adjacent mudstone matrix. There are two replicates per sample. Sample ID's are as in *Appendix 1—table 2*: B = matrix surrounded subterranean *Centrosaurus* bone core (surface scraped prior to powdering; 5g indicates that this sample was analysed using the DNeasy PowerMax Soil kit), BEDTA = matrix surrounded subterranean Centrosaurus bone core (EDTA demineralised, surface scraped prior to powdering), M = adjacent mudstone matrix of subterranean *Centrosaurus* bone (10g indicates that the samples were analysed using the DNeasy PowerMax Soil kit), S = Surface scrapings from matrix-surrounded subterranean *Centrosaurus* bone. Both the bone core data and the EDTA demineralized bone core data are technique replicates deriving from single DNA extractions of each sample category, while both the mudstone matrix and surface scrapings data columns are from two separate DNA extractions of each sample category (*Appendix 1—table 25*).

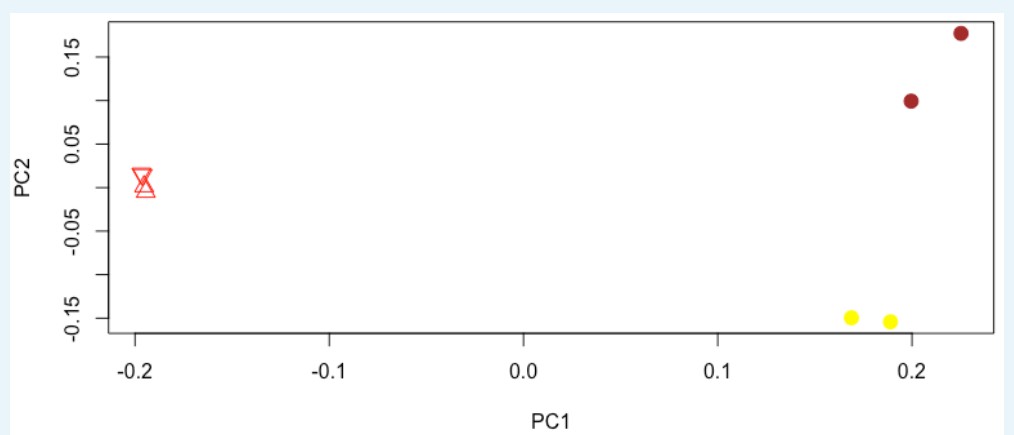

**Appendix 1—figure 60.** PCA of species-level 16S rRNA amplicon sequence data (percentages without additional normalization). Red triangles with one vertex upward are from the bone core, red triangles with one vertex downward are the EDTA demineralized bone core, yellow circles are the bone surface scrapings, and brown circles are the mudstone. PC1 and PC2 account for 75.87% and 21.65% of the variation in the data, respectively.

**Appendix 1—table 27.** Pairwise F values from PERMANOVA of species-level sequence percentages.

|  | Mudstone | Bone surface scrapings | Bone core | EDTA demineralized bone core |
|---|---|---|---|---|
| Mudstone | - | 17.62 | 47.05 | 46.28 |
| Bone surface scrapings | 17.62 | - | 162.8 | 168.6 |
| Bone core | 47.05 | 162.8 | - | 33.41 |
| EDTA demineralized bone core | 46.28 | 168.6 | 33.41 | - |

