## [Decision Letter]

Thank you for submitting your article "The microbiome inside of a late cretaceous centrosaurus dinosaur fossil bone" for consideration by *eLife*. Your article has been reviewed by two peer reviewers, and the evaluation has been overseen by a Reviewing Editor and Diethard Tautz as the Senior Editor. The following individuals involved in review of your submission have agreed to reveal their identity: Alan Cooper (Reviewer #2); Charles Lee (Reviewer #3).

The reviewers have discussed the reviews with one another and the Reviewing Editor has drafted this decision to help you prepare a revised submission.

At the most fundamental level, this manuscript explores the possibility of dinosaur fossil bones containing ancient (i.e., derived from original dinosaur soft tissues) organic materials. The authors examined this hypothesis using a combination of analytical chemistry, physical, and molecular genetic approaches. As a microbial ecologist with a cursory understanding of many of the physical and chemical techniques used, I think the results collectively make a strong case that dinosaur fossil bones do not contain ancient organic material and that any organic material found is likely associated with living microorganisms that originated from the surrounding subsurface environments.

Reviewer #2:

This is a nice study, although the choice of samples gives the impression of it being perhaps a bit haphazard and opportunistic – e.g. Florida shark tooth samples from a beach site (why Florida, why shark teeth?), vs. the bones for the other samples.

The study should be commended in terms of using multiple approaches, and finding very convincing results – but I think that more effort should be put into presenting and explaining what is going on, imagining that many readers will not be familiar with the methods, or whether certain results are, or are not, likely for something >65Ma.

Essential revisions:

Overall, the impact of the manuscript would be a lot greater if more attention was made to:

1) Signposting the meaning of tests and results, rather than waiting until the Discussion for the 'reveal'.

For those readers not used to the techniques and assumptions and logic behind these types of studies, it will be difficult to follow quite what is going on, as the delivery is very passive and technical, and readers would be left wondering what result was expected vs what was observed – and the meaning of either.

*Reviewer #3:*

I will refrain from commenting on the physical and chemical analyses, but there are a number of issues with the microbiological analyses that need to be addressed:

- Sequence data does not equal to "microbial activity" or "thriving".

- The microbial ecology analysis is very, very superficial (pun not intended). It would seem to me that OTU-based analysis (as opposed to phylum-level composition) is more appropriate for making the case that bone fossils contain unique microbial communities?!

- There is really very little insight that a microbial ecologist can gain from this manuscript in its current format, making the title "the microbiome inside of […]" a click-bait. Seems like the data support a much more conclusive title that disputes the existence of ancient collagen in bone fossils.

- I take issues with the conclusion drawn from HPLC data (subsection “Fluorescence microscopy, DNA extraction, and 16S rRNA gene amplicon sequencing, first paragraph”). Please explain and justify.

- Statistics need to be provided to justify words like "unique" and "distinct".

---

## [Author Response]

Reviewer #2:This is a nice study, although the choice of samples gives the impression of it being perhaps a bit haphazard and opportunistic – e.g. Florida shark tooth samples from a beach site (why Florida, why shark teeth?), vs. the bones for the other samples.

The justification of use of shark teeth has been added as follows: “The decision to include subfossil shark teeth was made based on their ready availability (i.e., they are incredibly common fossils and are easy to collect from the surface of the sand), the minimal loss to science when destructively analyzed due to their ubiquity, and that the protein composition of the tooth dentine would be dominated by collagen, as in bone”.

The study should be commended in terms of using multiple approaches, and finding very convincing results – but I think that more effort should be put into presenting and explaining what is going on, imagining that many readers will not be familiar with the methods, or whether certain results are, or are not, likely for something >65Ma.

See response to comment below.

Essential revisions:Overall, the impact of the manuscript would be a lot greater if more attention was made to:1) Signposting the meaning of tests and results, rather than waiting until the Discussion for the 'reveal'.For those readers not used to the techniques and assumptions and logic behind these types of studies, it will be difficult to follow quite what is going on, as the delivery is very passive and technical, and readers would be left wondering what result was expected vs what was observed – and the meaning of either.

We have added short summaries of the significance of the results at the end of each section in the Results.

Reviewer #3:I will refrain from commenting on the physical and chemical analyses, but there are a number of issues with the microbiological analyses that need to be addressed:- Sequence data does not equal to "microbial activity" or "thriving".

We have replaced “microbial activity” with “microbial presence” and “thriving” with “localized”.

- The microbial ecology analysis is very, very superficial (pun not intended). It would seem to me that OTU-based analysis (as opposed to phylum-level composition) is more appropriate for making the case that bone fossils contain unique microbial communities?!

We have moved the phylum-level figure into the appendix and replaced it with a figure in the manuscript at the class level, which we consider to be sufficient to show the distinction of microbial community between the bone and adjacent mudstone. Furthermore, the comparison of microbial community on species or OTU-level was also provided in the appendix.

- There is really very little insight that a microbial ecologist can gain from this manuscript in its current format, making the title "the microbiome inside of […]" a click-bait. Seems like the data support a much more conclusive title that disputes the existence of ancient collagen in bone fossils.

We have changed the title of the manuscript to “Cretaceous dinosaur bone contains recent organic material and provides an environment conducive to microbial communities”.

- I take issues with the conclusion drawn from HPLC data (subsection “Fluorescence microscopy, DNA extraction, and 16S rRNA gene amplicon sequencing, first paragraph”). Please explain and justify.

We have clarified this as follows, “This is fairly similar to the observed THAA concentration indicating ~3x10^8^ cells/g (calculation of cell abundance from total amino acids based on that of Onstott et al., 2014 and Lomstein et al., 2012), consistent with the idea that the amino acids within the bone are likely to be largely cellular (i.e., lipid-bound within living organisms) due to the discrepancy between DNA and amino acid stability over time”.

- Statistics need to be provided to justify words like "unique" and "distinct".

We have added one-way PERMANOVA test results and PCA of the species-level data to the manuscript and appendix, respectively. We have also altered the sentence to, “Analyses of nucleic acids reveal a diverse, unusual microbial community within the dinosaur bone, even when compared to the immediate mudstone matrix or the exterior surface of the bone, as evidenced by a strong enrichment in DNA and differing community composition in the bone relative to the surrounding matrix”.